# Mechanical properties measured by atomic force microscopy define health biomarkers in ageing *C. elegans*

Clara L. Essmann [1,2,3,4✉], Daniel Martinez-Martinez[3,4], Rosina Pryor[2,3,4], Kit-Yi Leung[5], Kalaivani Bala Krishnan [2], Prudence Pokway Lui[2], Nicholas D.E. Greene [5], André E.X. Brown[3,4], Vijay M. Pawar [1], Mandayam A. Srinivasan[1,6] & Filipe Cabreiro [2,3,4✉]

Genetic and environmental factors are key drivers regulating organismal lifespan but how these impact healthspan is less well understood. Techniques capturing biomechanical properties of tissues on a nano-scale level are providing new insights into disease mechanisms. Here, we apply Atomic Force Microscopy (AFM) to quantitatively measure the change in biomechanical properties associated with ageing *Caenorhabditis elegans* in addition to capturing high-resolution topographical images of cuticle senescence. We show that distinct dietary restriction regimes and genetic pathways that increase lifespan lead to radically different healthspan outcomes. Hence, our data support the view that prolonged lifespan does not always coincide with extended healthspan. Importantly, we identify the insulin signalling pathway in *C. elegans* and interventions altering bacterial physiology as increasing both lifespan and healthspan. Overall, AFM provides a highly sensitive technique to measure organismal biomechanical fitness and delivers an approach to screen for health-improving conditions, an essential step towards healthy ageing.

[1] Department of Computer Science, University College London, Engineering Building, Malet Place, London WC1E 7JG, UK. [2] Institute of Structural and Molecular Biology, University College London and Birkbeck, London WC1E 6BT, UK. [3] MRC London Institute of Medical Sciences, Du Cane Road, London W12 0NN, UK. [4] Institute of Clinical Sciences, Imperial College London, Hammersmith Hospital Campus, Du Cane Road, London W12 0NN, UK. [5] UCL Great Ormond Street Institute of Child Health, University College London, London WC1N 1EH, UK. [6] Department of Mechanical Engineering and Research Laboratory of Electronics, Massachusetts Institute of Technology, Cambridge, MA 02139, USA. ✉email: c.essmann@ucl.ac.uk; f.cabreiro@lms.mrc.ac.uk

The major burden of ill health falls on older people, and their increasing proportion in most developed societies highlights an urgent challenge to find ways of maintaining health for longer. Ageing has some consistent hallmarks, and is malleable to genetic, dietary and pharmacological interventions[1]. Although the effects on lifespan for a few genetic, dietary and pharmacological interventions are well-documented and show improvements in health measures at chronological age[2–5], it is less clear whether lifespan and healthspan are increased in the same proportion. One of the great challenges in ageing research that has slowed progress in developing healthspan-improving strategies is the lack of an accurate definition of healthspan and of reliable health biomarkers. In humans, several frailty indices exist to characterise different inter-related pathophysiological parameters that are associated with the loss of health during ageing and encompass visible exterior signs of decay to identify the elderly, such as unintentional loss of body mass, the weakening of body tissues including muscle and skin, and functional decay such as slow walking speed[6,7]. Similarly, ageing in *Caenorhabditis elegans* is also accompanied by a decrease in muscle function[8], loss of maximum velocity[9], body size shrinkage[10], increased organ deterioration[11,12] and changes in cuticle collagen production[13]. Despite similarities in health-related decay between evolutionarily distant organisms, the adoption of a frailty index in worms and other model organisms is rarely implemented[4,10]. Partly, this is because currently used health biomarker measurements remain loosely defined in the context of the ageing process and are prone to human or biological bias, suggesting the need for additional health biomarkers for the assessment of organismal healthspan with age.

The mechanical properties of cells and tissues are linked to important physiological processes, including proliferation, morphogenesis, stem cell differentiation, migration and adhesion. Cells are under constant mechanical pressure from shear stress and contractile stress from their own acto-myosin cytoskeleton and from neighbouring cells, through confinement in space, gravity or hydrostatic pressure. These factors have been shown to influence key molecular processes such as ion channel activity, gene regulation and protein expression[14,15]. In fact, dysregulation of the biomechanical processes that maintain cellular homoeostasis can lead to alterations of the molecular and physiological properties of tissues and ultimately organ function, giving rise to disease states. More recently, the focus has shifted towards understanding how changes in mechanical properties of cells may underlie pathologies in many organs or lead to tissue decay during ageing[16,17].

Here, we employ atomic force microscopy (AFM) to quantitatively assess changes in body frailty of worms in vivo during ageing and propose stiffness and cuticle senescence as biomarkers for ageing and healthspan. This work paves the way for the future dissection of the molecular pathways underlying healthspan, and potentially tilts the balance in the paradigm of healthspan gains versus lifespan extension.

## Results

**Stiffness and cuticle quality decrease with age**. Among ageing worm cohorts, a gradual loss in physical characteristics and increased frailty is readily apparent. However, parameters such as cuticle senescence and loss of body tension have yet to be used to objectively assess ageing or the health status of an animal. We adapted a recently published protocol[18] to test the feasibility of these ageing parameters as healthspan readouts by using AFM to quantitatively measure worm stiffness and cuticle senescence (roughness) over time. AFM is a type of scanning probe microscope with a resolution in the order of a few nanometres, which utilises the fine probe to: (1) detect changes in the mechanical

properties of a sample through force-indentation measurements (Fig. 1a, Supplementary Fig. 1a, Supplementary Data 1), which can be used to calculate the stiffness or Young's Modulus (YM) using the Hertz model for contact mechanics (Supplementary Fig. 1b); and (2) perform high-resolution imaging of the surface of a sample, which can be illustrated by three-dimensional topographic images.

To determine changes in stiffness and cuticle senescence with ageing, we set up lifespan assays (Fig. 1b, Supplementary Data 2) and recorded these two parameters in ageing wild-type worm populations over time (Fig. 1c, f–h, Supplementary Data 1). Performing quantitative analysis of force-indentation measures in ageing cohorts revealed a gradual decline in worm stiffness with age, as older worms show greater indentation values when a set force is applied (Supplementary Fig. 1b). The YM calculated from these indentation curves show the same decrease with age (Fig. 1c, Supplementary Fig. 1b, c, Supplementary Data 1). YM measures were reproducible between technical replicates from young and aged worms with an effect size of 1.00–1.47 fold-change (FC) ($t$ test $p$ value range $= 0.1602$–$0.9999$, post hoc power range $= 0.03$–$0.68$, Supplementary Fig. 1d, Supplementary Data 1). Based on the sample size and population variance, post hoc analysis suggests that our experiment was powered to detect a statistically significant difference at a fold-change $> 1.8$. Despite variability between biological replicates, similar trends in these measurements were observed in ageing cohorts (Supplementary Fig. 1c, Supplementary Data 1). In addition, these measurements were not affected by the compound 2,3-butanedione monoxime (BDM) (FC $= 1.03$, paired $t$ test $p = 0.475$, Supplementary Fig. 1e, Supplementary Data 1) used to paralyse the worms. As ageing is characterised by a marked decline in physical movement, a commonly used healthspan marker, we performed force-indentation measurements in aged cohorts of day 16 worms that were either moving freely (A), showing partial movement (B) or no movement (C) when prodded[19,20]. Calculating the mechanical properties from these measurements, our results show that healthier moving worms maintain higher stiffness with age (A vs B FC $= 2.19$, $t$ test $p < 0.0001$; A vs C FC $= 3.72$, $t$ test $p < 0.0001$, Supplementary Fig. 1f, Supplementary Data 1) further validating our approach. As temperature is a major factor determining animal lifespan[21,22], we investigated whether temperature influenced body stiffness. We observed an 88% and 44% increase in mean lifespan of worms grown at 15 and 20 °C compared with 25 °C, respectively, and a 30% increase for worms grown at 15 °C compared with 20 °C (Fig. 1d, Supplementary Data 2). We observed a statistically significant improvement in stiffness of 4.49-fold ($t$ test $p < 0.0001$) between worms grown at 15 °C compared with 20 °C but not between 25 and 20 °C measured on day 15 (25 vs 20 °C FC $= 1.19$, $t$ test $p = 0.1099$, post hoc power $= 0.36$, Fig. 1e, Supplementary Data 1). Post hoc analysis suggests that a significant difference between conditions would be observed for changes higher than 1.5-fold, indicating that the true effect of 25 °C is likely under 1.5-fold when compared with worms grown at 20 °C and well under the significant effects observed for worms grown at 15 °C.

Similar to our stiffness measurements, topographical images of the cuticle show a progressive decline of cuticle integrity with increasing age including the presence of severe breaks and cavities in the cuticle of day 19 animals compared with young day 1 adult worms (Fig. 1f), quantifiable as increased roughness (Fig. 1g, Supplementary Fig. 1g). Topographical images taken on day 1 and 19 of adulthood also illustrate a flattening of the annuli furrow structure with age, which are very prominent on day 1 adults (Fig. 1h). Overall, our data suggest that the mechanisms underlying the preservation of body stiffness and cuticle integrity are distinctively regulated with age.

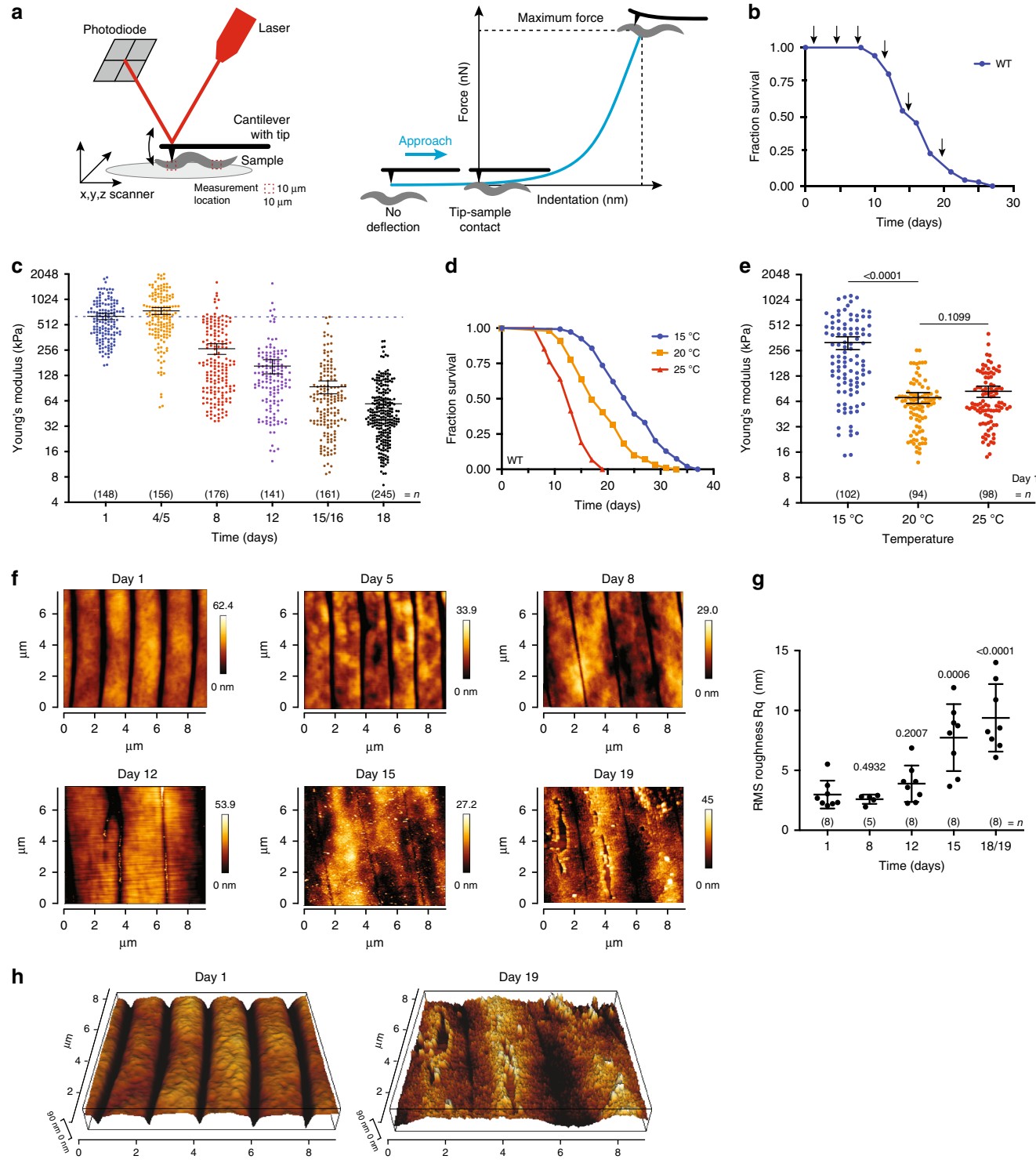

**Insulin pathway maintains stiffness and roughness with age.**
Genetic manipulations have been thoroughly applied to worms to investigate the role of genes in modulating ageing[23]. We investigated the role of the insulin/IGF-1 signalling pathway (IIS), which has been reported to markedly increase lifespan in worms[23–25]. However, whether reduced IIS signalling caused by mutations in the worm DAF-2 insulin receptor also leads to a proportional increase in healthspan remains a matter of debate[2,5,9]. Here, we used our AFM protocol to investigate the role of the DAF-2 insulin/IGF receptor and the transcriptional

regulator DAF-16/FOXO in maintaining stiffness and cuticle integrity with ageing. Consistent with previously reported data[23], the *daf-2(e1370)* mutation robustly extended lifespan (165% increase in mean lifespan compared with wild-type, log rank $p < 0.001$) in a *daf-16*-dependent manner (0.48% mean lifespan increase of *daf-2;daf-16* vs *daf-16*, log rank $p = 0.6554$; cox proportional hazards (CPH) < 0.001 for the interaction of terms: *daf-2* and *daf-16* Fig. 2a, Supplementary Data 2). Stiffness did not differ between *daf-2* mutant worms and day 1 wild-type worms (FC = 1.08, $t$ test $p = 0.3605$, Fig. 2b, Supplementary Data 1).

**Fig. 1 Atomic force microscopy (AFM) captures cuticle and stiffness decay of ageing *C. elegans*. a** Schematics show an AFM laser beam deflection system and a resulting force-indentation curve. The cantilever bends upon increasing force detected by the deflection of the laser beam reflected from the cantilever back onto a photodiode (left). Typical AFM force-indentation approach curve indicating stages of tip-sample interaction shown in blue (right). **b** Lifespan curve of wild-type (WT) *C. elegans* ($n = 222$). Arrows indicate time points at which AFM measurements were performed—day 1, 5, 8, 12, 15 and 18 of adulthood. **c** Mechanical properties as Young's Modulus (YM; kPa) of WT *C. elegans* at different ages. Error bars indicate 95% confidence intervals (CI), dotted line marks mean YM at day 1. **d** Lifespan curve of WT *C. elegans* grown at 20 °C and maintained at 15 °C (blue), 20 °C (yellow) or 25 °C (red) from the L4 stage throughout their entire lifespan ($n = 148, 99, 89$, respectively; log rank test $p < 0.001$ vs 20 °C) and **e** mechanical properties as YM (kPa) at day 15. Error bars indicate 95% CI. Two-tailed unpaired *t* test for statistical comparison of WT at 20 °C to 15 °C or 25 °C. **f** Representative AFM cuticle topography images of ageing *C. elegans* at different ages. **g** Roughness quantification of topographical images presented as RMS roughness Rq ± standard deviation. Two-tailed unpaired *t* test for statistical comparison with day 1. **h** Three-dimension representation of AFM cuticle topography of *C. elegans* on day 1 and 19 of adulthood. *n* represented above the graph, show number of biologically independent worm samples; For lifespan measurements, *n* represents the number of worms scored as dead. For a summary of YM values and additional statistics for independent trials see Supplementary Data 1 and for a summary of worm lifespan trials see Supplementary Data 2. Source data are provided as a Source Data file.

However, long-lived *daf-2* mutant worms were better at maintaining stiffness with ageing than wild-type worms. Hence, comparison of the YM derived from force-indentation measurements of chronologically matched *daf-2* mutant and wild-type worms showed a gradual loss in stiffness over time for both genotypes, however within distinct time frames (Fig. 2b, Supplementary Data 1). In addition, *daf-2* mutant worms showed significantly higher YM values at all ages compared with wild-type past day 4 with fold differences in the range of 3.79–10.3 (*t* test $p < 0.0001$, Fig. 2b, Supplementary Data 1). Comparison between physiologically matched animals at approximately their mean lifespan value (day 46 for *daf-2* and day 18 for wild-type) showed that reduction of the IIS significantly preserved stiffness with age by 4.01-fold (*t* test $p < 0.0001$, Fig. 2b, Supplementary Data 1). Overall, these measures show that reduced insulin signalling in worms increased their stiffness both at chronological and physiological age.

As the longevity of *daf-2* mutants depends on the FOXO family transcription factor DAF-16[23] (Fig. 2a), we tested if *daf-16 (mgDf50)* also mediates the observed effects on stiffness imposed by the reduction of insulin signalling. We found only small fold changes and no significant differences when comparing YM values of day 1 old animals between genotypes (*daf-2;daf-16* vs WT FC = 0.82, analysis of variance (ANOVA) $p = 0.2184$; *daf-16* vs WT FC = 0.79, ANOVA $p = 0.1226$; Supplementary Fig. 2a, Supplementary Data 1), and when comparing *daf-16* mutants and wild-type over time (day 6 FC = 1.22, *t* test $p = 0.2332$; day 10 FC = 0.94, *t* test $p = 0.7188$; day 13 FC = 1.05, *t* test $p = 0.8203$; Supplementary Fig. 2b, Supplementary Data 1). Instead, measurements performed at day 11 showed a statistically significant 4.19-fold increase in stiffness for *daf-2* single mutants, and a lower but significant 2.02-fold increase for *daf-2;daf-16* double mutant worms compared with wild-type (ANOVA *daf-2* vs WT $p < 0.0001$; *daf-2;daf-16* vs WT $p = 0.0028$). There was a significant interaction between the effects of *daf-16* and *daf-2* on worm stiffness with an effect size of 2.17-fold, which was obtained by subtracting the fold-change in stiffness of *daf-2* mutants compared with wild-type from the fold-change in stiffness of double mutants *daf-2;daf-16* compared with *daf-16* mutants (interaction effect size = 4.19-fold for FC between *daf-2* and WT; – 2.02 for FC between *daf-2;daf-16* and *daf-16*). Together with a significant value of $p < 0.0001$ for the interaction of terms, our data suggest that the effects of *daf-2* on stiffness maintenance with age are modulated by *daf-16* (Fig. 2c, Supplementary Data 1).

Ablating the germline of worms using laser microsurgery or with genetic mutations such as *glp-1* make adult animals live longer than intact controls by inhibiting signals emitted from proliferating germ cells that reduce longevity in a *daf-16*-dependent manner[26,27]. Therefore, to further investigate the role

of DAF-16 in the insulin signalling-mediated effects on lifespan and stiffness, we investigated the well-characterised germline-less mutant *glp-1(e2141)*, whose effects on healthspan have not been explored. As previously reported, genetic ablation of the germline robustly extended lifespan (67% mean lifespan extension compared with wild-type, log rank $p < 0.001$) in a *daf-16*-dependent manner (0.2% mean lifespan decrease of *glp-1;daf-16* versus *daf-16*, $p = 0.5984$; CPH $< 0.001$ for the interaction of terms: *glp-1* and *daf-16* Fig. 2d, Supplementary Data 2). Similar to the observed effects of *glp-1* on lifespan, force measurements at day 10 and at mean lifespan showed that *glp-1* mutants were considerably stiffer with age compared with wild-type animals (day 10 FC = 3.84, ANOVA $p < 0.0001$; at mean lifespan FC = 4.26, *t* test $p < 0.0001$, Fig. 2e, f, Supplementary Data 1), despite being softer as young day 2 adults (FC = 0.67, ANOVA $p < 0.0001$, Supplementary Fig. 2c, Supplementary Data 1). As observed for *daf-2* mutants, the maintenance of stiffness of *glp-1* mutants with age was also modulated by *daf-16* as suggested by the effect size for the interaction between *glp-1* and *daf-16* = 2.13-fold, and a statistical significance of $p < 0.0001$ for the interaction of terms (Fig. 2e, Supplementary Data 1).

When assessing cuticle roughness with ageing, comparing genotypes at physiological mean lifespan (Fig. 2g, h) revealed that *daf-2* and *glp-1* animals have reduced cuticle senescence, despite no observable differences at day 1 (Supplementary Fig. 2d). We conclude that the reduction of insulin signalling not only increases lifespan, but also importantly reduces physical decay driven by ageing in a *daf-16*-dependent manner.

**Dietary restriction controls stiffness and roughness.** Dietary restriction (DR) increases lifespan throughout the animal kingdom from invertebrates to mammals, and delays age-related diseases in many organisms[1]. There are multiple ways to achieve DR in worms including the use of genetic mutations affecting food intake, DR mimetic drugs, absence of food or low-calorie food sources[28]. Even though DR reproducibly leads to increases in lifespan, its impact on healthspan or the physical state of the animal has not been thoroughly assessed. For this purpose, we tested two mutants that affect feeding: *eat-2* mutants carrying a genetic mutation known to impact food uptake through reduction of pharyngeal pumping[29], and a *phm-2* mutant carrying a genetic mutation involved in the positioning and morphology of the pharyngeal grinder, and a complete removal of food protocol also known as bacterial deprivation (BD)[30].

As in *eat-2* mutants and BD animals, *phm-2* mutants show all the characteristics of DR in worms, such as extended lifespan (Fig. 3a, Supplementary Data 2), small and thin-transparent morphology (Supplementary Fig. 3a), reduced brood sizes (Supplementary Fig. 3b) and delayed reproductive output

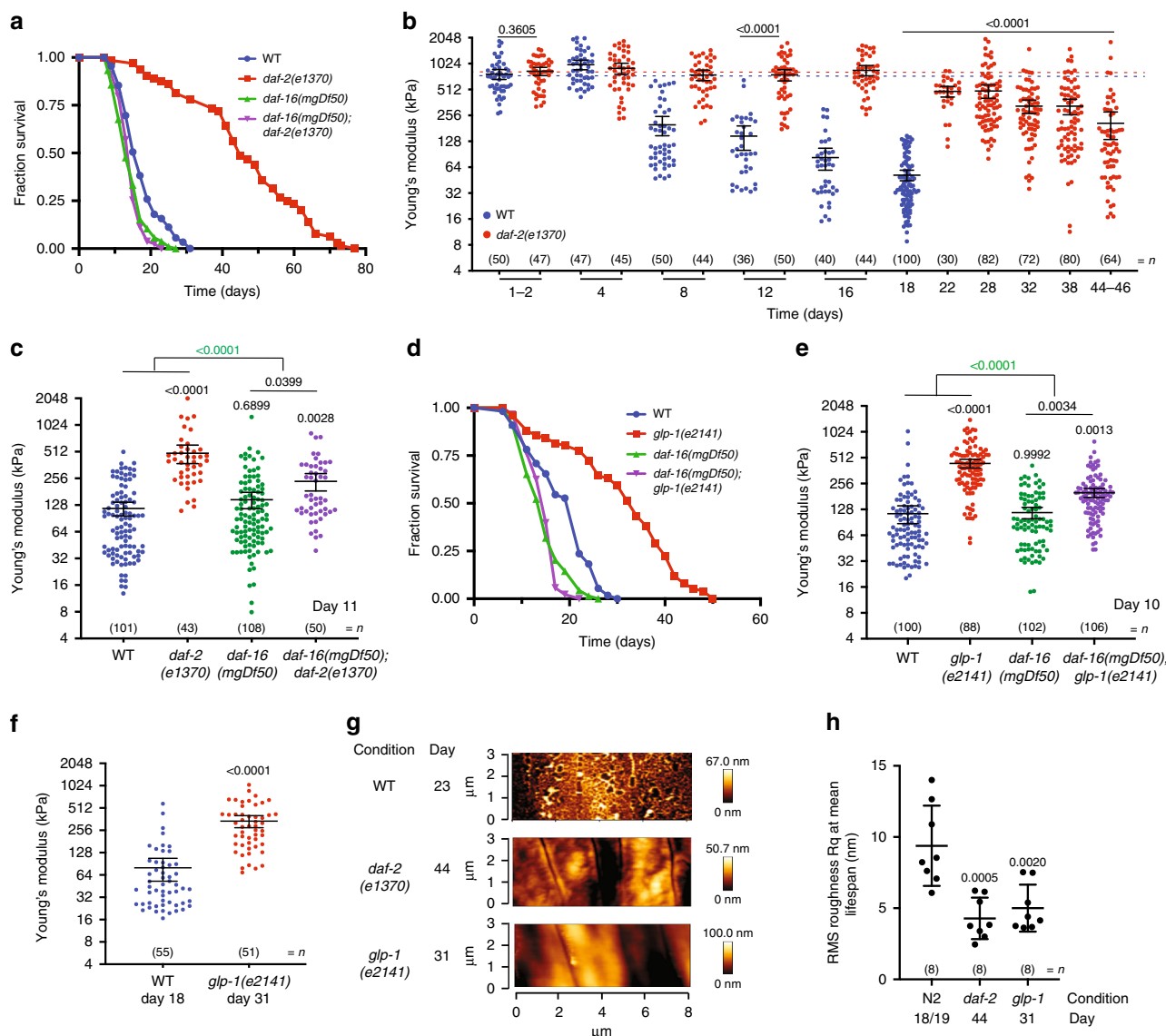

**Fig. 2 Reduction of the insulin signalling pathway increases lifespan, improves stiffness and reduces cuticle senescence with age. a** Lifespan curve of wild-type (WT) (blue), or *daf*-2 (red), *daf*-16 (green), *daf*-16;*daf*-2 (purple) mutant *C. elegans*. (*n* = 89, 64, 105 and 79, respectively; log rank test *p* < 0.001 vs WT). **b** Longitudinal study of mechanical properties as Young's Modulus (YM; kPa) comparing WT (blue) and *daf*-2 mutant (red) *C. elegans* until mean lifespan (D18 for WT; D44/46 for *daf*-2). Error bars indicate 95% CI, dotted line marks mean YM at day 1 for WT (blue) and *daf*-2 (red). Two-tailed unpaired *t* test for statistical comparison of WT to *daf*-2 at chronological age day 1, 12 or at mean lifespan. **c** Mechanical properties as YM (kPa) of WT (blue), *daf*-2 (red), *daf*-16 (green) and *daf*-16;*daf*-2 (purple) mutant *C. elegans* at chronological age of 11 days. Error bars indicate 95% CI. Two-way ANOVA Tukey's multiple comparison test for statistical comparison and interaction of terms (green). **d** Lifespan curve of WT (blue), *glp*-1 (red), *daf*-16 (green) and *daf*-16;*glp*-1 (purple) mutant *C. elegans*. (*n* = 55, 76, 69 and 88, respectively; log rank test *p* < 0.001 vs WT) and **e** mechanical properties as YM (kPa) at chronological age of 10 days. Two-way ANOVA Tukey's multiple comparison test for statistical comparison and interaction of terms (green). **f** Mechanical properties as YM (kPa) of WT (blue), or *glp*-1 (red) mutant *C. elegans* at mean lifespan. Two-tailed unpaired *t* test was used for statistical comparison. **g** Representative AFM cuticle topography images of WT, *daf*-2 or *glp*-1 mutant at mean lifespan and **h** roughness quantification of topographical images presented as RMS roughness Rq ± standard deviation. Two-tailed unpaired *t* test for statistical comparison of WT with mutants. *n* represented above the graph, show number of biologically independent worm samples. For lifespan measurements, *n* represents the number of worms scored as dead. For a summary of YM values and additional statistics for independent trials see Supplementary Data 1 and for a summary of worm lifespan trials and statistical comparison between genotypes see Supplementary Data 2. Source data are provided as a Source Data file.

(Supplementary Fig. 3c). In addition, these worms accumulate intact bacteria in their intestinal tract earlier than wild-type, due to their inability to masticate bacteria, which is consistent with reduced nutrient uptake (Supplementary Fig. 3d–f). Given that all these DR interventions lead to an increase in lifespan (Fig. 3a, Supplementary Fig. 3g, Supplementary Data 2), whereas in contrast, *eat*-2 mutants improve some healthspan markers such as reproductive span[4] but not others such as movement and

resistance to stress[2], we set out to investigate whether differences could be observed in healthspan measures such as stiffness and cuticle senescence.

We observed that the decline in stiffness over time is different between the diverse DR interventions tested (Fig. 3b, Supplementary Fig. 3h, Supplementary Data 1). Interestingly, *phm*-2 mutants that have a similar lifespan extension to that of *eat*-2 mutants (27.9% and 32.63% increase in mean lifespan,

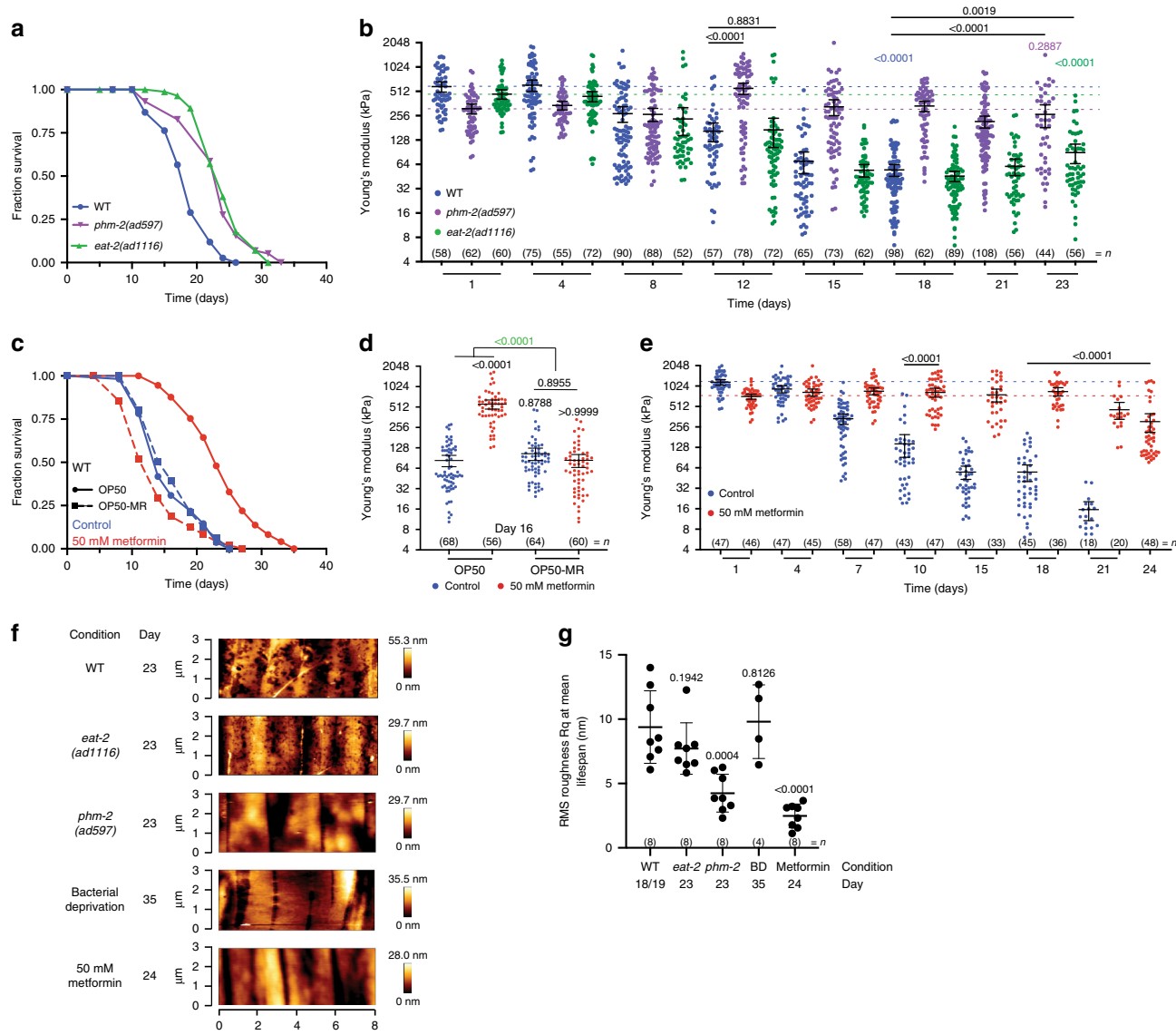

**Fig. 3 AFM identifies differences in stiffness and cuticle senescence with age among diverse dietary restriction regimens. a** Lifespan curves of wild-type (WT) (blue), or *eat-2* (green) and *phm-2* (purple) feeding impaired mutant *C. elegans* ($n = 98$, 104 and 125, respectively; log rank test $p < 0.001$ vs fully-fed WT) and **b** longitudinal study of mechanical properties as Young's Modulus (YM; kPa) until mean lifespan (D18 for WT; D23 for *eat-2* and *phm-2*). Error bars indicate 95% CI, dotted lines mark mean YM at day 1 for WT (blue), *eat-2* (green) or *phm-2* (purple). Two-tailed unpaired *t* test for statistical comparison of WT with *eat-2* or *phm-2* at chronological age day 12 and at mean lifespan, WT at mean lifespan to day 1 (blue), *phm-2* at mean lifespan to day 1 (purple), and *eat-2* at mean lifespan to day 1 (green). **c** Lifespan curve of untreated (0 mM) (blue) and metformin-treated (50 mM) (red) WT *C. elegans* grown on sensitive OP50 (full line) or grown on metformin-resistant OP50-MR (dotted line). ($n = 107$, 142, 119 and 175, respectively; log rank test $p < 0.001$, $p = 0.0883$ and $p = 0.0004$ vs OP50 0 mM) and **d** mechanical properties as YM (kPa) at chronological age of 16 days. Error bars indicate 95% CI. Two-way ANOVA Tukey's multiple comparison test for statistical comparison and interaction of terms (green). **e** Longitudinal study of mechanical properties as YM (kPa) of untreated (0 mM) (blue) and metformin-treated (50 mM) (red) WT *C. elegans* until mean lifespan (D18 for WT; D24 for metformin-treated). Error bars indicate 95% CI, dotted lines mark mean YM at day 1 for untreated (blue) and metformin-treated (red). Two-tailed unpaired *t* test for statistical comparison of untreated to metformin-treated at chronological age day 10 and mean lifespan. **f** Representative AFM cuticle topography images of WT, *eat-2* and *phm-2* mutant worms, and WT under bacterial deprivation or metformin-treated at mean lifespan. **g** Roughness quantification of topographical images presented as RMS roughness Rq ± standard deviation. Two-tailed unpaired *t* test for statistical comparison of WT with mutants/conditions. *n* represented above the graph, show number of biologically independent worm samples. For lifespan measurements, *n* represents the number of worms scored as dead. For a summary of YM values and additional statistics for independent trials, see Supplementary Data 1 and for a summary of worm lifespan trials and statistical comparison between genotypes see Supplementary Data 2. Source data are provided as a Source Data file.

respectively, compared with wild-type; log rank $p < 0.001$, Supplementary Data 2) maintained their stiffness with age (day 23 vs day 1 FC = 0.85, *t* test $p = 0.2887$), unlike both wild-type (FC = 0.01, *t* test $p < 0.0001$) and *eat-2* mutants (FC = 0.19, *t* test $p < 0.0001$; Fig. 3b, Supplementary Data 1). Comparison between physiologically matched *phm-2* or *eat-2* mutants and wild-type

worms at approximately their mean lifespan age (day 17 for wild-type and day 23 for *phm-2* and *eat-2*) shows that both mutants have higher stiffness than wild-type, but a greater magnitude of effect was observed for the *phm-2* mutant (*phm-2* vs WT FC = 4.86, *t* test $p < 0.0001$; *eat-2* vs WT FC = 1.64, *t* test $p = 0.0019$, Fig. 3b, Supplementary Data 1). Force measurements in worms

under BD show that even if very long-lived under these conditions (90% increase mean lifespan compared with wild-type; log rank $p < 0.001$, Supplementary Data 1) stiffness was not significantly improved at mean lifespan (FC = 0.87, $t$ test $p = 0.3700$, Supplementary Fig. 3h, Supplementary Data 1). Overall, our data suggest that distinct DR interventions impact stiffness differently with age.

DR can also be achieved by treating animals with DR mimetic drugs. The anti-diabetic drug metformin is a putative DR mimetic that has been shown to regulate health and increase lifespan in worms and mice[31]. Consistent with its DR effect in worms, metformin does not further extend the lifespan of phm-2 (Supplementary Fig. 4a, Supplementary Data 2) or eat-2 mutants, as previously shown for the latter[32]. Metformin also improves healthspan as measured by movement, pharyngeal pumping rates[32] and cuticle senescence[33]. More recently, the effects of metformin on health improvements have been linked to a modulation of the gut microbiota[34–37]. We sought to investigate whether the effects of metformin on host healthspan were first in line with our stiffness measurements, and second, whether they depend on the interaction of the drug with microbial metabolism.

Consistent with our previously published data[34,36], we show that worms treated with metformin are only long-lived compared with non-treated worms if grown in the presence of control Escherichia coli OP50 but not OP50-MR, which have developed resistance against the effects of metformin on their metabolism (Fig. 3c, Supplementary Data 2). YM values of day 16 worms under these conditions show that metformin preserves stiffness of worms with age through a bacteria-dependent mechanism (Fig. 3d, Supplementary Data 1). Here, YM values were significantly higher by 6.77-fold for wild-type worms treated with metformin grown on OP50 (ANOVA $p < 0.0001$), but not on OP50-MR (FC = 0.8, $p = 0.8955$). The effect size for the interaction of terms metformin and bacterial strain was 6.97-fold and statistically significant (ANOVA $p < 0.0001$, Fig. 3d, Supplementary Data 1). This implies a bacteria-dependent mechanism for the effects of metformin on worm stiffness. Strikingly, longitudinal stiffness measurements show that worms treated with metformin and grown on OP50 maintained their stiffness over time, and were significantly stiffer by 5.54-fold at their mean lifespan age of day 24 when compared with control worms at mean lifespan age ($t$ test $p < 0.0001$, Fig. 3e, Supplementary Data 1). Also, late-life administration of metformin after the reproductive period still increases lifespan[33,34] and maintained stiffness significantly by 3.02-fold ($t$ test $p < 0.0001$, Supplementary Fig. 4b, Supplementary Data 1). Therefore, our findings show that metformin acts on worms through bacteria to improve not only their lifespan, but also to maintain their stiffness chronologically with age and at mean lifespan.

As DAF-16 mediates the effects of reduced IIS on stiffness (Fig. 2c), we tested whether it also regulated the effects of metformin. Consistent with previous observations[32], metformin increased lifespan independently of daf-16 (24.04% mean lifespan increase of metformin + daf-16 versus daf-16 log rank $p < 0.001$; CPH < 0.097 for the interaction of terms: metformin and daf-16 Supplementary Fig. 4c, Supplementary Data 2). Similarly, comparing stiffness measures of metformin-treated aged wild-type and daf-16 mutant worms, metformin did not require daf-16 to maintain stiffness during ageing as shown by the small effect size of 0.79-fold between the terms daf-16 and metformin and the lack of statistical significance (ANOVA $p = 0.4473$, Supplementary Fig. 4d, Supplementary Data 1).

We next evaluated cuticle senescence of these DR mutants and treatments, and obtained topographical images of the cuticle at mean lifespan age (Fig. 3f), which were the same at day 1 (Supplementary Fig. 3i). Similar to the data on stiffness, we observed a strong difference in cuticle senescence between diverse DR interventions with age with phm-2 mutants showing less cuticle decay compared with wild-type worms, eat-2 mutants or worms under bacterial deprivation (Fig. 3f, g). We also find that the cuticle of worms on metformin appears healthier at mean lifespan (Fig. 3f) and is quantifiably less rough (Fig. 3g) compared with that of wild-type worms or other DR interventions at mean lifespan. This supports previously reported findings on worms treated with metformin and obtained by electron microscopy[33], and further validates our approach to measure cuticle health in aged worms. Altogether, our data suggest the requirement of bacterial nutrients for adequate maintenance of the cuticle with age. Overall, our data show that DR interventions, despite leading to similar outcomes in terms of lifespan can lead to distinct outcomes when evaluating healthspan parameters such as stiffness and cuticle senescence using AFM.

**Bacteria are a key driver of host stiffness and roughness.** The microbiota is an important environmental factor regulating host health[38,39]. Notably, treatments that alter bacterial physiology such as UV irradiation, genetic deletions, antibiotic supplementation or host-targeted drugs like metformin extend worm lifespan[3,34,40]. In addition, treatments preventing bacterial colonisation of the gut can chronologically and proportionally increase health of daf-2 mutants compared with wild-type worms[5]. Here, we explored in detail the contribution of bacterial physiology to host stiffness and cuticle maintenance during ageing using disruptive methods such as treatment with UV, heat, antibiotics (e.g. carbenicillin, trimethoprim), or nutritional approaches by altering the source of protein for bacterial growth and targeted genetic deletions of metabolic genes in E. coli, all aimed at modulating distinct aspects of bacterial physiology such as metabolism and proliferation parameters including total growth (total bacterial population over time, measured as area under the curve) and growth rate (maximum rate of bacterial population doubling, measured as $t^{-1}$) (Supplementary Table 1).

First, we impaired bacterial physiology using heat and UV treatment methods and assessed their effect on host lifespan and stiffness. UV treatment leads to a non-proliferative state in bacteria with altered cellular metabolism[41], whereas heat treatment leads to a cellular collapse owing to loss of proteins with key functions in maintaining cellular homoeostasis[42] resulting in a non-proliferative state with severe loss of metabolic properties[43,44]. These previous observations were first confirmed in our experimental setting using E. coli OP50. Both treatments altered metabolism to distinct degrees (Supplementary Fig. 5c, d, Tables 1 and Supplementary Data 3) and fully inhibited both total growth and growth rate (Supplementary Table 1, Supplementary Fig. 5a, b). In particular, heat treatment led to significant loss of metabolites measured in E. coli (172 out of 228; 75%) compared with UV treatment (73 out of 228; 32%) and control condition (Bactopeptone—40 out of 228; 18%) (Supplementary Fig. 5e, Supplementary Data 3). Our lifespan measurements of wild-type worms maintained on UV- and heat-treated bacteria show that both treatments robustly increase lifespan (65.28% and 38.42%, mean lifespan extension compared with wild-type non-treated E. coli respectively, log rank $p < 0.001$, Fig. 4a, Supplementary Data 2). In contrast, stiffness measures were only significantly higher by 3.67-fold for wild-type worms fed UV-treated E. coli OP50 ($t$ test $p < 0.0001$) but not for heat-treated bacteria both chronologically (past day 4 of adulthood) and at mean lifespan (FC = 1.05, $t$ test $p = 0.7078$, Fig. 4b, Supplementary Data 1).

Second, to test whether an alteration in bacterial metabolism or other factors such as total growth and growth rate underlie the effects of bacterial physiology on worm stiffness with age, we

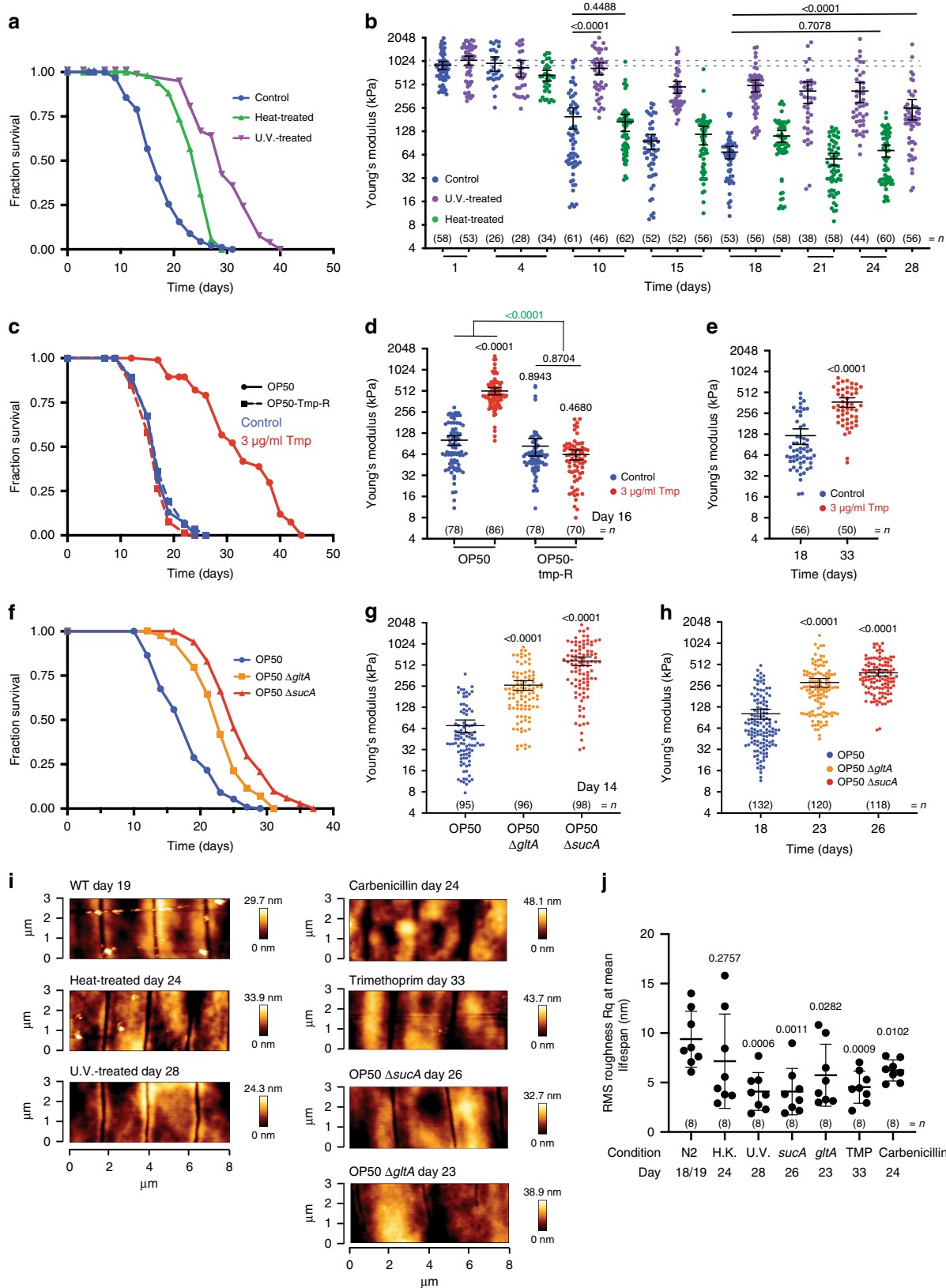

employed a nutritional and a pharmacological approach. Given that nutrition is a major regulator of metabolism in all organisms, we tested if changing the nutritional source affects the bacterial metabolome, total growth and growth rate, and worm stiffness with age. Indeed, we observe that media type affected bacterial total growth but not growth rate (Supplementary Fig. 5a, b), and

regulated worm stiffness to various degrees (Supplementary Fig. 6a, Supplementary Data 1). In particular, distinct protein sources such as Bactopeptone—an enzymatic digest of animal protein and Bactotryptone—a pancreatic digest of casein, did not significantly alter both total growth and growth rate (Supplementary Table 1, Supplementary Fig. 5a, b), but did alter the

**Fig. 4 Bacterial physiology impact stiffness and cuticle senescence with age. a** Lifespan curve of wild-type (WT) *C. elegans* on untreated control bacteria (blue), heat-treated (green) or UV-treated bacteria (purple) (*n* = 90, 85 and 106, respectively; log rank test *p* < 0.001 vs untreated control bacteria) and **b** longitudinal study of mechanical properties as Young's Modulus (YM; kPa) until mean lifespan (D18 for WT, D24 for heat-treated, D28 for UV-treated). Error bars indicate 95% CI, dotted lines mark mean YM at day 1 for WT (blue), heat-treated (green) or UV-treated bacteria (purple). Two-tailed unpaired *t* test for statistical comparison of WT to treatments at chronological age day 10 and mean lifespan. **c** Lifespan curve of untreated (0 µg/ml) (blue) and trimethoprim (Tmp)-treated (3 µg/ml) (red) WT *C. elegans* grown on sensitive OP50 (line) or grown on trimethoprim-resistant OP50-TmpR overexpressing a dihydrofolate reductase cassette (dotted line) (*n* = 83, 70, 100 and 91, respectively; log rank test *p* < 0.001, *p* = 0.4094 and *p* = 0.0206 vs OP50 0 µg/ml) and **d** mechanical properties as YM (kPa) at chronological age of 16 days. Error bars indicate 95% CI. Two-way ANOVA Tukey's multiple comparison test for statistical comparison and interaction of terms (green). **e** Mechanical properties as YM (kPa) at mean lifespan (D18 for WT, D33 for Tmp-treated). Error bars indicate 95% CI. Two-tailed unpaired *t* test for statistical comparison. **f** Lifespan curve of WT *C. elegans* grown on OP50 (blue) or OP50 mutants of the TCA cycle genes *gltA* (yellow) or *sucA* (red) (*n* = 111, 113 and 101, respectively; log rank test *p* < 0.001 vs OP50), and **g** mechanical properties as YM (kPa) at chronological age of 14 days or **h** at mean lifespan (D18 for OP50, D23 for OP50Δ*gltA* and D26 for OP50Δ*sucA*). Error bars indicate 95% CI. Two-tailed unpaired *t* test for statistical comparison of OP50 to bacterial mutants. **i** Representative AFM cuticle topography images of WT *C. elegans* at mean lifespan grown on untreated control, heat-treated, UV-treated, carbenicilin-treated (50 µg/ml) and trimethoprim-treated (3 µg/ml) OP50 bacteria or on OP50Δ*gltA* and OP50Δ*sucA* bacterial mutants. **j** Roughness quantification of topographical images presented as RMS roughness Rq ± standard deviation. Two-tailed unpaired *t* test for statistical comparison of WT with conditions. *n* represented above the graph, show number of biologically independent worm samples. For lifespan measurements, *n* represents the number of worms scored as dead. For a summary of YM values and additional statistics for independent trials, see Supplementary Data 1 and for a summary of worm lifespan trials and statistical comparison between genotypes see Supplementary Data 2. Source data are provided as a Source Data file.

metabolome of *E. coli* OP50 (Supplementary Fig. 5c, d, Supplementary Data 3). Strikingly, a 3.62-fold improved worm stiffness observed at day 12 and mediated by growth on Bactotryptone (*t* test *p* < 0.0001) was no longer observed if bacteria were heat-treated (FC = 0.54, ANOVA *p* = 0.5594, Supplementary Fig. 6b, Supplementary Data 1) or UV-treated (FC = 1.19, *t* test *p* = 0.249, post hoc power = 0.213, Supplementary Fig. 6c, Supplementary Data 1). Next, we investigated whether inhibition of proliferation parameters benefits host stiffness with age by using two antibiotics with distinct modes of action. The bacteriostatic antibiotic trimethoprim leads to metabolically impaired bacteria with decreased total growth and growth rate by inhibiting 1-carbon cellular metabolism, with accumulation of dihydrofolate[45]. In contrast, the bactericidal antibiotic carbenicillin leads to full inhibition of bacterial growth and growth rate by inhibiting cell wall synthesis[46]. Previous reports have shown that genetic or pharmacological impairment of bacterial folate metabolism increases worm lifespan independently of its effects on bacterial proliferation parameters[34,47–49]. We first confirmed that trimethoprim treatment at a concentration below the minimum inhibitory concentration impaired total bacterial growth (Supplementary Fig. 5a, b) and growth rate (Supplementary Table 1), and induced wide changes in the metabolome (Supplementary Fig. 5c, d, Supplementary Data 3) by specifically impairing folate metabolism in our *E. coli* strain OP50 (Supplementary Fig. 5f, g). Trimethoprim treatment of OP50 or OP50-TMPR overexpressing a trimethoprim-resistant dihydrofolate reductase cassette showed that trimethoprim impairs bacterial growth by specifically targeting dihydrofolate reductase (Supplementary Fig. 5a, b). Our mass spectrometry measurements of folate intermediates further support this finding as we observed a marked increase in dihydrofolate levels and a concomitant reduction of tetrahydrofolate and other folate forms in trimethoprim-treated OP50, but not in untreated or treated resistant OP50-TMPR (Supplementary Fig. 5f). Trimethoprim treatment also affected the polyglutamylation profiles of all folate forms in OP50, which is consistent with previously reported effects of trimethoprim treatment in other *E. coli* strains (Supplementary Fig. 5g)[45,50]. Finally, trimethoprim treatment of *E. coli* OP50 leads to broad changes in the metabolome (Supplementary Fig. 5c, d, Supplementary Data 3), to the same degree as observed by UV treatment (Supplementary Fig. 5d). In parallel with the effects of trimethoprim on bacterial proliferation parameters and metabolism, this treatment led to a robust

increase in lifespan of worms fed with OP50 but not OP50-TMPR (86.36% mean lifespan extension compared with OP50 non-treated *E. coli*, log rank *p* < 0.001 and −3.91% mean lifespan reduction compared with OP50-TMPR non-treated, log rank *p* = 0.0850, respectively; CPH < 0.001 for the interaction of terms: *E. coli* type and trimethoprim Fig. 4c, Supplementary Data 2). Importantly, trimethoprim treatment significantly improved stiffness of worms both chronologically by 5.01-fold at day 16 (ANOVA *p* < 0.0001, Fig. 4d, Supplementary Data 1) and by 3.04-fold at mean lifespan (*t* test *p* < 0.0001, Fig. 4e, Supplementary Data 1). These effects were bacteria-dependent as OP50-TMPR fed worms showed no improvement upon trimethoprim treatment (FC = 0.76, ANOVA *p* = 0.8704) and a significant interaction of terms for trimethoprim and bacterial strain was observed with a large effect size of 5.25-fold (*p* < 0.0001, Fig. 4d, Supplementary Data 1). On the other hand, treatment of *E. coli* with carbenicillin, which fully inhibits bacterial total growth and growth rate (Supplementary Fig. 5a, b, Supplementary Table 1) without significantly altering the bacterial metabolome (Supplementary Fig. 5c, d, Supplementary Data 3), extended worm lifespan (36.16% mean lifespan extension compared with control non-treated *E. coli*, log rank *p* < 0.001, Supplementary Fig. 6d, Supplementary Data 2) but did not significantly improve the stiffness of aged worms compared with worms fed with non-treated bacteria, both chronologically or at mean lifespan (day 4 FC = 0.97, *t* test *p* = 0.6903; day 15 FC = 1.36, *t* test *p* = 0.0897, post hoc power = 0.388; at mean lifespan FC = 1.09, *t* test *p* = 0.6152, Supplementary Fig. 6e, Supplementary Data 1). Post hoc analysis of the data from day 15 suggested this experiment was powered to detect a statistically significant difference between treatment and control with changes larger than 1.6-fold and that the true effect of carbenicillin-treatment on worm stiffness at day 15 is likely under 1.6-fold.

Finally, we tested the role of bacterial metabolism in regulating our stiffness phenotype using a genetic approach. Inhibition of bacterial respiration through genetic impairment of the respiratory chain has been previously shown to increase worm lifespan[51,52]. Here, we show that mutations in the tricarboxylic acid (TCA) cycle genes *gltA* and *sucA*, required for aerobic respiration[53], impaired bacterial total growth (Supplementary Fig. 5a, b) and bacterial metabolism (Supplementary Fig. 5c, d, Supplementary Data 3), but not growth rate (Supplementary Table 1), and robustly increased worm lifespan (28.78% and 44.83% mean lifespan extension compared with control OP50, log

rank $p < 0.001$, Fig. 4f, Supplementary Data 2). Moreover, growing wild-type worms on these bacterial mutants dramatically improved their stiffness with age, both chronologically at day 14 (OP50$\Delta$gltA vs OP50 FC = 3.77, $t$ test $p < 0.0001$; OP50$\Delta$sucA vs OP50 FC = 8.31, $t$ test $p < 0.0001$) and at mean lifespan (OP50$\Delta$gltA vs OP50 FC = 2.78, $t$ test $p < 0.0001$; OP50$\Delta$sucA vs OP50 FC = 3.81, $t$ test $p < 0.0001$, Fig. 4g, h, Supplementary Data 1).

Further, we investigated whether bacterial physiology also impacts cuticle senescence by comparing signs of cuticle decay at mean lifespan age (Fig. 4i, j). Here, we found that conditions that improved stiffness with age also improved cuticle homoeostasis (Fig. 4i, j). Altogether, our findings do not support the hypothesis that inhibiting bacterial proliferation per se is a driving factor for improved stiffness and cuticle senescence in aged worms. Alterations to other aspects of bacterial physiology, potentially including metabolism, appear to drive these changes in worm stiffness and cuticle maintenance.

**Signalling sensors regulate bacterial effects on stiffness.** We next set out to investigate the role of microbial diets in worm lifespan and our health measures, and of host genes that regulate these phenotypes. Growing worms on bacteria other than the usual *E. coli* OP50 leads to differences in worm lifespan[54], but their effects on healthspan have not been tested. We therefore chose to investigate the roles of the two bacterial strains *Comamonas aquatica* and *Bacillus subtilis* as models for altered bacterial diets.

We found that growing worms on *B. subtilis* increased lifespan (19.81%, log rank $p < 0.001$, Fig. 5a, Supplementary Data 2), whereas worms fed on *C. aquatica* were short-lived when compared with worms fed on *E. coli* OP50 (−7.47%, log rank $p < 0.001$, Fig. 5a, Supplementary Data 2), as previously reported[54–56]. Interestingly, longitudinal stiffness measurements showed that *E. coli* diets, though being standard laboratory food, are surprisingly detrimental to the maintenance of stiffness with age when compared with worms on either *B. subtilis* (*B. subtilis* vs OP50 at day 12 FC = 5.79, $t$ test $p < 0.0001$; *B. subtilis* vs OP50 at mean lifespan FC = 4.10, $t$ test $p < 0.0001$) or *C. aquatica* diets (*C. aquatica* vs OP50 at day 12 FC = 4.14, $t$ test $p < 0.0001$; *C. aquatica* vs OP50 at mean lifespan FC = 4.32, $t$ test $p < 0.0001$; Fig. 5b, Supplementary Data 1). Finally, to investigate the role of bacterial diets on cuticle senescence with ageing, we acquired topography images of worms fed with these two bacteria and control *E. coli* (Fig. 5c). Our images taken at mean lifespan (Fig. 5c), but not at day 1 (Supplementary Fig. 7a), showed clear signs of disintegration of the cuticle for worms fed on *B. subtilis* but not *C. aquatica* (Fig. 5c, d). Collectively our data imply that the effects of bacterial diets on lifespan do not correlate with our health biomarkers of stiffness or cuticle senescence with age. In fact, feeding *B. subtilis*, while benefiting some age-related physiological outputs such as lifespan and stiffness, may be detrimental to others such as cuticle ageing.

Next, we sought to investigate if transcription factors known to regulate the effects of diet on lifespan also regulate the effects on worm stiffness we observed with *B. subtilis* or *C. aquatica* bacterial diets. First, we investigated the role of the transcription factor DAF-16/FOXO, which regulates improved lifespan and stiffness with ageing induced by reduced IIS (Fig. 2a, c). As the Cox proportional hazards statistical model cannot resolve and compare survival curves that cross (Supplementary Fig. 7b), close inspection of the longevity curves and the mean lifespan values for both the wild-type and mutants in these dietary conditions suggest an interaction between *daf-16* and both dietary interventions in mediating longevity effects (Fig. 5a, e, Supplementary

Fig. 7b, Supplementary Data 2). Similarly, we show that the observed improvements in stiffness of worms with age through feeding on *B. subtilis* (FC = 6.42, ANOVA $p < 0.0001$, Fig. 5f, Supplementary Data 1) and *C. aquatica* (FC = 6.39, ANOVA $p < 0.0001$, Fig. 5f, Supplementary Data 1) diets were both modulated by *daf-16*. Effect sizes for the interaction of terms *daf-16* and bacterial diet were 2.33-fold for *B. subtilis*-*E. coli* ($p = 0.0167$) and 3.68-fold for *C. aquatica*-*E. coli* ($p < 0.0001$, Fig. 5f, Supplementary Data 1), respectively.

Another transcription factor that has a key role in sensing nutritional and environmental factors, and that is involved in the metabolic regulation of lipids, carbohydrates and amino acids is PPAR/NHR-49[57–59]. Deletion of *nhr-49(nr2014)* shortens worm lifespan[58] when fed *E. coli* OP50 (−41.48%, log rank $p < 0.0001$, Supplementary Data 2). Importantly, we show that *nhr-49* regulated the effects of nutrition on lifespan as *nhr-49* mutant worms fed *C. aquatica* were longer lived than on *E. coli*, and worms fed *B. subtilis* were even further long-lived when compared with wild-type worms on these two diets (CPH = 0.0052 for the interaction of terms: bacterial diet: *C. aquatica* vs *E. coli* and genotype: wild-type vs *nhr-49*; CPH < 0.0001 for the interaction of terms: Bacterial diet: *B. subtilis* vs *E. coli* and Genotype: wild-type vs *nhr-49*; Fig. 5a, g, Supplementary Fig. 7c, Supplementary Data 2). In contrast, stiffness measurements showed that effects of *C. aquatica* (FC = 8.14, ANOVA $p < 0.0001$, Fig. 5h, Supplementary Data 1) and *B. subtilis* (FC = 8.51, ANOVA $p < 0.0001$, Fig. 5h, Supplementary Data 1) diets on worm stiffness with age were positively modulated by *nhr-49*. Effect sizes for the interaction of terms *nhr-49* and bacterial diet were 6.39-fold for *C. aquatica*-*E. coli* (ANOVA $p < 0.0001$) and 3.63-fold for *B. subtilis*-*E. coli* (ANOVA $p = 0.0006$, Fig. 5h, Supplementary Fig. 7d, Supplementary Data 1). Overall, our AFM method captures the complex bacteria diet-evoked effects on host health parameters such as stiffness and cuticle roughness and the associated host transcription factors modulating these effects.

## Discussion

The quantitative measure of the mechanical properties of cells and/or tissues has emerged as a tool to identify, characterise and classify disease states, but not yet to define health[16,17]. As in mammalian cells and tissues, *C. elegans* physiology is also regulated and modulated by mechanical forces[60–62] but how these change during ageing, and by intrinsic (e.g. genetics) or extrinsic factors (e.g. bacteria) that control organismal lifespan has not been assessed until now.

Multiple technologies have been developed to measure the mechanical properties of tissues including piezos[63], force probes[64], micropipette aspiration[65], optical tweezers[66] and AFM[18] with various force resolutions. AFM utilises a fine probe that can be as soft as 0.01 N/m to determine the stiffness of delicate biological material. With a tip diameter of 2 nm touching the sample surface, AFM produces higher imaging resolutions than those achieved by using a traditional optical microscope and comparable to those obtained using scanning electron microscopy[18]. These AFM features enabled us to detect finer differences in stiffness and topography during ageing that were not captured previously by hydrostatic pressure measurements of bulk mechanical properties of entire worms[67]. Our measurements show that worms undergo a marked loss of stiffness and increased cuticle senescence during ageing. These stiffness measurements provide a quantitative approach for measuring a common observable feature when handling aged worms—increased softness possibly resulting from the loss in the hydrostatic body tension and increased shrinkage. In line with our observations, previous findings show that human fibroblasts from older donors

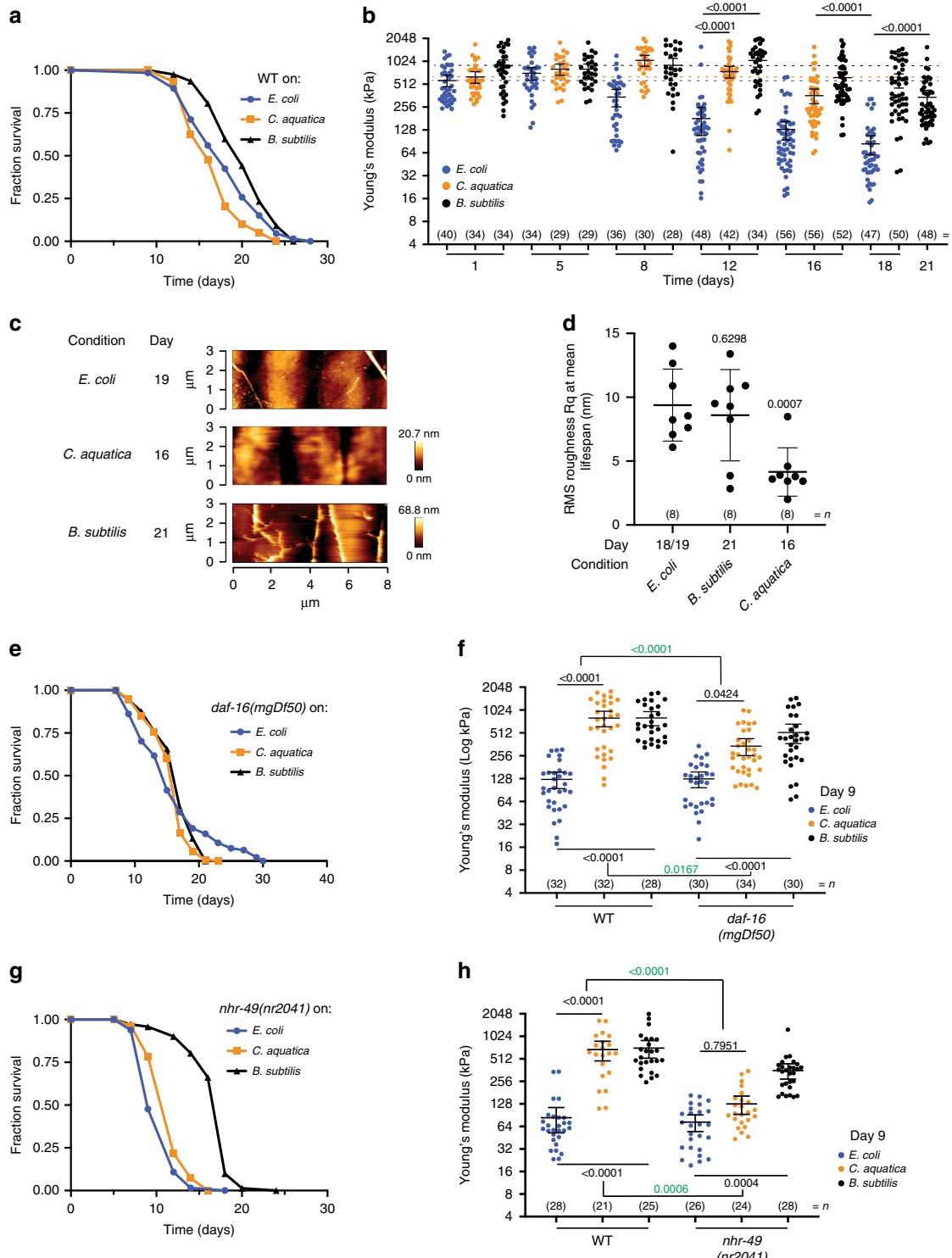

(> 30 years) are less stiff than those obtained from younger donors (< 25 years)[68], and measurements of human dermal tissue obtained from donors at different ages show an upward trend in stiffness in the age groups from 25 to 55 and a drop in the age group from 56 to 65 years[69], suggesting loss of stiffness as a potential evolutionarily conserved process[17].

The skin, the largest and most visible organ in the human body, is arguably the most used phenotypic classifier of human ageing. Like in humans, age-dependent deterioration of the outermost tissue is a prominent feature of ageing in worms[70,71] and remarkably, collagen remodelling can modulate the ageing

process[13]. AFM allowed us to acquire changes in cuticle topography during ageing at high-resolution on unfixed and untreated tissue resulting in measurements of direct, physiologically relevant properties. Our AFM data are consistent with observations for worms of various species showing a progressive decay in cuticle structure and permeability with ageing[72]. We show that the mechanisms leading to the senescent decay of the cuticle and the loss of stiffness with ageing correlate with each other at mean lifespan (Supplementary Fig. 8a), but neither correlates with the effects of correspondent genetic, pharmacological and dietary interventions on mean lifespan (Supplementary Fig. 8b, c),

**Fig. 5 Microbial diet effects on stiffness and cuticle senescence with age are regulated by host nutrient sensors. a** Lifespan curve of wild-type (WT) *C. elegans* grown on control *E. coli* OP50 (blue), *C. aquatica* DA2123 (yellow) and *B. subtilis* PY79 (black) bacteria (*n* = 256, 250 and 292, respectively; log rank test *p* < 0.001 vs OP50) and **b** longitudinal study of mechanical properties as Young's Modulus (YM; kPa) until mean lifespan (D18 for OP50, D16 for *C. aquatica* and D21 for *B. subtilis*). Error bars indicate 95% CI, dotted lines mark mean YM at day 1 for OP50 (blue), *C. aquatica* (yellow) and *B. subtilis* (black). Two-tailed unpaired *t* test for statistical comparison of OP50 with other bacteria at chronological age day 12 and at mean lifespan. **c** Representative AFM cuticle topography images of WT *C. elegans* grown on *E. coli*, *C. aquatica* or *B. subtilis* at mean lifespan and **d** roughness quantification of topographical images presented as RMS roughness Rq ± standard deviation. Two-tailed unpaired *t* test for statistical comparison of OP50 to other bacteria. **e** Lifespan curve of *daf-16* mutant *C. elegans* grown on *E. coli* (blue), *C. aquatica* (yellow) or *B. subtilis* (black) (*n* = 94, 120 and 128, respectively; log rank test *p* = 0.9265 and *p* = 0.3676 vs *daf-16* on *E. coli*). **f** Mechanical properties as YM (kPa) of WT or *daf-16* mutant *C. elegans* grown on control *E. coli* (blue), *C. aquatica* (yellow) or *B. subtilis* (black) at chronological age of 9 days. Error bars indicate 95% CI. Two-way ANOVA Tukey's multiple comparison test for statistical comparison and interaction of terms (green). **g** Lifespan curve of *nhr-49* mutant *C. elegans* grown on *E. coli* (blue), *C. aquatica* (yellow) or *B. subtilis* (black) (*n* = 65, 55 and 71, respectively; log rank test *p* < 0.0001 vs *nhr-49* on *E. coli*). **h** Mechanical properties as YM (kPa) of WT or *nhr-49* mutant *C. elegans* grown on control *E. coli* (blue), *C. aquatica* (yellow) or *B. subtilis* (black) at chronological age of 9 days. Error bars indicate 95% CI. Two-way ANOVA Tukey's multiple comparison test for statistical comparison and interaction of terms (green). *n* represented above the graph, show number of biologically independent worm samples. For lifespan measurements, *n* represents the number of worms scored as dead. For a summary of YM values and additional statistics for independent trials, see Supplementary Data 1 and for a summary of worm lifespan trials and statistical comparison between genotypes see Supplementary Data 2. Source data are provided as a Source Data file.

suggesting that our quantifiable health parameters can be dissociated from lifespan.

Previous reports have shown that maintenance of hydrostatic pressure in worms is controlled by tissues like the gonad and the intestine[67]. In worms, a major ageing pathology is the atrophy of the intestine[12], resulting from a gut-to-yolk mass bioconversion driven by the IIS pathway as a consequence of a run-on reproductive programme[11]. Consistent with this view, we find that germline-less *glp-1*-mutant animals, or those with reduced IIS signalling by *daf-2* mutations exhibit improved body stiffness with ageing in a *daf-16*-dependent manner. Our data also implicate diet (i.e., distinct bacterial diets or DR regimens) and bacterial physiology as factors regulating stiffness during ageing. This is consistent with previous findings showing that worms fed *B. subtilis* have reduced gut atrophy with age[11], and that stochastic or genetic factors also influence tissue-specific decline in ageing animals[19]. Moreover, Zhang et al.[10] have recently shown that even within wild-type isogenic populations there is great variability of health phenotypes with ageing. Similarly, we observe an increasing variance in stiffness values from aged isogenic wild-type worms compared with young worms (Supplementary Fig. 8d). Interestingly, none of the tested interventions reduced this larger variability in stiffness observed within ageing worm cohorts at mean lifespan, with the potential exception of trimethoprim (Supplementary Fig. 8d).

Whether extension of lifespan leads to a concomitant extension of healthspan is a widely debated issue. Our data support the view that lifespan extending interventions do not always translate into healthspan gains as measured by reduced cuticle senescence (Supplementary Fig. 8b) or increased stiffness (Supplementary Fig. 8c). However, unlike previous studies[2], we show that reduced IIS significantly improves our health measures, both chronologically and at mean lifespan. This suggests that in addition to maximum velocity or worm movement, which was previously shown to be a valid biomarker to predict healthspan[9], our two additional health biomarkers provide functional parameters that capture improved health outputs mediated by reduced IIS. In line with our findings, recent evidence shows that reduced IIS in late life in mice increases healthspan and lifespan[73].

Utilising four distinct DR regimens, we show that despite similar effects on extending organismal lifespan, the effects on cuticle and stiffness maintenance with age were remarkably distinct. In line with previous reports by Bansal et al.[2] exploiting widely used health biomarkers, we show here using AFM that a mutation in the *eat-2* gene did not benefit our health measures of cuticle senescence or dramatically improve stiffness with age. On the contrary, supplementation with the DR mimetic

metformin not only extended lifespan but also improved our biomarkers both chronologically and at mean lifespan. Using a *Pacs-2*::GFP transcriptional reporter line that captures an effective degree of DR on host physiology[74], we show that *acs-2* expression is significantly increased in all our tested DR regimes: the *eat-2* and *phm-2* mutants, BD and metformin conditions (Supplementary Fig. 8e). Although expression of *acs-2* correlates with longevity (Supplementary Fig. 8f) in the interventions tested in this study, it did not correlate with either stiffness (Supplementary Fig. 8g) or cuticle roughness (Supplementary Fig. 8h). Generally, this suggests that the DR-like physiological response measured by *Pacs-2*::GFP and induced in worms by genetic mutations, metformin and interventions that alter bacterial physiology, may not be the driving mechanism regulating these two health biomarkers. Notably, our data highlight that interventions aimed at improving specific health phenotypes may not positively impact others, and argue for a battery of healthspan parameters together with lifespan measurements for each genetic, dietary and pharmacological intervention in order to accurately assess general improvements in healthspan.

The ability of an organism to adapt, survive and persist in its environment intimately and ultimately depends on the capacity of its cells and tissues to withstand mechanical pressure. Hence, the status of the hydrostatic skeleton could act as an indicator of physical age and a parameter to define the health state of animals. Development of high-throughput approaches using AFM could help to quantitatively measure these health biomarkers at an unprecedented scale, and to reveal not only genetic drivers regulating health, but also pharmacological and nutritional strategies to improve it.

## Methods
**Strains**. *C. elegans* maintenance was performed using protocols previously established[75]. Strains were grown and maintained at 20 °C on the bacterial diets indicated on each figure.

*C. elegans* strains: N2 CGCH, GR1307 *daf-16 (mgDF50)*, EFS7 *daf-16 (mgDF50)*; *daf-2 (e1370)*, CB1370 *daf-2(e1370)*, CB4037 *glp-1(e2141)*, AU147 *daf-16(mgDf50)*; *glp-1(e2141)*, DA597 *phm-2(ad597)*, DA1116 *eat-2(ad1116)*, STE68 *nhr-49 (nr2041)*, WBM392 *wbmIs33[Pacs-2*::GFP+*rol-6(su1006)]*, FGC26 *eat-2 (ad1116)*; *wbmIs33[Pacs-2*::GFP+*rol-6(su1006)]*, FGC28 *phm-2(ad597)*; *wbmIs33 [Pacs-2*::GFP+*rol-6(su1006)]*.

Bacterial strains: *E. coli* OP50, OP50 Δ*sucA::kan*, OP50 Δ*gltA::kan*, OP50-MR metformin-resistant[34], OP50-TMPR[34] trimethoprim-resistant overexpressing a dihydrofolate reductase cassette, OP50 (pSF-RecA-daGFP), *Comamonas aquatica* DA1877, *Bacillus subtilis* PY79.

**Nematode culture conditions**. Standard nematode growth media was used to grow and maintain worms. Where indicated molten agar was supplemented with carbenicillin (50 μg/ml), metformin (50 mM), trimethoprim (TMP; 3 μg/ml).

Seeding of drug and control plates was performed using 100 μl of an overnight culture of OP50, OP50-MR or OP50-TMPR grown on LB. Lawns were left to grow for 96 h at 20 °C before being used experimentally. For UV treatment, OP50 bacteria from an overnight culture were mixed in a proportion of 1:1 with fresh LB and irradiated for 30 min on a CL-1000 Ultraviolet Crosslinker (UVP) containing bulbs irradiating at 254 nm. For heat treatment, OP50 bacteria from an overnight culture were incubated for 30 min at 70 °C in a waterbath with regular shaking at 10 min intervals. For UV and heat treatment of bacteria, the resulting UV and heat-treated bacteria were centrifuged at 16.1 g for 20 min at 4 °C, then resuspended in phosphate-buffered saline at a final OD 600 nm = 24, and 100 μl was seeded onto nematode growth media (NGM) plates. Bacterial deprivation was performed with NGM plates in the absence of E. coli, as previously described[76]. All chemicals were purchased from Sigma Aldrich.

**Lifespan analysis**. Lifespan measurements were performed as follows. Axenic worm eggs were obtained using alkaline hypochlorite treatment of gravid adult hermaphrodites that had been kept in optimal temperature and feeding conditions for at least three generations. These were then placed onto plates containing the test bacterial strain and maintained at 20 °C. Lifespan measurements were initiated by transfer of L4-stage worms (day 0) to plates containing bacteria grown in the presence or absence of treatment for 96 h. Worms were transferred to fresh plates every day during the reproductive period and thereafter, every other day until day 12. For the temperature shifting experiment, worms grown at 20 °C until the L4 stage were transferred to plates and placed experimentally at 15 °C, 20 °C and 25 °C for the rest of their lifespan measurement. For the glp-1 mutant experiments, N2 and respective glp-1, glp-1,daf-16 and daf-16 mutants worms were raised from L1 to L4 at 25 °C and shifted to 20 °C at the L4 developmental stage. For the daf-2 mutant experiments, all animals were grown and maintained at 20 °C.

For the metformin and trimethoprim experiments, worms were placed on drug treatment plates at the L4 stage and transferred every 4 days. To prevent progeny development in these conditions, plates were supplemented by adding topically 100 μl of 5-fluoro-2-deoxyuridine-FUdR (15 μM) on the day prior to use. For the metformin shifting experiment, worms grown on control bacteria were transferred at day 6 of adulthood to plates containing metformin.

For bacterial deprivation and UV-treated bacteria lifespans, bleached eggs were placed onto UV-irradiated bacterial plates until day 1 of adulthood. This allows worm development and prevents bacterial contamination. At day 1 of adulthood, worms were transferred to each of the treatment/condition plates containing FUdR (50 μM) and transferred every 4 days to new fresh plates until day 12. Worms that showed severe vulva protrusion or bagging were censored. Survival was monitored at regular time points and worms scored as dead if they did not show any movement when prodded with a platinum wire. Each experimental condition was tested with a total of at least 2–3 independent trials of ~ 60–100 animals in total (Supplementary Data 2).

**Worm preparation for AFM**. Worms for AFM analysis were taken from cohorts running in parallel with the control condition (wild type on OP50) and in parallel to measurements for lifespan. Two to four independent trials were performed per condition (Supplementary Data 1). Each condition per trial contained the measurement of at least eight individual animals (sampled randomly). Worms were prepared as follows. In brief, a cohort of worms was paralysed for 1 h in a droplet of 15 mg/ml BDM. Each worm was then transferred to a 1 mm thick 4% agarose bed in a 30 mm petri dish, adhered to the pad using tissue glue (DERMABOND) applied with a fine, blunt glass needle, and immediately covered with M9 buffer to prevent drying of the worm during AFM measurements. Force measurements were taken from the neck and hip region of the worm[77] (location 1 or location 3; Supplementary Fig. 1a), avoiding the mid body or vulva region. Topographical images were taken avoiding the alea (Supplementary Fig. 1g). Preparation of the worms and AFM measurements were all performed at 20 °C.

**Atomic force microscopy**. AFM uses force-indentation curves to determine local mechanical properties of samples as well as their 3D topography under physiological conditions in a nano-scale range. The technique is based on a laser beam deflection system where a laser is reflected from the back of a cantilever and onto a position-sensitive detector (Fig. 1a). The laser deflection serves as a feedback loop to control tip position and force applied to the sample during the scan. We used the AFM NanoWizard3 (JPK) to acquire individual force-indentation curves in force spectroscopy mode (set force 450 nN, 0.5 μm/s indentation speed), and topographical images of the worm cuticle in imaging mode (256 px, scanning speed 0.7 Hz, deflection at 0.2–0.4 V). The force curves were acquired using a 10-μm glass bead as tip attached to a tipless cantilever of $k$ = 5.79–10.81 N/m stiffness (NSC12 7.5 N/m μMasch produced by sQUBE www.sQUBE.de) to prevent the cuticle from being pierced at larger indentations. The topographical images were acquired using very soft cantilevers designed for imaging of biological samples (qp-CONT-10 $k$ = 0.01 N/m; nanosensors). Cantilever sensitivity and spring constant were calibrated using the JPK calibration tool (thermal noise method—http://www.nanophys.kth.se/nanophys/facilities/nfl/afm/jpk/manuf-manuals/handbook-2.2a.pdf) prior to each experiment.

**AFM data analysis**. Raw AFM data were analysed using JPK analysis software. All individual force curves were processed to zero the baseline, to determine tip-sample contact point and to subtract cantilever bending. To calculate the Young's Modulus, each individual force-indentation curve was further analysed by fitting the Hertz/Sneddon model for contact mechanics to the entire curve by using the JPK software and by taking the indenter shape (10-μm bead) into account (see Supplementary Fig. 1b). Young's Modulus values with 95% CI are displayed using the GraphPad Prism 8 software.

All topographical images are flattened using the plane fitting option of the JPK software at 2 degree to correct for sample tilt and natural curvature of the worm. The roughness of the cuticle was determined using the histogram function of the software by averaging the roughness (RMS roughness Rq) of four independent 1 μm$^2$ areas within the annuli region of a topographical image (Supplementary Fig. 1g). At least eight independent images (worms) were analysed per condition, with the exception of WT day 8 ($n$ = 5) and BD ($n$ = 4) at mean lifespan.

**Fecundity and size measurements**. Synchronised L1 larvae were obtained by bleaching of gravid adults left overnight in M9 solution. Larvae of each genotype were raised on OP50 until reaching the L4 stage. Worm size was measured at day 1 of adulthood using a Zeiss Axioplan V16 dissecting microscope and captured using the ZEN pro software (Carl Zeiss). For brood size and fecundity timing assays, starting with L4 worms, animals were transferred daily to fresh new plates during the reproductive period and progeny numbers counted. All data show the average of at least three independent trials.

**Bacterial mutant construction**. OP50 bacterial strains generated in this study were obtained by transduction of mutations from the respective E. coli single deletion mutants obtained from the Keio collection[78] by transfer of kanamycin-resistant-tagged mutations via P1vir phage-mediated transduction. All bacterial mutants were confirmed by colony PCR using the following primers: in general, binding sites of the -cseq-F and -R primers are located up and downstream of the mutation site, respectively, and were used to confirm kanamycin-sensitive mutants, whereas the K1, which binds to the kanamycin resistance gene, was used in conjunction with the appropriate -cseq-F primer to confirm kanamycin-resistant mutants. K1 - kanamycin$^R$: CAGTCATAGCCGAATAGCCT, sucA-cseq-F: GCCCAGAACAGCGCTTTGAAAGCC sucA-cseq-R: GGAACGTCGAACGCCA GTTAG, gltA –cseq-F: GGATCCTTTACCTGCAAGCG, gltA –cseq-R: GGGGG GTATAGATAGACGTCA.

**Bacterial colonisation measurements**. Worms were grown on seeded NGM plates with OP50-C bacteria containing a plasmid that confers resistance to Kanamycin at 20 °C (OP50 (pSF-RecA-daGFP)). Worms at day 1, 5 and 8 of adulthood were anesthetised by adding topically 1% levamisole to prevent the release of the gut content. Ten worms were picked into 20 μl of M9 with 0.01% Triton X-100. Worms were grinded using a micro-pestle and 5 μl of the solution was transferred into 96-well plate containing 200 μl of LB with 50 μg/ml kanamycin. Bacterial growth was performed at 37 °C and measured every 5 min for an 18 h incubation period with regular shaking using Tecan Infinite M200 PRO microplate reader and Magellan V6.5 software.

Visualisation of wild-type N2 and phm-2 worm colonisation was captured at day 1 and 5 of adulthood. Images were acquired using a Leica DMRXA2 microscope and Orca digital camera (Hamamatsu) and captured using Volocity 6.3 Software. Green fluorescence was observed through a GFP filter cube ($\lambda_{ex}/\lambda_{em}$ 450–490 nm/500–550 nm). All data present the average of at least three independent trials.

**Bacterial growth assays**. Bacterial growth assays were performed in transparent, flat-bottomed 96-well plates. Unless otherwise stated, plates were prepared by loading each well with 200 μl of liquid NGM solution containing an overnight bacterial culture grown on LB and diluted 100-fold and with drug supplements at the desired concentration as required (50 mM Metformin, 50 μg/ml Carbenicillin, 3 μg/ml Trimethoprim). If metformin or other supplements were added, an equivalent volume of water was added to negative control wells. To investigate the effect of different types of media on bacterial growth, modified liquid NGM containing bactopeptone (2 g/l) was replaced with either soy peptone (2 g/l), or bactotryptone (2 g/l). For bacterial growth assays, measurements and shaking (50 s) were performed every 10 min over an 18-h period. The absorbance of each well was measured at OD 600 nm using a Tecan Infinite M2000 microplate reader operated via Magellan V6.5 software (Tecan). At least three independent trials were carried out per experiment. Data was analysed using R (R Core Team). The total bacterial growth was estimated as the OD area under the curve (AUC) integral and was calculated with the function auc from the pROC library in R (v. 1.14.0). AUC values were $log_2$ transformed to enable relative comparisons in logFC scale to be made. Bacterial growth rates were calculated from the temporal series using an adaption of the algorithm from Hall et al.[79], implemented in the growth rates R package (v0.7.2). This method calculates all linear regressions from the log-transformed data given a window size, taking the steepest slope as the maximum growth rate. The first three points from every series were excluded from the

analysis to avoid biases produced by experimental measurement noise. The window size used was 15 time points.

**C. elegans fluorescence reporter measurements**. *Pacs-2*::GFP worms were grown accordingly with the conditions previously outlined in the lifespan analysis methods section. Worms at the L4 stage were transferred to each respective experimental condition for an additional 24 h at 20 °C. Worms were anesthetised with 1% levamisole on NGM plates and were imaged under a ×63 objective using a Zeiss Axio Zoom V16 microscope system equipped with an AxioCam MRm camera operated by Zen 2 software (Zeiss) and a GFP filterset (excitation: 450–490 nm; emission: 500–550 nm). All images were exported in TIFF or CZI format and fluorescence levels were quantified using Volocity 5.2 software (PerkinElmer) run on a Surface tablet (Microsoft). The fluorescence intensity of individual worms was calculated as the pixel density of the entire cross-sectional area of the worm from which the pixel density of the background had been subtracted divided by the area of the worm. Three independent trials were carried out with a minimum of eight worms imaged per condition per trial. Data were analysed using Graphpad Prism8.

**Bacterial 1-carbon metabolism analysis**. For bacterial sample collection and metabolite extraction, 3-day-old bacterial lawns of OP50 and OP50-TMPR grown at 20 °C on standard NGM plates were scraped using a 24 cm cell scraper and transferred to an eppendorf tube and snap frozen in liquid nitrogen. The bacterial pellet was kept at −80 °C until required.

At least four biological replicates from each type of bacterial sample were collected for each measurement. Samples were resuspended in MS sample buffer at pH 7.0 containing 20 mM ammonium acetate, 0.1% ascorbic acid, 0.1% citric acid, 100 mM dithiothreitol (all from Sigma Aldrich). Bacterial suspensions were sonicated for 10 s at 40% amplitude using a hand-held sonicator (Q700 sonicator, Qsonica). Protein was removed by acetonitrile (Sigma Aldrich) precipitation and centrifugation for 15 min at 12,000 × g, 4 °C. Supernatants were transferred to fresh tubes, lyophilised and stored at −80 °C until required. Prior to analysis, lyophilised samples were resuspended in 60 µl of MS sample buffer and centrifuged for 15 min at 12,000 × g, 4 °C. The resulting supernatants were transferred to glass sample vials.

Folate measurements were performed by LC-MS/MS. Metabolites were resolved by reversed-phase chromatography using an Acquity UPLC BEH C18 column (50 mm × 2.1 mm; 1.7 µm bead size) (Waters Corporation, UK). Solvents for UPLC were: Buffer A, 5% methanol, 95% Milli-Q water and 5 mM dimethylhexylamine at pH 8.0; Buffer B, 100% methanol. The column was equilibrated with 95% Buffer A: 5% Buffer B. The UPLC was coupled to a XEVO-TQS mass spectrometer (Waters Corporation, UK) operating in negative-ion mode using the following settings: capillary 2.5 kV, source temperature 150 °C, desolvation temperature 600 °C, cone gas flow rate 150 L/h and desolvation gas flow rate 1200 L/h. Folates were measured using multiple reaction monitoring with optimised cone voltage and collision energy for precursor and product ions. Mass spectrometric data were analysed as in[80] using MassLynx Software (Waters).

**Bacterial untargeted metabolomics**. Bacterial cultures (OP50 or OP50 *ΔgltA:: kan*) were prepared by using 500 µl of overnight bacterial culture grown on LB to inoculate 50 ml of control liquid NGM (with Bactopeptone or Bactotryptone) and liquid NGM (with Bactopeptone) supplemented with TMP (3 µg/ml). For Heat-treated and UV-irradiation conditions, OP50 bacteria were grown on LB and heat-treated or UV-irradiated as previously stated. Bacteria in all these conditions were grown at 25 °C for 24 h with constant shaking at 180 rpm. For the carbenicillin condition, OP50 bacteria were grown at 25 °C for 20 h with constant shaking at 180 rpm in control liquid NGM (with Bactopeptone) and treated with carbenicillin (50 µg/ml) for an additional 4 h. All cultures at OD 600 nm = 20–25 were then chilled on ice for 5 min before being centrifuged for 10 min at 6400 × g, 4 °C. Supernatant was removed except for 500 µl that was used to resuspend the bacterial pellet. Samples were then transferred to 5 ml tubes and were centrifuged as before. The supernatant was completely removed and tubes were flash frozen in liquid nitrogen. Samples were then stored at −80 °C until metabolite extraction. Four independent biological replicates were prepared for each condition.

Samples were shipped to Human Metabolome Technologies (HMT), Inc. on dry ice for further processing and metabolomic analysis (Yamagata, Japan). Metabolome analysis was performed in 28 samples of *E. coli* using Capillary Electrophoresis Time-of-flight Mass Spectrometry (CE-TOFMS) in two modes for cationic and anionic metabolites. We detected 228 metabolites (119 metabolites in Cation mode and 109 metabolites in Anion mode) on the basis of HMT's standard library. In HMT, the samples were mixed with 1600 µL of methanol. Then, 1000 µL of Milli-Q water containing internal standards (5 µM) were added, mixed thoroughly and centrifuged (2300 × g, 4 °C, 5 min). The supernatant (350 µL Å ~ 2) was filtrated through 5-kDa cutoff filter (ULTRAFREE-MC-PLHCC, Human Metabolome Technologies, Yamagata, Japan) to remove macromolecules. The filtrate was centrifugally concentrated and resuspended in 25 µL of ultrapure water immediately before the measurement.

Cationic metabolites were measured by CE-TOFMS in the positive electrospray ionization (ESI) mode and anionic metabolites were measured by CE-TOFMS in the negative ESI mode. Samples were diluted to improve the quality of analysis. Peaks detected in CE-TOFMS analysis were extracted using MasterHands

ver.2.17.1.11 automatic integration software (Keio University) in order to obtain peak information including m/z, migration time (MT) and peak area.

The peak area was then converted to relative peak area by the following equation (relative peak area = metabolite peak area/(internal standard peak area × sample amount). The peak detection limit was determined based on signal-noise ratio S/N = 3. Putative metabolites were then assigned from HMT's standard library and Known–Unknown peak library on the basis of m/z and MT. The tolerance was ±0.5 min in MT and ±10 ppm (mass error (ppm) = (measured value −theoretical value)/measured value × $10^6$ in m/z. If several peaks were assigned the same candidate, the candidate was given the branch number. In addition, absolute quantification was performed in target metabolites. All the metabolite concentrations were calculated by normalising the peak area of each metabolite with respect to the area of the internal standard and by using standard curves, which were obtained by single-point (100 µM or 50 µM) calibrations (Supplementary Data 3).

Concentrations were $\log_2$ transformed and a linear model was fitted to the data for multiple univariate analysis. R programming language (v. 3.5.1) was used to analyse the metabolomics data, where the tidyverse package (v. 1.2.1) was used to work with the data sets, FactoMineR package (v. 1.41) was used to analyse PCA, and ComplexHeatmap package (v. 1.20) was used to represent the complete heatmap. The heatmap represents the z score of each metabolite, and was calculated from the putative metabolite data (Supplementary Data 3). The z score represents the standard deviations that a particular observation is away from the population mean, given a specific metabolite. Metabolites for which no quantifiable measurements were obtained were represented as ND. Metabolites (rows) and samples (columns) were further clustered by euclidean distances, shown the each dendrogram. For the PCA analysis, each ND value was imputed with an epsilon of 2E-52 according to 'substitution by an epsilon value' method[81].

**Statistical analysis**. Statistical analysis of lifespan curves was performed by the log rank test and the Cox proportional hazards test using JMP 11 software (SAS Institute). Statistical analysis of Young's Modulus measurements was performed using two-tailed unpaired *t* test, two-tailed paired *t* test (for before/after BDM data set), or two-way ANOVA test correcting for multiple comparisons using the Tukey's test by GraphPad Prim 8 software as indicated in the Figure legends. Effect sizes were measured by dividing the means of the YM measurements of two conditions, and magnitude of effects in two-way ANOVA interaction analysis were obtained by subtracting the means of the effects sizes between the variables tested. Post hoc analysis was performed using the online calculator https://clincalc.com/ stats/power.aspx considering two independent study groups, continuous primary endpoints, the YM values ± standard deviations and respective number of worms and assuming a type I/II error rate Alpha of 0.05. Statistical analysis of brood size and fecundity timing was performed using Graphpad Prism 8 software applying a two-tailed unpaired *t* test and multiple *t* tests correcting for multiple comparisons using the Benjamini–Hochberg adjustment. Gut colonisation measurements were analysed using a two-tailed unpaired *t* test by GraphPad Prim 8 software. Bacterial growth measurements were statistically analysed by performing ANOVA tests to all the combinations of interest. The results were corrected for multiple comparisons with the Tukey's test. The interaction between groups was studied applying a two-way ANOVA. One-carbon metabolism comparisons were performed using two-way ANOVA multiple comparison tests by GraphPad Prism 8 software, correcting for multiple comparisons using the Tukey's test. Significant differences in metabolite levels in our untargeted metabolomics data sets were estimated in R using post hoc Tukey's multiple comparison statistical test. Benjamini–Hochberg multiple comparison adjustment was applied with an FDR threshold of < 0.05. Statistical analysis for *Pacs-2*::GFP comparisons between experimental conditions was performed using a one-way ANOVA correcting for multiple comparisons using the Dunnett's test by GraphPad Prim 8 software. Statistical analysis for the correlation between stiffness, roughness, mean lifespan and *Pacs-2*::GFP expression parameters (Supplementary Fig. 8) was performed using linear regression by GraphPad Prism 8 software.

**Reporting summary**. Further information on research design is available in the Nature Research Reporting Summary linked to this article.

## Data availability

Data that support the findings of this study are available in the Source data file and are available from the corresponding author upon reasonable request.

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

## Acknowledgements

This project was part funded by the European Research Count Advanced Grant: 247041 (MicroNanoTeleHaptics), the Engineering and Physical Sciences Research Council Grant: EP/K005030/1 (Robotic Teleoperation for Multiple Scales: Enabling Exploration, Manipulation and Assembly Tasks in New Worlds), the Wellcome Trust/Royal Society (Sir Henry Dale Fellowship-102532/Z/12/Z and 102531/Z/13/A) and MRC (MC-A654-5QC80). We thank David Gems for comments on the manuscript, and Arantza Barrios, Richard Poole and their lab members, as well as Muna Elmi and Helena Cochemé for critical discussions and encouraging support throughout the duration of this project. Special thanks go to Helena Cochemé for designing the AFM working schemes. We thank the JPK support team, especially Alex Winkler (now University of Cambridge) for his outstanding technical support around the AFM. Worm strains were provided by the CGC, which is funded by NIH Office of Research Infrastructure Programs (P40 OD010440).

## Author contributions

C.L.E. and F.C. developed the concept and design of the project. C.L.E., D.M.M., K.B.K., R.P., K.-Y.L., P.P.L. and F.C. performed research and collected data. C.L.E., D.M.M., R.P., K.-Y.L., A.B. and F.C. performed analysis. C.L.E. and F.C. wrote the manuscript. N.D.E.G., A.B., V.M.P., M.A.S. and F.C. critically revised the manuscript and secured funding.

## Competing interests

The authors declare no competing interests
