## [Peer Review File · Nature Communications]

Reviewers' comments:

Reviewer #1 (Remarks to the Author):

SUMMARY

In this work, the authors use atomic force microscopy to characterize patterns of cuticle aging in *C. elegans*. The author establish that stiffness decreases with age in wild-type animals. Stiffness appears to be related to movement during aging as animals with decreased movement tend to be less stiff. The authors use this measure of cuticle quality to study physiological function in genetic and dietary restriction (DR) paradigms that alter lifespan. Loss-of-function alleles in the insulin-signaling pathway increase maintenance of stiffness in a *daf-16* dependent manner; the authors suggest that these perturbations increase both absolute and relative duration of life spent with good cuticle quality. Surprisingly, effects of genetic models of DR on stiffness are variable; one genetic model of DR shows increased maintenance of stiffness while another model shows decreased stiffness compared to wild-type. A DR mimetic that alters bacterial metabolism, metformin, also increases maintenance of stiffness. These findings prompt the authors to investigate the effect of bacterial metabolism on cuticle quality. The authors show that some but not all perturbations to standard *E. coli* food bacteria increase maintenance of stiffness. Finally, the authors test the effects of feeding on *B. subtilis* and *C. aquatica* on stiffness.

IMPRESSION

Overall, this is an interesting characterization of a new and potentially valuable marker of the process of senescence. The authors rightly conclude that many different aspects of so-called "health" are surprisingly uncorrelated with lifespan and with one another, so broadening the repertoire of measures of "health" will be critical to better understanding aging and senescence. In addition, the authors have performed experiments to map out the effects of a large variety of interventions on cuticle stiffness and integrity, which would do a great deal to establish these as useful measures of health -- if the data appeared to be reliable. However, the data, as presented, seem to be quite irreproducible even between seemingly identical experiments in the authors' lab. Perhaps these are minor errors in data formatting or in my understanding that can be explained on revision. Otherwise, however, I'm not sure how much of what the authors claim to conclude is really supported by the data.

This manuscript is also marred by substantial inattention to detail, such as failing to specify key experimental parameters (e.g. what temperature were the worms raised at for various experiments? What days of adulthood were the measures made for each figure panel? Where anatomically were AFM force measurements made?), lack of analytic/statistical details (e.g. how were force-indentation curves calculated? what do the error bars represent?), and carelessness in nomenclature (e.g. *daf-2(e1370)* is not a null allele, and should not be referred to as a "knockout" or as "*daf-2(0)*").

The authors also make a series of over-broad claims about "bacterial metabolism" being a critical variable with regard to changes in cuticle stiffness under different feeding regimes, but (as described below), these claims dramatically outpace the actual evidence that the authors provide.

MAJOR CONCERNS

(1) The authors' own data make the AFM force-indentation measurements appear to be highly irreproducible, either quantitatively or qualitatively. The following is a non-exhaustive list of places where the authors' data appear to be internally inconsistent:

- Fig 1E shows WT worms at day "15/16" to have 900 nm indentation at 450 nN. However, the results of Figure 1F (indentations of 750, 1450, and 2000 nm for class A, B, and C, respectively) could only be consistent with this if the vast majority of animals were in class A at 16 days. However, per the literature, only a minority are in class A at 16 days of adulthood at 20C.

- More problematic, is the comparison to Fig 1H, where day-15 WT at 20C is shown to have approximately 1700 nm indentation at 400 nN, over TWICE what was reported for what appear to be identical conditions in 1E (800 nm at 400 nN).

- The force-indentation curves for WT are similarly discordant between Figure 1E and 2B. For example, at 450 nN, Fig 1E shows 700 nm indentation for 12 days of adulthood, while Fig. 2B shows 1200 nm. At day 16, 2B shows ~1700 nm, which is consistent with Fig. 1H but not 1E.
- Figure 2F is labeled in the text (but not the legend) as showing 9-day-old worms. Here, WT indents ~550 nm at 450 nN force. In 2B, 8-day-old WT indents ~950 nM at the same force. At least 2F appears consistent with 1E, however.
- Figure 2H and 2B also appear discordant. In 2H, 10-day WT animals indent ~1300 nm at 450 nN, while in 2B, not even 12-day adults indent that far (~1100 nm).
- Figure 3B shows worms at day 14 (again the legend fails to mention that detail, but it fortunately is in the text) with WT having 1750 nm indentation at 450 nN. This is highly inconsistent with both Figure 2B (which suggests that WT would reach 1750 nm indentation between day 16 and 18) and Figure 1E (which suggests that WT would reach 1750 nm indentation only well after day 20).
- More worrisome, Figure 3C shows very different values for WT at day 14 than those shown in 3B! In 3C, the WT mean at day 14 is ~2750 nm at 450 nN, while in 3B the value is 1750 nm.
- Figures 3G and 3H are similarly discordant. The text (but not legend) identifies 3G as referring to 16-day-old animals, where the OP50/no-metformin control has indentation of ~1250 nm at 450 nN. (This is itself discordant both with 2B, which gives 1700 nm for those conditions, and with 1E, which gives 900 nm.) However, Fig 3H has no-metformin conditions indenting at approximately 2000 nm on days 15 and 18 -- a value that is dramatically different from 2B, 1E, or 3G.
- While the above discrepancies are all quantitative, some features of the authors' data do not even appear to replicate qualitatively. In particular, 2F shows a substantial (~1.6-fold) difference in indentation at 450 nN between WT and daf-16 at day 9 (~550 vs 875 nm). Figure 5E also has results for WT and daf-16, though the day that this comparison was made is not specified beyond stating that the worms are "aged". Nevertheless, the WT and daf-16 curves in 5E are within each other's error bars. So does daf-16 decrease cuticle stiffness or not? The authors' data offer two different answers.

Overall, the authors do not appear to have a great handle on the inter-replicate repeatability of their measurements. The fact that the procedures for gathering and analyzing the force-indentation curves are completely omitted from the methods section is not reassuring on this point. Many questions arise, such as:

- Where, anatomically, are the force measurements made on each worm? (Assuming that anatomical location was controlled for at all...)
- How repeatable are the force-indentation curves for nearby (say 1-2 microns) and more distant (10s to 100s of microns) locations on the same individuals?
- How repeatable are the force-indentation curves across individuals on the same plate? On neighboring plates in the same incubator? Across experiments conducted at different times?
- How are the force-indentation curves that are shown generated? To what do the error bars refer? In particular, the curves seem far too smooth, given the reported error bars (whether they are standard deviation or standard error). The curves seem to fall along an analytically smooth line, with almost zero experimental or sampling error introducing deviations from that line. How can this possibly be, given the size of the error bars?? (I suppose that if the error bars are standard deviations, and if each worm acts as a truly perfect spring over tested force range, but each worm has a different spring constant, then you would expect a smooth force-indentation curve with a large population standard deviation. Is this the case?)

Without these answers, and a set of carefully conducted controls, the authors claim that their stiffness measurements are "reproducible and quantitative" is not supportable.

(2) Given the smoothness of the force-indentation curves, it seems like they contain a lot of

redundant information. The authors should be able to measure some meaningful parameter that captures the relevant behavior of the entire curve for each individual -- something like an effective spring constant for each individual tested. This would make the results a lot easier to display over time and across genotypes, and to compare across figures.

(3) A summary statistic such as the above would make it more straightforward for the authors to provide a clear definition of "cuticle health". (E.g. time until the spring constant falls below some fixed value.) For the *daf-2* analysis (Fig 2C), the authors define "healthspan" as "time until the indentation at 450 nm is statistically significantly different from that at day 1", but this is not a good approach. For one, statistical significance is a function of both sample size and effect size, so in this scheme a larger population would have a shorter healthspan compared to a biologically identical but smaller population.

(4) In addition, the wide time-spacing of the *daf-2* samples precludes really accurately calculating a "health span" anyway. The first big decrease in stiffness in WT comes between days 4 and 8, while a comparable decrease in *daf-2* comes sometime between day 22 and 44. Given this, it seems odd to even attempt to calculate absolute or relative healthspan durations. Moreover, the authors use of "maximum lifespan" as the denominator for the relative healthspan figures is a bit questionable, given the statistical non-robustness of population maxima. Dividing healthspan by mean lifespan is probably a better choice.

(5) The authors should either define fixed criteria for interpreting the topographical images as "healthy" vs "unhealthy" or refrain from such interpretations. As is, it's far from clear what features the authors are examining to make some of the more subtle distinctions. In addition, all of the topographical images are shown with very different intensity scalings. This makes it quite difficult to compare the "image smoothness" (which appears to be the authors' main criterion) between an image with a wide intensity range and one with a narrow range. (The latter will naturally look more rough and noisy.) Ideally, the authors might devise some quantitative smoothness score to get around these issues of qualitative interpretation completely.

Moreover, I note that "healthier" images seem in general to have a wider range of depths. Do "healthy" cuticles have deeper grooves in general? Would an average groove-to-peak depth be a useful metric here?

(6) I have a significant complaint about the authors' use of "health" as a synonym for "cuticle stiffness" (or "cuticle integrity"). In some sense, the authors set out to answer the question "is cuticle stiffness a good measure of physiological health?" However, by textually equivocating between the two terms from the start of the text onward, the authors beg their own question.

At the outset and in the discussion, the authors make a good and clear point that "health" is tricky and multivalent, and that it's important to use a panel of measurements to properly characterize whatever "health" might actually be. I completely agree! As such, "cuticle stiffness" is at best one aspect of "health".

The data presented in this work are about cuticle stiffness, so the authors should use that language to describe their results. Really only in the discussion would it be appropriate to start asking the question "given the data we have presented, what is the role for measurements of cuticle stiffness in health and healthspan analysis"?

(7) The authors show a lot of results in the vein of "on a given day of adulthood, cuticle stiffness is higher in a particular longevity-extending mutant / condition than in the control". But a result like that would only be interesting if one's baseline null hypothesis is that longevity mutants in general alter only lifespan and are not expected to also alter the rate of physiological decline.

But most longevity mutants / conditions are already well-known alter a huge swath of physiology! E.g. for almost every physiological measurement anyone has ever cared to make, *daf-2* mutants are more "youthful" compared to age-matched WT. These days, it seems that the only reasonable null hypothesis to start out with is "longevity mutants slow physiological declines proportionately to their degree lifespan extension".

All of the recent results / debate regarding worm healthspans has been over the extent to which various mutations depart from that latter null hypothesis. E.g. the big surprises are finding aspects of physiology that are *not* affected by longevity mutants, or are affected to greater or lesser extents than one would expect based on the lifespan effects.

So in some sense, many of the experiments presented in this work are set up to ask the wrong question. Only when the authors compare populations at the same point on their survival curves, or show that lifespan-extending conditions *don't* slow cuticle softening / degeneration, are they presenting data that departs from the general expectation of the field these days.

Again, the real novelty in this work is not results like "cuticle stiffness is higher at day 10 in *daf-2* vs. WT", but in the claims about "cuticle stiffness is higher in *daf-2* vs. WT at a time when an equivalent fraction of the population has died" (or, better yet, plotting the percent-alive-vs.cuticle-stiffness phase space, as Podshivalova et al. did for movement rates).

As such, the authors should try to make as many of the latter types of claims as possible, and to be clear about where their results are basically confirming the "standard null model" (that longevity mutants in general slow physiological aging) vs. where the results depart from that model in interesting ways.

MINOR CONCERNS

(1) Given that the authors aim to establish cuticle stiffness as a measure of health, it would be helpful to have more overview in the Introduction regarding how mechanical stiffness is relevant for aging and physiological function. The authors provide some of this material in the Discussion, and moving relevant material from there into the Introduction would be helpful.

(2) Line 48: "Since ageing is the primary cause of ill health". This is a hypothesis masquerading as fact. It would be valid to say that aging is a primary "risk factor" for many diseases, but "causes" and "risk factors" are quite distinct. (I.e. "having lung cancer is a risk factor for being a smoker" is just as correct a statistical statement as the converse!) Aging and disease correlate, but the causation is far from clear. Perhaps ill health is the primary cause of ageing?

(3) Lines 63-66: "In fact, while it is commonly assumed in ageing research that genetic interventions that lead to lifespan increases would concomitantly lead to healthspan improvements and reduce end-of-life morbidity, supporting evidence is not conclusive." This is a blatant misrepresentation of the literature cited here by the authors! All of those papers clearly show that all genetic interventions tested increase the chronological span of basically every health measure tested. (There are a bunch more papers that show this too.) What's not clear is whether healthspan and lifespan are increased in the same proportion. But this is a different, more subtle (and perhaps even irrelevant!) matter.

(4) Line 69: "the frailty index". There are multiple competing frailty indices, and various related approaches! I think there are some reviews covering the different approaches, which the authors may prefer to cite.

(5) Line 76: "a frailty index in *C. elegans* and other model organisms is rarely implemented". The authors should also cite Zhang et al., which did in fact implement something very much like a frailty index for *C. elegans*.

(6) Line 90: "unbiased and artifact-free" is another hypothesis that the authors are presenting as fact.

(7) Line 93 (and throughout): "Importantly, we show that improvements in health and lifespan do not always correlate". No, the authors show that improvements in cuticle stiffness and lifespan do not always correlate. Whatever "health" is, "cuticle stiffness" is -- at best! -- only one particular aspect of it.

(8) Line 127: Should cite Hosono et al. as well as Herndon et al. for the movement classes. The

former predates the latter by a good 20 years.

(9) Line 141: It is a bit of a stretch to call temperature an "interventions that mediate[s] lifespan extension".

(10) Line 150: e1370 is not a knock-out. This allele is mis-represented in figure legends as "daf-2(0)". "daf-2(-)" would be more appropriate, or ideally just "daf-2(e1370)".

(11) Line 162 and throughout: "homeostatic pressure" is used but "hydrostatic pressure" would be more appropriate. Potentially the pressure maintenance is homeostatic (i.e. it is maintained at a set point through feedback from baroreceptors or something), but that is a specific biological hypothesis.

(12) Line 238-241: "Comparison between physiologically matched phm-2 mutants and wild-type worms at approximately their mean lifespan age (day 17 for wild-type and day 21 for phm-2) shows that reduction of nutrient intake driven by mechanical (Figure 3C) rather than neuronal mechanisms as in eat-2 mutants (Figure 3D) preserves homeostatic pressure with age." This is a stretch at best. There are a lot of things different between eat-2 and phm-2 beyond the mechanical vs. neuronal difference in the DR. All that can reliably be concluded is that different mutants that both seem to extend lifespan via DR have different effects on the cuticle. This could be due to the specific mechanisms of the DR, or due to some other pleiotropies of the mutants.

(13) Line 255: The authors (of all people) should know that "blue fluorescence accumulation" is not a particularly good measure of health (as compared to a measure of dying worms).

(14) Line 282-283: "Finally, we also show that metformin slightly reduces cuticle senescence with ageing (Figure 3I)". What criteria are being applied here? It's hard to see much difference at all in the images shown.

(15) Throughout, the authors persist in describing many treatments which have fairly broad effects on bacteria as "metabolic alterations". For example, line 290: "Notably, changes in bacterial metabolism induced by UV-irradiation, antibiotic treatment or host-targeted drugs like metformin extend *C. elegans* lifespan". It seems to be a bit of a stretch to pin the lifespan extension in all of these cases on "changes in metabolism" -- as compared to, say, decreases in bacterial proliferation. At best, it could be a mix of both. Indeed, doesn't UV treatment primarily prevent replication while largely leaving metabolism intact?

Similar concerns apply to line 300, where the authors discuss "impaired bacterial metabolism using heat and UV-irradiation". It seems like a rather severe understatement to describe heat-inactivating bacteria (which does many things, including denaturing proteins and even permeabilizing cell walls and membranes) as a "metabolic impairment".

(16) Line 308-311: "However, force-indentation curves and 450 nN force indentation values are significantly lower for wild-type worms fed UV-irradiated *E. coli* OP50 but not heat-killed bacteria (Figure 4B), suggesting that bacterial metabolism rather than bacterial proliferation regulates healthspan in worms". I'm not sure how this follows.

The authors compare metabolically active, non-proliferating bacteria (UV), which have an effect, to metabolically inactive, non-proliferating bacteria (heat), which do not have an effect. So at best the authors can claim that, among their limited sample of treatments, lack of bacterial proliferation is necessary but not sufficient to improve cuticle stiffness. But even that is likely to be an over-generalization from insufficient evidence. For example, given that the two treatments extend lifespan by different degrees, it might also be that the authors are simply looking too late in life, at a point by which heat-killed bacteria no longer improve the cuticle, but UV treatment still does. (As with most everywhere else, the authors fail to state the day on which the measurements in Fig. 4B were made.)

(17) Line 314-315: "carbenicillin, which kills bacteria by inhibiting bacterial cell wall synthesis". What does "kills" mean? Prevents replication? Prevents metabolic activity? The authors were

previously careful about this distinction, so why not here?

(18) Lines 320-321: "bacterial proliferation per se is not a driving factor for improved homeostatic pressure in aged worms". Again, this is not supported by the authors' data! Except in the case of metformin, all the stiffness-improving regimes involved bacteria that cannot proliferate. So the best the authors can claim is that some but not all conditions that render bacteria non-proliferative can stiffen the cuticle. I.e. that preventing proliferation is not sufficient -- but may be necessary -- for improving the cuticle.

(21) Lines 342-343: "Overall, our data support the hypothesis that viable but metabolically altered bacteria can improve homeostatic maintenance with age." Again, this is not supported by the data, and is also terminologically confused. I assume here that "viable" means "metabolically active but not necessarily capable of further cell division"? But that's not totally clear.

As above, all the conditions that do stiffen the cuticle involve bacteria that are not actively proliferating. However, some conditions like heat and carbenicillin that stop proliferation fail to stiffen the cuticle. Absent clear mass-spec / Seahorse / whatever results about what's going on metabolically in UV- vs. carb- vs. TMP-treated bacteria, it's hard to claim that there's some set of metabolic alterations that is specifically responsible for the cuticle stiffening.

Clearly not all treatments that prevent bacterial replication improve the cuticle. But saying that it is metabolic alterations specifically that cause cuticle improvement seems to be a significant overreach. What if carbenicillin and heat-killing just render the bacteria indigestible in some way that cancels out the health benefit of preventing proliferation? That possibility is just as consistent with the data as the authors' claim. So absent any real data about mechanism, the authors should really just stick to stating that different bacterial treatments have different effects on the cuticle.

(22) Lines 350-351: "Taken together, our data convincingly show that bacterial metabolism is a key determinant for preserving pressure homeostasis and therefore health with age". As above, this reviewer strongly disagrees about how convincing a case is made for metabolism per se.

Reviewer #2 (Remarks to the Author):

The paper by Essmann et al. investigates the mechanical properties of *C. elegans* by AFM and relates those measurements to the healthspan of the organisms. While the approach is highly interesting and timely and could be of interest to a wider audience, the overall presentation hampers the reviewer's enthusiasm somewhat.

First, the figures are too crowded and highly repetitive in their appearance. They should be streamlined and condensed to essential results while all other graphs should be moved to Supp Info.

Since AFM is the central technique, it must be explained in much more detail. Although the authors refer to an publication about the method, it is essential for the understanding to mention more details. It would be good to show a typical raw force indentation plot in SI.

In particular, it is not clear how exactly mechanical properties are analyzed and quantified other than the indentation at a particular force. Throughout the manuscript the term 'homeostatic pressure' is used many times, however the authors do not explain how they determine this 'pressure' from their experiments.

In general, a real quantification of the 'new biomarker' in terms of mechanical properties is needed that takes in particular the tip geometry into account so that other groups can also relate their measurements when not using the exact same measurement geometry.

Quantification of the topography images is also needed.

Temperature: The authors state that 'temperature is a major factor ...'. What was the temperature for AFM measurements and would that affect mechanical properties?

How exactly were the worms immobilized? Is this affecting the measurements?

minor issues:

the title should be reworded to include 'mechanical properties' or similar

abstract:

'loss of biomechanical properties' makes no sense. The sample still exhibits mechanical properties, however e.g. the stiffness can be higher or lower, or mechanical integrity could be lost. Please reword.

It is essential to mention AFM measurements and obtained mechanical properties in the abstract.

Why is the force in Fig 1 H only up to 400 nN when all other plots are up to 450 nN?

In the Methods section AFM, at the end of the last sentence there seems the reference Essmann et al 2017 NBM was not correctly inserted.

I. 428 cells are in addition also under contractile stress from their own acto-myosin cytoskeleton and neighboring cells as well as 'pressure' due to confined space.

We have utilised here a colour code to indicate the reviewer's comments in blue and our response in black. Also, to simplify the reviewer's task we have combined points addressing the same question. This has been indicated when needed.

Reviewer 1:

SUMMARY

In this work, the authors use atomic force microscopy to characterize patterns of cuticle aging in *C. elegans*. The author establish that stiffness decreases with age in wild-type animals. Stiffness appears to be related to movement during aging as animals with decreased movement tend to be less stiff. The authors use this measure of cuticle quality to study physiological function in genetic and dietary restriction (DR) paradigms that alter lifespan. Loss-of-function alleles in the insulin-signaling pathway increase maintenance of stiffness in a *daf-16* dependent manner; the authors suggest that these perturbations increase both absolute and relative duration of life spent with good cuticle quality. Surprisingly, effects of genetic models of DR on stiffness are variable; one genetic model of DR shows increased maintenance of stiffness while another model shows decreased stiffness compared to wild-type. A DR mimetic that alters bacterial metabolism, metformin, also increases maintenance of stiffness. These findings prompt the authors to investigate the effect of bacterial metabolism on cuticle quality. The authors show that some but not all perturbations to standard *E. coli* food bacteria increase maintenance of stiffness. Finally, the authors test the effects of feeding on *B. subtilis* and *C. aquatica* on stiffness.

IMPRESSION

Overall, this is an interesting characterization of a new and potentially valuable marker of the process of senescence. The authors rightly conclude that many different aspects of so-called "health" are surprisingly uncorrelated with lifespan and with one another, so broadening the repertoire of measures of "health" will be critical to better understanding aging and senescence. In addition, the authors have performed experiments to map out the effects of a large variety of interventions on cuticle stiffness and integrity, which would do a great deal to establish these as useful measures of health -- if the data appeared to be reliable. However, the data, as presented, seem to be quite irreplicable even between seemingly identical experiments in the authors' lab. Perhaps these are minor errors in data formatting or in my understanding that can be explained on revision. Otherwise, however, I'm not sure how much of what the authors claim to conclude is really supported by the data.

We thank the reviewer for their appreciation of our work and its potential. We will address in detail each of the reviewer's queries/points and provide further evidence that the data presented is replicable. Please see below.

This manuscript is also marred by substantial inattention to detail, such as failing to specify key experimental parameters (e.g. what temperature were the worms raised at for various experiments? What days of adulthood were the measures made for each figure panel? Where anatomically were AFM force measurements made?), lack of analytic/statistical details (e.g. how were force-indentation curves calculated? what do the error bars represent?), and carelessness in nomenclature (e.g. *daf-2(e1370)* is not a null allele, and should not be referred to as a "knockout" or as "*daf-2(0)*").

We apologise for not having provided some of these details, which add to the clarity of the manuscript and we appreciate the effort of the reviewer to point these out. In the revised version, all those points have been fully addressed.

What temperature were the worms raised at for various experiments?

We included the following sentence (in red) to the previous paragraph in the material and methods section: Lifespan measurements were performed as follows. Axenic worm eggs were obtained using alkaline hypochlorite treatment of gravid adult hermaphrodites that had been kept in optimal temperature and feeding conditions for at least 3 generations. These were then placed onto plates containing the test bacterial strain and maintained at 20°C. Lifespan measurements were initiated by transfer of L4-stage worms (day 0) to plates containing bacteria grown in the presence or absence of treatment for 96 h. Worms were transferred to fresh plates every day during the reproductive period and thereafter, every other day until day 12. For the temperature shifting experiment, worms grown at 20°C until the L4 stage were transferred to plates and placed experimentally at 15°C, 20°C and 25°C for the rest of their lifespan measurement. **For the *glp-1* mutant experiments, N2 and respective *glp-1*, *glp-1,daf-16* and *daf-16* mutants worms were raised from L1 to L4 at 25°C and shifted to 20°C at the L4 developmental stage. For the *daf-2* mutant experiments, all animals were grown and maintained at 20°C.**

What days of adulthood were the measures made for each figure panel?

The days for each measurement are now clearly indicated in each figure panel.

Where anatomically were AFM force measurements made?

We have now included additional data in figure S1A, displaying the areas where measurements have been performed throughout the entirety of our experiments in this manuscript. Further details have also been included in the material and methods section. Briefly, we have made measurements in the so-called neck and hip areas (Essmann et al., 2018, Plos one PMID:30240421).

lack of analytic/statistical details (e.g. how were force-indentation curves calculated? what do the error bars represent?),

We have now included a new figure panel S1B to show how the force indentation curves are obtained. As suggested by both reviewers we have also now adopted the use of Young's Modulus as a measure of the material properties, which takes into account the entire curve, and properties of the indenter. These have replaced the previous force curve panels. The only three remaining force curve panels in the manuscript (Fig 1C, S1B, S1C) display mean force indentation curve and error bars as SEM. This was previously indicated in the figure legends and remains clearly indicated in the new figure legends.

and carelessness in nomenclature (e.g. *daf-2(e1370)* is not a null allele, and should not be referred to as a "knockout" or as "*daf-2(0)*").

We have now adopted a clear nomenclature using the name of the genes in italics and the respective alleles both in the figures and the main text.

The authors also make a series of over-broad claims about "bacterial metabolism" being a critical variable with regard to changes in cuticle stiffness under different feeding regimes, but (as described below), these claims dramatically outpace the actual evidence that the authors provide.

This point has been fully addressed below in the Minor points sections 15-22. Briefly, in addition to the evidence presented in the manuscript which may have been overlooked by the reviewer (e.g. Metformin, Trimethoprim (concentrations used below MIC) and bacterial mutants *gltA* and *sucA*- which represent conditions where bacterial cells can proliferate but are metabolically impaired), we now provide further evidence that bacterial physiology (e.g. additional bacterial metabolomics data, bacterial growth data, bacterial respiration data and new stiffness experiments with different nutrients) is indeed one of the major aspects regulating these two novel parameters of health (see below in

minor points for comments addressing specific points). We have made further adjustments to accommodate the reviewer's comments, and we are now stating bacterial physiology (a more encompassing term which refers to proliferation, metabolism, growth, respiration and other parameters) instead of bacterial metabolism, and in agreement with our data to describe our findings in the most scientific and accurate way.

MAJOR POINT 1

(I) The authors' own data make the AFM force-indentation measurements appear to be highly irreproducible, either quantitatively or qualitatively. The following is a non-exhaustive list of places where the authors' data appear to be internally inconsistent:

Fig 1E shows WT worms at day "15/16" to have 900 nm indentation at 450 nN. However, the results of Figure 1F (indentations of 750, 1450, and 2000 nm for class A, B, and C, respectively) could only be consistent with this if the vast majority of animals were in class A at 16 days. However, per the literature, only a minority are in class A at 16 days of adulthood at 20C.

- More problematic, is the comparison to Fig 1H, where day-15 WT at 20C is shown to have approximately 1700 nm indentation at 400 nN, over TWICE what was reported for what appear to be identical conditions in 1E (800 nm at 400 nN).

- The force-indentation curves for WT are similarly discordant between Figure 1E and 2B. For example, at 450 nN, Fig 1E shows 700 nm indentation for 12 days of adulthood, while Fig. 2B shows 1200 nm. At day 16, 2B shows ~1700 nm, which is consistent with Fig. 1H but not 1E.

- Figure 2F is labeled in the text (but not the legend) as showing 9-day-old worms. Here, WT indents ~550 nm at 450 nN force. In 2B, 8-day-old WT indents ~950 nm at the same force. At least 2F appears consistent with 1E, however.

- Figure 2H and 2B also appear discordant. In 2H, 10-day WT animals indent ~1300 nm at 450 nN, while in 2B, not even 12-day adults indent that far (~1100 nm).

- Figure 3B shows worms at day 14 (again the legend fails to mention that detail, but it fortunately is in the text) with WT having 1750 nm indentation at 450 nN. This is highly inconsistent with both Figure 2B (which suggests that WT would reach 1750 nm indentation between day 16 and 18) and Figure 1E (which suggests that WT would reach 1750 nm indentation only well after day 20).

- More worrisome, Figure 3C shows very different values for WT at day 14 than those shown in 3B! In 3C, the WT mean at day 14 is ~2750 nm at 450 nN, while in 3B the value is 1750 nm.

- Figures 3G and 3H are similarly discordant. The text (but not legend) identifies 3G as referring to 16-day-old animals, where the OP50/no-metformin control has indentation of ~1250 nm at 450 nN. (This is itself discordant both with 2B, which gives 1700 nm for those conditions, and with 1E, which gives 900 nm.) However, Fig 3H has no-metformin conditions indenting at approximately 2000 nm on days 15 and 18 -- a value that is dramatically different from 2B, 1E, or 3G.

Quantitatively: We appreciate the effort of the reviewer in highlighting what appear to be inconsistencies within our datasets. We have now performed additional experiments as suggested by the reviewers and re-plotted all our data previously presented in the paper together with the new data as Young's Modulus calculated using the Hertz model, which considers the entire shape of the curve for each worm, rather than representing solely the indentation value at 450 nN force. We have performed additional experiments to demonstrate show the replicability of our data between distinct

locations in the worm (Figure S1A), between different worms in one plate and between plates run in parallel in the same experiment (Figure S1C) and between experiments throughout the entire study (data shown for matching days – Day 1, Day 12 and Day 18) for WT worms growing and maintained in the same conditions (Figure S1D). One can now observe that our measurements are replicable inter-experiments and over-time (spanning over three years of work).

We do agree that there is variability in our datasets (Figure S1D), and in particular in the datasets of Figure 4D and 4F. We have performed additional experiments for this revision and show that changes in the agar composition between different lots of the same catalog number or different peptone compositions can influence worm stiffness with age (Figure S6A, B). Importantly, we always include a control condition that is run in parallel with the condition being tested, and therefore, despite specific values vary the effects are robust (addressed below). We would also like to stress that force indentation measures rely on knowing the stiffness of the indenter (cantilever), which despite being calibrated prior to each experiment can vary in accuracy due to technical reasons and will hence result in slightly lower or higher values but that remain constant for the entire experiment. Moreover, we would like to stress that the maximum variability observed between independent experiments performed on the same day of worm age for Figure S1D is 1.66-fold for day 1 and 1.55-fold for day 18, compared to the mean of the values obtained for day 1 and day 18, respectively, from different experiments. Importantly, compared to a 8.93-fold difference between the mean of YM values obtained at day 1 and 18, and given that the AFM allows us to make reliable measurements of worms as stiff as a conservative value of approximately 2000 kPa and as soft as 4-8 kPa, hence an approximate >250-fold difference, we argue a quantifiable difference of 1.66 between experiments is an acceptable error. We can also take into account that technical errors from the instrument due to deviations in cantilever calibration can account for up to a 1.20- fold difference between experiments (<http://www.nanophys.kth.se/nanophys/facilities/nfl/afm/jpk/manuf-manuals/handbook-2.2a.pdf>).

Overall, taken aside explainable technical discrepancies, we do not disagree that there is inter-experimental variability but we are confident that the reviewer appreciates this is also a long-standing observable effect in all published ageing studies when measuring diverse outputs of the ageing process. For example, this has been specifically addressed in a recent study by Lucanic et al., 2017 Nature Comms; PMID: 28220799 by the labs of Lithgow, Driscoll and Phillips. I copy here an extract from the paper: **“Replicate variation within each laboratory is relatively high. Interestingly, although we found that systematic differences among labs were minor, we calculated replicate-to-replicate variation within each lab to be relatively high. After accounting for other sources of variation, strong among-replicate differences remained, representing roughly 15% of the total variation in individual lifespan observed in our study (9% derived from the trial-specific effects, 6% from the among-plate in same trial differences and none from experimenter-specific differences). Thus, although the results obtained on any given day of a replicate trial tended to be fairly consistent with one another, conducting the same assay a month later could yield results as different as looking at a strain from a different species.**

Given that we observe a relatively large amount of variation among trials across each of the three labs, despite strict adherence to standardized procedures and culture conditions, we conclude that a major challenge to reproducibility in this system may arise from trial-specific cohort responses to unidentified and apparently subtle differences in the assay environment, which vary similarly within each laboratory”

Similarly, mean lifespans from recent studies from other well-established *C. elegans* ageing labs, show discrepancies in the mean lifespans of their inter-experimental replicates when performed in seemingly the exact same experimental conditions in the same paper (Kenyon lab- Podishivalova et

al., Cell reports, 2017; PMID: 28423308; range from 13.23 to 19.04), (Gems and Pincus lab – Zhao et al. 2017 Nature Comms, PMID: 28534519; range from 16.6-19.1) Dillin lab - C. Daniel de Magalhaes Filho 2017 Nature Comms, PMID: 29500338; range from 13.7 to 19.4).

Similarly, as observed in our study, other age-related pathology measurements from the Gems lab in wild type cohorts can vary importantly between inter-experiments in aged wild-type worms (e.g. Ezcurra et al., Curr Biol. PMID:30352178 – Pathological Lipid Pool scores can vary at day 14 from 20% to 50% or Relative Intestinal Width at day 14 from 50% +/- 5% or 25% +/- 20%).

More recently, work by the Stroustrup/Riedel lab (Janssens et al., Cell reports 2019, PMID: 30970250), using a fully automated approach – the lifespan machine - for measuring worm movement, shows not only inter-experimental variability between 4 independent N2 worm populations measurements but also distinct trajectories and heterogeneity in movement decay over time between isogenic populations of worms that are ageing in seemingly identical conditions.

Figure S5. Population health span derived from the ‘lifespan machine’, related to Figure 4 and STAR Methods

This is an accompanying figure for steps described in the methods section. (A) Worm positions were assessed based on the position of worm objects identified in scanner images from the ‘lifespan machine’. Left panel: Worm positions identified at an early time point (representing young animals). Abundant changes in worm locations can be seen between consecutive frames. Right panel: The same plot, but at a late time point (representing old animals), showing that worms now remain mostly in the same position. (B) Assessing correlations between subsequent time points as described in (A), shows trends towards increasing correlations in time, indicating less movement of the population. Four independent N2 worm populations are shown. (C) Merging all replicates and fitting a smoothed spline provides a general consensus of the population’s trend.

In addition, the great work from several labs working in this area, is developing tools and approaches that have already importantly contributed to explaining variability between worms of the same cohort (e.g. Zhang et al., Cell systems 2016, PMID: 27720632). Yet, many of the factors that impact the lifespan, pathology or healthspan trajectories in wildtype worms throughout ageing remain to be identified.

Overall, we acknowledge this is a long-standing issue in the field which does not invalidate our approach as we always include a wildtype control for each experimental comparison.

While the above discrepancies are all quantitative, some features of the authors' data do not even appear to replicate qualitatively. In particular, 2F shows a substantial (~1.6-fold) difference in indentation at 450 nN between WT and daf-16 at day 9 (~550 vs 875 nm). Figure 5E also has results for WT and daf-16, though the day that this comparison was made is not specified beyond stating that the worms are "aged". Nevertheless, the WT and daf-16 curves in 5E are within each other's error bars. So does daf-16 decrease cuticle stiffness or not? The authors' data offer two different answers.

Qualitatively: We thank the reviewer for bringing this point to our attention. We investigated what could be the source of this discrepancy within the WT population. After looking carefully into our records, we noticed that the WT strain used as a control in experiments of old Figure 1E and 2F were

performed with the CGCM male stock rather than the widely used CGCH hermaphrodite stock. Since, it has come to our attention (communication with David Gems and work presented by his group at the Madison Worm Meeting in 2018) that the worms from the CGCM stock are mutants for the *fln-2* gene (<http://wbg.wormbook.org/2018/12/11/n2-male-is-a-long-lived-fln-2-mutant/>) and show fewer pathologies with age compared to the CGCH. We have therefore performed an experiment comparing these two strains and show that CGCM males are indeed stiffer with age when compared to CGCH worms (See below). We have now added a new dataset of CGCH (WT) longitudinal measurement of stiffness with age to Figure 1C and 1D. If reviewer feels this is relevant information, we are happy to include it in the paper.

We have also performed a new longitudinal comparison with age of WT vs *daf-16* (Figure S2B) and replaced old figure 2F by novel data in Figure 2C with the adequate WT CGCH controls that shows no differences in stiffness between the WT and *daf-16* mutants and is consistent with data in other datasets throughout the paper such as (Figure 2C, 2E, S2A, S2B, S2C, S4D, 5F).

A) Mechanical properties as Young's Modulus (YM; kPa) of wild type *C. elegans* strain from the CGC male stock (CGCM) at different ages. Error bars indicate 95% CI, dotted line marks mean YM within 95% CI at day 1. **B)** Mechanical properties as YM (kPa) of wild type hermaphrodite stock CGCH (blue) and male stock CGCM (red) *C. elegans* at 12 and 18 days of age. Error bars indicate 95% CI. One-way ANOVA test n.s. = not significant, ***= $p < 0.001$.

Overall, the authors do not appear to have a great handle on the inter-replicate repeatability of their measurements. The fact that the procedures for gathering and analyzing the force-indentation curves are completely omitted from the methods section is not reassuring on this point. Many questions arise, such as:

- Where, anatomically, are the force measurements made on each worm? (Assuming that anatomical location was controlled for at all...).

We apologise for not having included this information. We have made measurements in the so-called neck and hip areas (Essmann et al., 2018, Plos one PMID:30240421). We have now included a schematic drawing to illustrate this (Figure 1A and S1A) and referred to it in the material and methods section.

-How repeatable are the force-indentation curves for nearby (say 1-2 microns) and more distant (10s to 100s of microns) locations on the same individuals?

We have now included new data, which show no statistical significant difference in measurements made at two distinct anatomical locations (neck and hip) but significant compared to the mid body.

- How repeatable are the force-indentation curves across individuals on the same plate? On neighboring plates in the same incubator?

We have now included additional data (Figure S1C) showing that different worms show distinct stiffness profiles within the same plate but no significant differences are observed in neighbouring plates run in parallel in different locations in the same incubator.

Across experiments conducted at different times? Addressed previously.

How are the force-indentation curves that are shown generated? To what do the error bars refer? In particular, the curves seem far too smooth, given the reported error bars (whether they are standard deviation or standard error). The curves seem to fall along an analytically smooth line, with almost zero experimental or sampling error introducing deviations from that line. How can this possibly be, given the size of the error bars?? (I suppose that if the error bars are standard deviations, and if each worm acts as a truly perfect spring over tested force range, but each worm has a different spring constant, then you would expect a smooth force-indentation curve with a large population standard deviation. Is this the case?)

The force-indentation-curves are generated by the AFM through measuring the displacement in relation to the increasing force. Each curve is made up of individual force displacement points (XY) at a sampling rate of 2048 Hz. A number of "n" curves were measured per experiment. These curves were then averaged and represented as mean force indentation curve and the error bars indicate SEM. This is now clearly shown in figure S1B and described in the material and methods. In the previous manuscript, we chose to display the force curves as such as they show the mean of the original dataset for each worm using no model fitting. These curves are now only seen in Figure 1C, S1B, S1C.

Without these answers, and a set of carefully conducted controls, the authors claim that their stiffness measurements are "reproducible and quantitative" is not supportable.

We have provided additional experimental evidence with carefully conducted controls together with supplementary material and methods highlighting data acquisition, reproducibility and quantification to support our claim.

MAJOR POINT 2

Given the smoothness of the force-indentation curves, it seems like they contain a lot of redundant information. The authors should be able to measure some meaningful parameter that captures the relevant behavior of the entire curve for each individual -- something like an effective spring constant for each individual tested. This would make the results a lot easier to display over time and across genotypes, and to compare across figures.

Following your advice and also as suggested by reviewer 2, we have now calculated the Young's modulus from each one of these individual worm curves, giving a more universal measure of material properties, which also considers the shape of both the indenter and the curve. This has improved and streamlined the paper allowing for more robust comparisons between conditions and reflecting more accurately the individual variability in our measurements (rather than the previously used 450 nN force indentation depth).

MAJOR POINT 3

(3) A summary statistic such as the above would make it more straightforward for the authors to provide a clear definition of "cuticle health". (E.g. time until the spring constant falls below some fixed value.) For the *daf-2* analysis (Fig 2C), the authors define "healthspan" as "time until the indentation at 450 nm is statistically significantly different from that at day 1", but this is not a good

approach. For one, statistical significance is a function of both sample size and effect size, so in this scheme a larger population would have a shorter healthspan compared to a biologically identical but smaller population.

We agree and thank the reviewer for this comment. We have now included a summary statistic line at mean Young's Modulus value for all our experiments performed at day 1, to allow comparisons throughout the manuscript. Additionally, for our longitudinal measurements, statistical comparisons have only been made between Young's Modulus values at mean lifespan for the different conditions.

MAJOR POINT 4 and 7.

Major point (4) In addition, the wide time-spacing of the *daf-2* samples precludes really accurately calculating a "health span" anyway. The first big decrease in stiffness in WT comes between days 4 and 8, while a comparable decrease in *daf-2* comes sometime between day 22 and 44. Given this, it seems odd to even attempt to calculate absolute or relative healthspan durations. Moreover, the authors use of "maximum lifespan" as the denominator for the relative healthspan figures is a bit questionable, given the statistical non-robustness of population maxima. Dividing healthspan by mean lifespan is probably a better choice.

We agree. We have now performed measurements of both stiffness and cuticle roughness at mean lifespan instead for all the conditions tested and also included additional data points for the *daf-2* longitudinal study and the other main conditions of the manuscript (Figures 2B, 3B, 4B and 5B).

Major point (7) The authors show a lot of results in the vein of "on a given day of adulthood, cuticle stiffness is higher in a particular longevity-extending mutant / condition than in the control". But a result like that would only be interesting if one's baseline null hypothesis is that longevity mutants in general alter only lifespan and are not expected to also alter the rate of physiological decline.

We agree. This has been addressed in the previous point.

But most longevity mutants / conditions are already well-known alter a huge swath of physiology! E.g. for almost every physiological measurement anyone has ever cared to make, *daf-2* mutants are more "youthful" compared to age-matched WT. These days, it seems that the only reasonable null hypothesis to start out with is "longevity mutants slow physiological declines proportionately to their degree lifespan extension".

We agree. This has been addressed in the previous point and our new data supports the point raised by the reviewer.

All of the recent results / debate regarding worm healthspans has been over the extent to which various mutations depart from that latter null hypothesis. E.g. the big surprises are finding aspects of physiology that are *not* affected by longevity mutants, or are affected to greater or lesser extents than one would expect based on the lifespan effects.

We agree. Our new datasets by comparing our two health span parameters at mean lifespan support this idea that some interventions depart from the null hypothesis. For example, *daf-2* or *glp-1* mutants, which extend lifespan also have higher stiffness and lower cuticle roughness when measured at mean lifespan. On the other hand, we have tested genetic mutants such as *eat-2* that extend lifespan without concomitantly increasing stiffness compared to WT, deviating from the null hypothesis and consistent with previous published data. Finally, we show that conditions such as raising and ageing *C. elegans* on *Comamonas aquatica*, shortens lifespan but improves stiffness chronologically and at mean lifespan compared to control, a result that has not been previously reported.

So in some sense, many of the experiments presented in this work are set up to ask the wrong

question. Only when the authors compare populations at the same point on their survival curves, or show that lifespan-extending conditions *don't* slow cuticle softening / degeneration, are they presenting data that departs from the general expectation of the field these days.

We agree. This has been addressed above.

Again, the real novelty in this work is not results like "cuticle stiffness is higher at day 10 in daf-2 vs. WT", but in the claims about "cuticle stiffness is higher in daf-2 vs. WT at a time when an equivalent fraction of the population has died" (or, better yet, plotting the percent-alive-vs.cuticle-stiffness phase space, as Podshivalova et al. did for movement rates).

We agree. Now all comparisons have also been performed at mean lifespan.

As such, the authors should try to make as many of the latter types of claims as possible, and to be clear about where their results are basically confirming the "standard null model" (that longevity mutants in general slow physiological aging) vs. where the results depart from that model in interesting ways.

We agree. All conditions/interventions have now been tested at mean lifespan.

MAJOR POINT 5

(5) The authors should either define fixed criteria for interpreting the topographical images as "healthy" vs "unhealthy" or refrain from such interpretations. As is, it's far from clear what features the authors are examining to make some of the more subtle distinctions. In addition, all of the topographical images are shown with very different intensity scalings. This makes it quite difficult to compare the "image smoothness" (which appears to be the authors' main criterion) between an image with a wide intensity range and one with a narrow range. (The latter will naturally look more rough and noisy.) Ideally, the authors might devise some quantitative smoothness score to get around these issues of qualitative interpretation completely.

We appreciate the suggestion made by the reviewer. This was also highlighted by reviewer 2. With increasing age of the worm, in particular the annuli region shows breaks and gaps, this is what we refer to as cuticle senescence now quantified as roughness. Following the AFM manufacturer's (JPK) suggestions and instructions, we have now introduced the quantification of a roughness factor to better define cuticle health. How this has been measured is now also fully described in the material and methods section. These new data have provided a better analytical approach to capture visible cuticle deterioration in a fully quantitative manner, which allowed us to assess differences between genotypes and environmental conditions at mean lifespan.

Moreover, I note that "healthier" images seem in general to have a wider range of depths. Do "healthy" cuticles have deeper grooves in general? Would an average groove-to-peak depth be a useful metric here.

We thank the reviewer for providing this suggestion. As AFM records images by scanning over the surface it also records height changes. Height changes can be due to the natural curvature of the worm, which is a tube and not a flat surface, but also depend on worm length and width. Worms are still gaining length and width after the last cuticle change (molt) into adulthood, when annuli and furrow number is fixed. Naturally, when the worm expands in length and width, the cuticle stretches by anatomically flattening the annuli region and possibly by widening the furrows. Overall, the groove-to-peak depth can become less noticeable with age as seen in figure 1G, but this phenomenon does not reflect the observable differences in cuticle senescence prominent mostly in the annuli region obtained by the high resolution of AFM. To quantify cuticle senescence instead, we now analysed the

roughness of the cuticle in the anulli for all conditions at mean lifespan, as suggested by our collaborator Andre Brown and the JPK representative. This is now thoroughly described in the material and methods section.

MAJOR POINT 6.

(6) I have a significant complaint about the authors' use of "health" as a synonym for "cuticle stiffness" (or "cuticle integrity"). In some sense, the authors set out to answer the question "is cuticle stiffness a good measure of physiological health?" However, by textually equivocating between the two terms from the start of the text onward, the authors beg their own question.

At the outset and in the discussion, the authors make a good and clear point that "health" is tricky and multivalent, and that it's important to use a panel of measurements to properly characterize whatever "health" might actually be. I completely agree! As such, "cuticle stiffness" is at best one aspect of "health".

The data presented in this work are about cuticle stiffness, so the authors should use that language to describe their results. Really only in the discussion would it be appropriate to start asking the question "given the data we have presented, what is the role for measurements of cuticle stiffness in health and healthspan analysis"

We thank the reviewer by this suggestion. We have now changed the entire manuscript replacing healthspan to stiffness or cuticle senescence, when appropriate. The terms are no longer used interchangeably.

MINOR CONCERNS.

(1) Given that the authors aim to establish cuticle stiffness as a measure of health, it would be helpful to have more overview in the Introduction regarding how mechanical stiffness is relevant for aging and physiological function. The authors provide some of this material in the Discussion, and moving relevant material from there into the Introduction would be helpful.

We agree and thank the reviewer for this suggestion. We have moved the paragraph "The mechanical properties of cells... tissue decay during ageing" into the introduction.

(2) Line 48: "Since ageing is the primary cause of ill health". This is a hypothesis masquerading as fact. It would be valid to say that aging is a primary "risk factor" for many diseases, but "causes" and "risk factors" are quite distinct. (I.e. "having lung cancer is a risk factor for being a smoker" is just as correct a statistical statement as the converse!) Aging and disease correlate, but the causation is far from clear. Perhaps ill health is the primary cause of ageing?

We have rephrased to "Since ageing is a primary risk factor for many diseases,..." – Line 50

(3) Lines 63-66: "In fact, while it is commonly assumed in ageing research that genetic interventions that lead to lifespan increases would concomitantly lead to healthspan improvements and reduce end-of-life morbidity, supporting evidence is not conclusive." This is a blatant misrepresentation of the literature cited here by the authors! All of those papers clearly show that all genetic interventions tested increase the chronological span of basically every health measure tested. (There are a bunch

more papers that show this too.) What's not clear is whether healthspan and lifespan are increased in the same proportion. But this is a different, more subtle (and perhaps even irrelevant!) matter.

We have rephrased the text to state “While the effects on lifespan for a few genetic, dietary and pharmacological interventions are well-documented and show improvements in health measures at chronological age^{6,18-20}, it is less clear whether lifespan and healthspan are increased in the same proportion” - Line 63

(4) Line 69: "the frailty index". There are multiple competing frailty indices, and various related approaches! I think there are some reviews covering the different approaches, which the authors may prefer to cite.

Thank you. We agree with the suggestion and have included the review from Clegg A, 2013 Lancet PMID: 23395245. We have also rephrased the sentence to better reflect the existence of several frailty indices.

“In humans, several frailty indices exist to characterise different inter-related pathophysiological parameters that are associated with loss of health during ageing and encompass visible exterior signs of decay to identify the elderly” - Line 68

(5) Line 76: "a frailty index in *C. elegans* and other model organisms is rarely implemented". The authors should also cite Zhang et al., which did in fact implement something very much like a frailty index for *C. elegans*.

Thank you. We agree with the suggestion and have included this reference in Line 72.

(6) Line 90: "unbiased and artifact-free" is another hypothesis that the authors are presenting as fact.

We have amended this sentence. “...we provide a detailed characterisation of two additional age-related measures in this model organism.” – Line 100

(7) Line 93 (and throughout): "Importantly, we show that improvements in health and lifespan do not always correlate". No, the authors show that improvements in cuticle stiffness and lifespan do not always correlate. Whatever "health" is, "cuticle stiffness" is -- at best! -- only one particular aspect of it.

We have rephrased it to “we show that improvements in our health parameters and lifespan do not always correlate, and that similar lifespan-extending interventions can lead to dissimilar outcomes in these health parameters.” – Line 103

(8) Line 127: Should cite Hosono et al. as well as Herndon et al. for the movement classes. The former predates the latter by a good 20 years.

Thank you. We have included this reference. – Line 136

(9) Line 141: It is a bit of a stretch to call temperature an "interventions that mediate[s] lifespan extension".

We agree and longer make that statement. The sentence reads instead:

“Overall, our data suggest that the mechanisms underlying the preservation of body stiffness and cuticle integrity are distinctively regulated with age”. – Line 149

(10) Line 150: e1370 is not a knock-out. This allele is mis-represented in figure legends as "daf-2(0)". "daf-2(-)" would be more appropriate, or ideally just "daf-2(e1370)".

We apologise for this mistake. We thank the reviewer for pointing it out. We have corrected our nomenclature throughout the manuscript to adopt your suggestion.

(11) Line 162 and throughout: "homeostatic pressure" is used but "hydrostatic pressure" would be more appropriate. Potentially the pressure maintenance is homeostatic (i.e. it is maintained at a set point through feedback from baroreceptors or something), but that is a specific biological hypothesis. We agree. Instead we changed it to stiffness throughout the manuscript for simplicity and consistency.

(12) Line 238-241: "Comparison between physiologically matched *phm-2* mutants and wild-type worms at approximately their mean lifespan age (day 17 for wild-type and day 21 for *phm-2*) shows that reduction of nutrient intake driven by mechanical (Figure 3C) rather than neuronal mechanisms as in *eat-2* mutants (Figure 3D) preserves homeostatic pressure with age." This is a stretch at best. There are a lot of things different between *eat-2* and *phm-2* beyond the mechanical vs. neuronal difference in the DR. All that can reliably be concluded is that different mutants that both seem to extend lifespan via DR have different effects on the cuticle. This could be due to the specific mechanisms of the DR, or due to some other pleiotropies of the mutants.

We had previously decided to accurately describe the functional effects of these mutants but agree with the reviewer that there are a lot of pleiotropic effects in these mutants. We no longer make that statement. Instead we state in Lines 233-242:

"Moreover, comparison between physiologically matched *phm-2* or *eat-2* mutants and wild-type worms at approximately their mean lifespan age (day 17 for wild-type and day 23 for *phm-2* and *eat-2*) shows that *phm-2* maintained their stiffness significantly while *eat-2* mutants did not. Our data is consistent with previous observations where no healthspan gains with age have been reported for the *eat-2* mutant¹⁸. Overall, our data suggest that distinct DR interventions impact stiffness differently with age."

(13) Line 255: The authors (of all people) should know that "blue fluorescence accumulation" is not a particularly good measure of health (as compared to a measure of dying worms).

We agree that it is not a particularly good measure of health. Yet, it is still very commonly used in the ageing field up to this date. We decided to include it as our sentence is an accurate representation of the previous literature concerning this point (Onken et al., 2010; Plos One. PMID: 20090912). In addition, the work performed by Zachary Pincus in (Coburn et al, 2013 PloS Biology; PMID: 23935448) shows that blue fluorescence still steadily increases with age but not to the same extent as green or red fluorescence, and has a burst at the time of death.

(14) Line 282-283: "Finally, we also show that metformin slightly reduces cuticle senescence with ageing (Figure 3I)". What criteria are being applied here? It's hard to see much difference at all in the images shown.

We agree that the images at chronological time were not very informative. We have now performed additional analysis on the metformin data for cuticle senescence at mean lifespan and we use the quantification of roughness instead as suggested by the reviewer. Our data in Figure 3G, now show that metformin treatment significantly maintains a youthful state of the cuticle at mean lifespan (day 24) when compared to untreated worms, also at mean lifespan (day 18).

(15) Throughout, the authors persist in describing many treatments which have fairly broad effects on bacteria as "metabolic alterations". For example, line 290: "Notably, changes in bacterial metabolism induced by UV-irradiation, antibiotic treatment or host-targeted drugs like metformin extend *C. elegans* lifespan". It seems to be a bit of a stretch to pin the lifespan extension in all of these cases on

"changes in metabolism" -- as compared to, say, decreases in bacterial proliferation. At best, it could be a mix of both. Indeed, doesn't UV treatment primarily prevent replication while largely leaving metabolism intact?

We utilised those terms because our work on metformin and trimethoprim (Cabreiro et al., Cell 2013) shows that these two drugs impair 1-carbon metabolism, which is a central metabolic pathway in most organisms. Further, additional work from us currently under revision and related to metformin and bacteria (Pryor et al.), together with our work presented here show that these two treatments have broad metabolic perturbations in bacteria, beyond 1-carbon metabolism (Figures S5C, S5D, S5E). The work by Fiksdal and Tryland 1999, Journal of Applied Microbiology; PMID: 10432588) also shows that high U.V.- and heat-treatment in doses similar to the ones applied in our study lead to severe metabolic impairment of *E. coli*. Nevertheless, we have performed further metabolomics experiments and the new data show that all of these interventions (U.V.-treatment, heat-treatment, deletion of Krebs cycle gene *gluA* in *E. coli*, trimethoprim and altering the nutritional source – Bactotryptone vs Bactopeptone) with the exception of carbenicillin, dramatically impact the metabolome of *E. coli* cells (Figure S5D, S5E). Quite remarkably, the impact of U.V.-treatment on the bacterial metabolome is as drastic as the effect of the bacteriostatic antibiotic trimethoprim at concentrations below MIC, a classical example of an anti-metabolite drug. We have rephrased to “effects on bacterial physiology” to represent more accurately the literature and match our current data since bacterial physiology refers to all the life-supporting functions and processes of bacteria (e.g. metabolism, etc) which allow bacterial cells to grow and divide (e.g. proliferation).

Similar concerns apply to line 300, where the authors discuss "impaired bacterial metabolism using heat and UV-irradiation". It seems like a rather severe understatement to describe heat-inactivating bacteria (which does many things, including denaturing proteins and even permeabilizing cell walls and membranes) as a "metabolic impairment".

Thank you for highlighting this. We had misplaced this sentence “Unlike other treatments, heat-killing induces a cellular collapse in bacteria caused by the loss of proteins with key functions in maintaining cellular homeostasis⁵⁸,” at the end of the section citing a recent paper published in Science. We have now moved this sentence to describe here the effects of heat-treatment on bacteria. Furthermore, our new current metabolomics data illustrate that heat-treatment leads to a loss of 132 or 140 /228 of the measurable metabolites in *E. coli* (Figure S5F). This is consistent with heat-treatment impairing metabolism amongst many other processes.

(16) Line 308-311: "However, force-indentation curves and 450 nN force indentation values are significantly lower for wild-type worms fed UV-irradiated *E. coli* OP50 but not heat-killed bacteria (Figure 4B), suggesting that bacterial metabolism rather than bacterial proliferation regulates healthspan in worms ". I'm not sure how this follows.

The authors compare metabolically active, non-proliferating bacteria (UV), which have an effect, to metabolically inactive, non-proliferating bacteria (heat), which do not have an effect. So at best the authors can claim that, among their limited sample of treatments, lack of bacterial proliferation is necessary but not sufficient to improve cuticle stiffness. But even that is likely to be an over-generalization from insufficient evidence. For example, given that the two treatments extend lifespan by different degrees, it might also be that the authors are simply looking too late in life, at a point by which heat-killed bacteria no longer improve the cuticle, but UV treatment still does. (As with most everywhere else, the authors fail to state the day on which the measurements in Fig. 4B were made.)

We agree that in this instance in the paper there is no sufficient evidence to support the stated claim and we have removed it. We have performed additional longitudinal stiffness assays and made

measurements at several timepoints for both heat-treatment and U.V.-treatment experiments. These now show that while U.V.-treatment improves stiffness both chronologically and at mean lifespan, heat-treatment of bacteria does not have beneficial effects on stiffness at any age, including at mean lifespan (Figure 4B). In addition, we show that altering the nutritional protein source of the nematode growth media (Bactotryptone rather than Bactopeptone) alters bacterial growth (Figure S5A, S5B), respiration (Figure S5C) and metabolism (Figure S5D, S5E) and improves worm stiffness with age at day 12. Importantly these positive effects on worm stiffness are abrogated by previous heat-treatment of bacteria (Figure S6B).

In summary, we have left the general conclusions to the end of this section where the extensive combined additional evidence from other treatments and interventions supports our main claim that bacterial physiology is a main driver of stiffness maintenance with age. These include 2 nutritional conditions (Bactopeptone and Bactotryptone), 3 drugs (carbenicillin, metformin, trimethoprim), 2 disruptive treatments (heat- and U.V.-treatment) and 2 genetic deletions of metabolic genes in *E. coli*. We apologise for not being thoroughly diligent with stating the age at which each indentation curves were taken. This has now been properly addressed and thank the reviewer for highlighting it.

(17) Line 314-315: "carbenicillin, which kills bacteria by inhibiting bacterial cell wall synthesis". What does "kills" mean? Prevents replication? Prevents metabolic activity? The authors were previously careful about this distinction, so why not here?

We previously used the general definition adopted by scientists in the microbiology field. Here is an extract from (Pankey GA et al., 2004 Clinical Infectious Disease, PMID: 14999632) which has been cited 600 times.

"The definitions of "bacteriostatic" and "bactericidal" appear to be straightforward: "bacteriostatic" means that the agent prevents the growth of bacteria (i.e., it keeps them in the stationary phase of growth), and "bactericidal" means that it kills bacteria".

To be consistent with our description of the effects of heat- and U.V.-treatment on bacteria, and with our new data we now state in lines 329-335:

"Next, we investigated whether inhibition of proliferation alone benefits host stiffness with age by using two antibiotics with distinct modes of action. The bacteriostatic antibiotic trimethoprim leads to metabolically-impaired bacteria (Figures S5D-E) with decreased proliferative capacity (Figures S5A-B) by inhibiting 1-carbon cellular metabolism, with accumulation of dihydrofolate (Figures S5G-H). In contrast, the bactericidal antibiotic carbenicillin leads to inviable non-proliferative bacteria (Figures S5A-B) by inhibiting bacterial cell wall synthesis."

(18) Lines 320-321: "bacterial proliferation per se is not a driving factor for improved homeostatic pressure in aged worms". Again, this is not supported by the authors' data! Except in the case of metformin, all the stiffness-improving regimes involved bacteria that cannot proliferate. So the best the authors can claim is that some but not all conditions that render bacteria non-proliferative can stiffen the cuticle. I.e. that preventing proliferation is not sufficient -- but may be necessary -- for improving the cuticle.

The statement "Except in the case of metformin, all the stiffness-improving regimes involved bacteria that cannot proliferate" does not accurately represent our data.

We would like to bring the reviewer's attention to the following information:

- 1) Metformin attenuates the growth and respiration of proliferative bacteria (Figures S5A, S5B, S5C and Cabreiro et al., Cell 2013, PMID: 23540700) that remain metabolically active (Pryor et al, in 2nd revision).

- 2) Trimethoprim at concentrations below MIC attenuate growth and respiration of proliferative bacteria (Figures S5A, S5B, S5C), specifically disturb 1-carbon folate metabolism (Figures S5G, S5H) and have general knock-on effects on general metabolic profiles (Figures S5D, S5E).
- 3) Genetic deletions of the non-essential genes *gltA* and *sucA* impair bacterial growth and respiration of proliferative bacteria (Figures S5A, S5B, S5C), and OP50 *gltA* mutants have altered metabolic profiles compared to OP50 (Figures S5D, S5E).
- 4) Alteration of the nutrient protein source alters bacterial growth and respiration of proliferative bacteria (Figures S5A, S5B, S5C) and their metabolome (Figures S5D, S5E). Both protein nutrient sources, significantly altered worm stiffness (Figure S6A). These positive effects on stiffness with age of worms grown on Bactotryptone were abolished by previous heat-treatment of bacteria.

(21) Lines 342-343: "Overall, our data support the hypothesis that viable but metabolically altered bacteria can improve homeostatic maintenance with age." Again, this is not supported by the data, and is also terminologically confused. I assume here that "viable" means "metabolically active but not necessarily capable of further cell division"? But that's not totally clear.

For the reasons pointed out by the reviewer, in the current version of the manuscript we no longer make this statement.

As above, all the conditions that do stiffen the cuticle involve bacteria that are not actively proliferating. However, some conditions like heat and carbenicillin that stop proliferation fail to stiffen the cuticle. Absent clear mass-spec / Seahorse / whatever results about what's going on metabolically in UV- vs. carb- vs. TMP-treated bacteria, it's hard to claim that there's some set of metabolic alterations that is specifically responsible for the cuticle stiffening.

This has been addressed earlier in point (18). In addition, we have now performed bacterial growth experiments (Figure S5A, S5B), respiration experiments (Figure S5C) and metabolomics experiments (Figure S5D, S5E) as suggested by the reviewer to support our current claims.

Clearly not all treatments that prevent bacterial replication improve the cuticle. But saying that it is metabolic alterations specifically that cause cuticle improvement seems to be a significant overreach. What if carbenicillin and heat-killing just render the bacteria indigestible in some way that cancels out the health benefit of preventing proliferation? That possibility is just as consistent with the data as the authors' claim. So absent any real data about mechanism, the authors should really just stick to stating that different bacterial treatments have different effects on the cuticle.

We would like to thank the reviewer for this suggestion. We have now included additional data aiming at understanding whether digestibility is a major factor determining stiffness. Since DR induces the up-regulation of the gene *acs-2* (Burkewitz et al, 2015 Cell, PMID:25723162), we have utilised the transgenic reporter line *Pacs-2::GFP* to test whether the bacterial treatments lead to reduced digestibility (Figure S8E-H). As expected, both bacterial deprivation, and the genetic mutants *eat-2* and *phm-2*, which have reduced "digestibility", significantly increase the expression of *Pacs-2::GFP*. Similarly, metformin, a putative DR-mimetic also increased significantly *Pacs-2::GFP* expression. While "indigestibility" correlates with the observed lifespan effects (Figure S8F), indigestibility did not correlate with either stiffness or roughness. Instead a non-significant negative or positive correlation, respectively, exists between these two variables (Figure S8G-H). Strikingly, carbenicillin treatment did not alter "digestibility" while heat-treatment did despite similar effects in regulating both host stiffness and roughness.

(22) Lines 350-351: "Taken together, our data convincingly show that bacterial metabolism is a key determinant for preserving pressure homeostasis and therefore health with age". As above, this reviewer strongly disagrees about how convincing a case is made for metabolism *per se*.

Since the reviewer agrees that bacterial proliferation *per se* does not improve the cuticle "Clearly not all treatments that prevent bacterial replication improve the cuticle ", we have changed this concluding sentence to – Line 379-382:

“Altogether, our findings support the hypothesis that inhibiting bacterial proliferation *per se* is not a driving factor for improved stiffness and cuticle senescence in aged worms, but altered bacterial physiology resulting from changes in respiration and/or metabolism play a major role in regulating host fitness.”

We hope that with additional clarification in this rebuttal and further substantial evidence provided in the revision of our manuscript that the reviewer agrees with its current interpretation and claims.

Reviewer 2:

The paper by Essmann et al. investigates the mechanical properties of *C. elegans* by AFM and relates those measurements to the healthspan of the organisms. While the approach is highly interesting and timely and could be of interest to a wider audience, the overall presentation hampers the reviewer's enthusiasm somewhat.

First, the figures are too crowded and highly repetitive in their appearance. They should be streamlined and condensed to essential results while all other graphs should be moved to Supp Info.

We thank the reviewer for their constructive comment and have followed their suggestion to simplify the figures substantially. In the newly arranged main figures we only display one example of the original force indentation curves (Figure 1C), and from there on we display our data as material properties (Young's Modulus). This new presentation form has improved and streamlined the paper allowing for more robust comparisons between conditions and reflecting more accurately the individual variability in our measurements (rather than the previously used 450 nN force indentation depth). It has also importantly decluttered the manuscript and we hope this improved version of the manuscript will please the reviewer.

Since AFM is the central technique, it must be explained in much more detail. Although the authors refer to an publication about the method, it is essential for the understanding to mention more details. It would be good to show a typical raw force indentation plot in SI.

We agree with the reviewer. Following the reviewer's suggestions, we have amended the "AFM part" in the material and methods section to improve the description of the method and the amount of detail provided. We have also dedicated a supplementary figure (Figure S1) with new explanatory graphs and data to show how the data was acquired and processed.

In particular, it is not clear how exactly mechanical properties are analyzed and quantified other than the indentation at a particular force. Throughout the manuscript the term 'homeostatic pressure' is used many times, however the authors do not explain how they determine this 'pressure' from their experiments.

We have amended the material and methods section to be more precise in how the data was acquired and processed. We have also designed a supplementary figure (S1), which includes schemes and graphs to illustrate the data processing. We agree, that since the worm is a "liquid filled cylinder" (Elmi, M. et al., Sci rep 2017; PMID: 28951574), the internal pressure together with the cellular mass, especially the outermost layer (cuticle) acting as exoskeleton all together contribute to the stiffness of the worm. Therefore, we agree with the reviewer that it is more correct to say "stiffness" rather than homeostatic pressure and have amended it throughout the manuscript.

In general, a real quantification of the 'new biomarker' in terms of mechanical properties is needed that takes in particular the tip geometry into account so that other groups can also relate their measurements when not using the exact same measurement geometry.

We thank the reviewer for the comment. Following this advice and also as suggested by reviewer 1, we have now calculated the Young's Modulus from each one of the individual worm curves we acquired using the Hertz/Sneddon model. The Young's Modulus presents indeed a more universal measure considering the shape of both the indenter and the curve. We have included illustrations about the force curve analysis and calculations of the Young's Modulus in Figure S1B, and also

included a more detailed description of the tips and tip shapes used into the material and methods section.

Quantification of the topography images is also needed.

We agree with the reviewer and have quantitatively analysed all the image datasets at mean lifespan. We have introduced a roughness factor, according to manufacturer's instructions to better define cuticle health as described in detail in the material and methods section. These new data have provided a better analytical approach to our imaging of cuticle senescence in a fully quantitative manner and allowed us to assess differences between worm genotypes and effects of environmental conditions at mean lifespan.

Temperature: The authors state that 'temperature is a major factor ...'. What was the temperature for AFM measurements and would that affect mechanical properties?

Temperature is a deciding factor contributing the lifespan of a worm population as shown in the lifespan curve in figure 1E. All growth and maintenance temperatures are now more clearly specified in the material and methods section. Independent of the worm culture conditions, all experimental AFM measurements were carried out at room temperature, which is kept constant at 20C to avoid any confounding factors. We apologize if this was not made clear before. We have now added further information in the Material and methods section.

How exactly were the worms immobilized? Is this affecting the measurements?

This is an interesting point and we thank the reviewer for highlighting it. The worms were immobilised using 2,3-butanodione monoxime (BDM), a drug commonly used to paralyse worms for microscopy. All experiments were carried out using the same conditions as described in detail now in the material and methods. We also included a new data set (Figure S1E) where we took measurements pre and post BDM exposure and observe no significant changes in YM values.

MINOR ISSUES

the title should be reworded to include 'mechanical properties' or similar abstract: 'loss of biomechanical properties' makes no sense. The sample still exhibits mechanical properties, however e.g. the stiffness can be higher or lower, or mechanical integrity could be lost. Please reword. We thank the reviewer for the suggestion. The title of the manuscript is now changed to: Mechanical properties measured by Atomic Force Microscopy define new health biomarkers in ageing *C. elegans*. We amended the sentence in the abstract to "...measure the change in biomechanical properties associated with..." - Line 31

It is essential to mention AFM measurements and obtained mechanical properties in the abstract.

We agree. It is mentioned in the abstract on two occasions:

"Atomic Force Microscope (AFM) to quantitatively measure the change in biomechanical properties"
- Line 30

"AFM provides a highly sensitive technique to measure organismal biomechanical fitness" - Line 37

And we now also refer to it in the title: "Mechanical properties measured by Atomic Force Microscopy define new health biomarkers"

Why is the force in Fig 1 H only up to 400 nN when all other plots are up to 450 nN?

We apologise to the reviewer for not making this clear in the material and methods. This was down to a technical reason. The sensitivity of the particular cantilever used in this experiment meant that the measures had to be set to a maximum force of 400 nN to stay inside the detectable range of the photodiode. This figure has now changed as we are displaying all results as material properties and it is now Figure 1F. Also we now state this clearly in the material and methods section – Line 655.

In the Methods section AFM, at the end of the last sentence there seems the reference Essmann et al 2017 NBM was not correctly inserted.

Thank you for identifying this issue. It has been corrected.

1. 428 cells are in addition also under contractile stress from their own acto-myosin cytoskeleton and neighboring cells as well as ‘pressure’ due to confined space.

Thank you for the suggestion. This sentence has been rephrased

The section has been moved to the introduction part as suggested by reviewer 1. We have included contractile stress and confined space through neighbouring cells, it now read as: “Cells are under constant mechanical pressure from shear stress and contractile stress from their own acto-myosin cytoskeleton and from neighbouring cells, from pressure due to confined space, and through gravity or hydrostatic pressure”. Line 87-89.

Reviewers' comments:

Reviewer #1 (Remarks to the Author):

Let me preface this by stating unambiguously: I like this work. I think it is interesting, novel science that should be published. However, I also think it should only be published once the authors confine themselves to claims that are actually warranted by their data.

The authors have responded well to many of the previous concerns; however, certain concerns remain inadequately addressed and some of the responses raise new concerns. These are detailed below.

Overall, I do not feel like there is any need for additional experiments or controls here -- just a need for the authors to be more careful with their language and their claims.

REMAINING CONCERNS:

1) Moving to Young's Modulus measurements seems to have been an excellent choice: the figures are much easier to understand, and showing the scatterplots of all of the YM measurements per condition is a really excellent way to do the visualization.

I have one question: does each point in the YM scatterplots represent a distinct individual *C. elegans*, or one of multiple measurements made on a smaller number of individual worms? I ask because in the latter case where each worm may be represented by multiple points in a scatterplot, the IID assumptions of standard statistical tests would be violated and the resultant p-values (etc.) incorrect. If the n represents the number of worms, though, the authors are to be commended for the large number of individuals assayed! 215 day-18 wild-type individuals, as in Fig 1D, for example, is a significant feat. (The reporting summary document says that the authors confirm the existence of a statement on whether measures are from distinct samples, but I couldn't find this anywhere for the YM data. My apologies if I missed it.)

Beyond that question, I do have a remaining concern about the YM fits: to what degree does the Hertz model accurately fit the data? This is not shown, so it's not easy to evaluate whether *C. elegans* cuticles (within the force ranges probed) meet the idealized assumptions of that model. At a minimum, it would be good to see something in Fig S1 showing several actual force-indentation curves vs. the analytic curves that would be expected for an ideal Hertzian material with the same YM. This would be reassuring that the data are well modeled with the Hertz assumptions.

2) Data reproducibility.

Despite the authors extensive response-to-reviewer text, these issues are not well addressed in the response or (more important) in the manuscript itself.

Overall, the response-to-reviewers document mentions a reasonable number of important caveats that limit data reproducibility, particularly surrounding cantilever calibration. The response also makes the very valid point that such inter-replicate variability is acceptable in the case that (a) side-by-side experiments are always conducted between control and experimental groups, and (b) replicates of those experiments always show the same relative relationship between control and experiment, regardless of the absolute magnitude of the measurements.

Unfortunately, the authors don't spend any time demonstrating that condition (b) holds. If that were clearly demonstrated, and if the authors restricted their analysis to within-replicate comparisons (and showed that all such comparisons across replicates went in the same direction), this would be completely reasonable.

This is how the field deals with replicate-to-replicate variability in lifespan, for example: careful labs do multiple replicates of experiments with side-by-side wild-type controls. They present all replicates in the figures (ideally), or at least 1 replicate and state that the relative effects in the other replicates were comparable.

I was sad to see, however, that none of these reasonable and important caveats from the response were incorporated in the manuscript! Instead, the ms. simply declares "Importantly, YM measures were reproducible between both technical (Figure S1C) and biological replicates (Figure S1D)". Examination of Fig S1C and D seems to contradict that claim, however.

First off, Fig S1D could be more accurately read to show that YM measurements are NOT reproducible among biological replicates. S1D does nicely show that the relative ordering between experimental groups (e.g. day 1 vs. 12 vs. 18) is indeed reproducible across replicates. However, this was not the point the authors chose to make. Instead, they assert that the YM measurements themselves are reproducible between replicates. I'll go into more detail below, but overall Fig S1D shows that replicates differ from one another within a ~2-fold range. That's not a lot to write home about. (Also compare Figure 4G, where a day-14 wild-type sample has a mean YM of around 32 kPa. In S1D, the range of day-12 replicates is ~128-256 kPa, and day-18 replicates from 64-128 kPa. So clearly, there are cases where inter-replicate variability is actually worse than S1D would indicate.)

Second, if Fig S1C is the best case for technical replicates, things are equally bad. What I take from S1C is that even within technical replicates (where there should be no batch differences between peptone, or between cantilevers and calibration), it's still not really possible to localize the mean YM better than to a 2-fold range. (In this case, 128-256 kPa).

In replicate lifespan experiments (which are, as the authors point out, among the noisiest of biological assays), 25-30% differences are common between replicates. In comparison, Figure S1C and D show close to 100% differences among replicates! (I.e. 2-fold.) Overall, the authors' comparison with lifespan measurements doesn't really hold water: the inter-replicate variation here is on a totally different scale.

The authors try to minimize this by comparing replicates to the overall mean in Fig S1D, and come up with a worst-case figure of ~1.5-1.6 fold. If there's a 1.5-fold difference from a bad-case replicate to the mean (e.g. the middle of the range of replicates), then one can conclude that all replicates will lie within a band of roughly 3-fold in width (1.5-fold above and 1.5-fold below the mean). My eyeball estimate of a ~2-fold range is thus entirely consistent with the authors' numbers.

In the response text, the authors try to paint this range in the best-possible light in comparison to the 8.93-fold difference between the means of day-1 and day-18 YM values. (They also irrelevantly compare this to the 250-fold range of individual worms measured, but that's beside the point for a comparison of means across replicates.) But 8.93-fold is basically the LARGEST difference the authors have encountered. Many of the experiments presented in the paper show differences between control and experimental groups in the range of 2-3 fold. This is uncomfortably close to the differences the authors seem to observe between biological and experimental replicates.

Moreover, the authors misleadingly call this a "[maximum] quantifiable difference of 1.66 between experiments", when really the 1.66-fold value is between an experiment and a pooled mean. As above, this leads to something more like a 2-3-fold maximum difference between any two actual experiments. Again, compare the day-14 sample in Fig 4G with a mean YM of 32 to any of the other comparable day-15-to-18 samples in the paper (with YMs ranging from 64-128). The imprecision between replicates does seem to be on the order of 2-3 fold, not the 1.66-fold that the authors suggest. (And 1.66-fold is bad enough compared to the 1.3-fold differences that make lifespan assays problematic and difficult to replicate.)

The authors may protest that their statistics say that there is no significant difference between the technical replicates in S1C. Well, first, what were the actual p-values? Figure S1C is silent on that score. (Were they closer to 0.5 or to 0.06? Given how far apart the 95% CIs are in S1C, I would wager that most of those "ns" p-values are pretty close to 0.05. Even if they didn't cross the totemic $p=0.05$ threshold, a moderately low p-value is not exactly reassuring for technical replicates. For example, $p=0.1$ would mean that just 1 time in 10 would sampling errors lead to inter-replicate differences as large as observed. So either the authors got extremely unlucky in the data from S1C, or there are other systematic problems.)

Moreover, regardless of the exact p-value, it's an elementary error in statistics to infer "no meaningful difference" from " $p \geq 0.05$ ". Absent a careful power analysis, one can't conclude much at all from the failure to reject the null hypothesis in Fig S1C.

On that note, the less said about S1E the better. One cannot conclude with an $n=4$ (per condition) experiment (i.e. absurdly underpowered) that "no significant p-value" means "the conditions are indistinguishable".

Now, the authors make a good point that there is a lot of inter-individual variance within populations in both lifespans and in many health measures. It could be that the replicate-to-replicate variability is exactly what one would expect given the wide range of YM values in the population. But the authors haven't carefully made that case either.

To do so, the authors would first need to characterize the degree of error inherent in measuring YM. Fig S1A is wholly insufficient: in it, one worm is demonstrated to have fairly reproducible YM scores across both the "neck" and "hip" regions. Is this typical? The best among 100 worms tested? Population statistics would be valuable here, rather than an $n=1$ example.

Next, if the measurement error is very small in magnitude compared to the population spread (which I do assume will be the case), the authors would then need to show that the variance among technical replicates observed is no greater than what would be expected given the population variation. This would demonstrate that there are no unexpected batch effects in the measurements (or other conditions): a necessary condition for statistical tests to give meaningful/interpretable p-values.

Fig S1C is an OK start, but more examples of multiple sets of technical replicates, with a proper multi-class ANOVA F-statistic used to compare the overall degree of between-group to within-group variation (and p-values shown to generally be well away from 0.05) would be reassuring on this point.

Hopefully the authors have more such data on hand, beyond what's shown in Fig S1C. (Also, why is just day 9 data shown?) I would hate to be asking for more experiments at this point -- but I would also find it somewhat shocking if the authors haven't bothered to carefully characterize the technical reproducibility of their central assay until two rounds of revision in.

So, what is my advice? I propose that the authors have two avenues:

(A) Be honest in the manuscript about the reproducibility between and within replicates; demonstrate clearly that qualitative differences among groups hold across multiple biological

replicates; and be cautious about interpreting differences in the 2-3 fold range. (And/or show that any claimed 2-3 fold differences are indeed repeatable across biological replicates.)

(B) Perform the appropriate experiments and analyses (described above) to show that the wide inter-replicate differences (even among technical replicates) are strictly attributable to sampling effects. This would build confidence that the statistical tests employed will have the expected false-positive rates.

3) The statistical analyses of the YM data are somewhat wanting.

- There is no examination of whether the YM data are (approximately) normal or lognormal or other (or what it would mean for worm YMs to be lognormally distributed). Given how the data seem to cluster nicely on the log-base-2 plot, I suspect some kind of logarithmic distribution. I'm pretty sure that the statistics employed perform fine for lognormal data, but the authors should also verify that presumption if the YMs are in fact logarithmically distributed.

- Next, there are many cases where " $p > 0.05$ " is incorrectly interpreted to mean "the conditions are biologically indistinguishable". Almost every figure contains panels where the "not significant" p-values are interpreted in this way. The correct approach is to perform a power calculation and use that to make a claim along the lines of "based on the sample size and population variance, our result suggests that these two groups are unlikely to be more than X% different". Or, potentially better, the authors might calculate confidence intervals around the difference in the means of the two groups, which would both give an indication about the effect size (i.e. difference in means) and the precision with which that effect was measured.

- The ANOVA-with-interaction-terms is a good way to quantify the extent of genetic (or other) interactions, and I applaud the authors' use of these approaches. However, there are a few problems to address. First, the authors don't indicate (or interpret) the magnitude of the interaction effects, just whether it was statistically significant. But the magnitude matters a lot for understanding the biology.

In the same vein, the authors need to be more careful with their terminology about describing the interaction results. For example, the authors state on line 422: "Furthermore, we show that the earlier observed improvements on stiffness of worms with age through feeding on *B. subtilis* and *C. aquatica* diets were also dependent on the regulatory role of *daf-16* (Figure 5F)" From that phrase one might imagine that in a *daf-16*-null background, there is no longer an effect on YM from feeding *B. subtilis*. But this is not the case at all! All that a non-zero interaction term indicates is that there is some degree of non-additivity in the relationship between diet and *daf-16*-genotype. Going from "non-zero non-additivity" to "dependent on" is a pretty major terminological leap, especially without

ever specifying the magnitude of interaction effect. Elsewhere the authors use the term "modulated by", which is much better in cases where the non-additivity isn't complete (e.g. total suppression).

- The authors may want to note (somewhere in the ms., or just for their own reference) that a 1-way ANOVA between just two groups is mathematically identical to a t-test. Usually the term ANOVA is reserved for comparisons among 3+ groups (for asking whether any one group is different than the rest). So I was momentarily confused to see pairwise comparisons of multiple groups described as using 1-way ANOVA. (E.g. Fig. S1C, but also elsewhere.)

- Regarding the above points, the "statistics" section of reporting summary document is a bit odd. Why have the authors marked "n/a" for the requirement to provide the exact p-value, the test statistic, confidence intervals and so forth for hypothesis tests? The authors perform null hypothesis testing, so this requirement should be met. As described above, simple things like effect sizes and exact p-values are routinely omitted.

4) The bacterial physiology section is dramatically improved in the precision of language and the claims made. In particular, the authors set things up well when they describe several different interventions "all aiming at modulating distinct aspects of bacterial physiology such as proliferation, growth, respiration and metabolism."

However, the authors should make it more clear which interventions influence which of those aspects of physiology. Figure S5 is helpful, but I also suggest a summary table (supplemental or main-text) with columns specifying the degree to which (either qualitative or quantitative would be fine) each condition alters (for example): bacterial proliferation, growth, respiration, metabolism, worm YM early in life, and worm YM late in life.

(Also, while I understand the distinction between "growth" and "proliferation" -- i.e. adding biomass vs. dividing -- it's not clear if the authors have actually tested those aspects of physiology as distinct from one another. Not that it would be necessary to do so, but the authors should be clear about what properties they are actually comparing.)

Next, the language does get imprecise in places. What specifically distinguishes a "viable non-proliferative" state (after UV treatment) from an "inviable non-proliferative" state (after heat treatment)? My understanding is that "viable non-culturable" refers to bacteria with intact membranes that presumably could proliferate in the right conditions, even if we don't necessarily know what those conditions are. But that's not exactly the state that the UV-treated bacteria are in... Do the authors mean to say that the UV treated bacteria are still performing some aspects of metabolism / respiration while the heat-killed bacteria are not? The authors should use the same terms throughout (making sure to define them carefully) and speak precisely about the effects of each condition.

This gets back to my comment in the original review that the authors should clearly distinguish between "killed" bacteria that are replication incompetent but metabolically active, and "killed" bacteria that are metabolically inert as well. (I note with some amusement that in their response statement, the authors cite a purportedly "straightforward" definition of bactericidal vs. bacteriostatic from a paper which was actually written to debunk the whole notion of a clear distinction between those two terms! Moreover nothing in that work actually addresses the point that there may be different kinds of "killed" bacteria. A patient in the clinic might not care about the specific cell biology of killed bacteria -- only that they will never divide again for whatever reason. But apparently *C. elegans* do care, and so the authors should endeavor to describe those cell biological states clearly and concisely.)

Next, I am concerned about the respiration measurements in S5C. The measurements basically seem to track with the growth measurements from S5B. This not surprising, since both were 18-hour timecourses, and obviously one will get higher total respiration from strains that are doubling faster. It would be good to measure respiration separate from growth, perhaps by normalizing by final OD or the area under the OD curve or something? (The authors would also need to make sure that the non-replicating bacteria are present at a high enough concentration throughout the experiment to be in the linear range of the assay.) As is, I'm not sure what if anything can be concluded from the respiration data given how confounded it is by growth effects. The authors should fix this or remove it.

I am also puzzled by the "missing metabolites" analysis. From both the heatmap in Fig S5D and the raw data in Table S2, there are no clear "missing data" in the abundances of the 228 metabolites. (Nowhere are the units for these values described, by the way! So I have no idea what the numeric values actually mean, but none are empty / marked as "missing" in any way.) Possibly the authors chose some numeric threshold for the abundance values below which the metabolite would be considered "missing"? I can't seem to find a threshold that gives the stated number of missing values for all of the conditions, however. So what do the authors mean by "missing metabolites"? And what do the numerical values in Table S2 actually mean?

Moreover, the "missing metabolites" analysis in Fig S5E suggests that the heat-killed bacteria are vastly different than all other conditions. But from the heatmap, PCA plot, and data in Table S2, there appears to be only minimal differences, metabolically, between heat- and UV-killed bacteria. What's going on here?

More generally, I only partially agree with the authors's conclusion (line 381), which states: "Altogether, our findings support the hypothesis that inhibiting bacterial proliferation per se is not a driving factor for improved stiffness and cuticle senescence in aged worms but altered bacterial physiology resulting from changes in respiration and/or metabolism play a major role in regulating host fitness".

The first part seems basically fine: they have identified one condition (carbenicillin) that is replication incompetent but has a normal metabolic profile and also has normal cuticle aging. So I agree that it doesn't seem to be replication-incompetence per se that's driving the cuticle changes.

However, the second claim represents a basic lapse in logic. Yes, the treatments examined change both the worm cuticle and respiration and metabolism in bacteria. But that doesn't mean there's a direct causal relationship between the two. There are probably thousands of aspects of bacterial physiology that the authors could have assayed, all of which are likely altered by these conditions. (Say the authors did proteomics instead of metabolomics, and found, as one might expect, a difference in the proteomes of heat-killed vs. tryptone-fed bacteria. Would it then be valid to conclude that the driving factor in worm cuticle stiffness is the bacterial proteome?)

Absent additional cases like carbenicillin that allow a clear-cut ruling out of different hypotheses, at the end of the day the authors show a collection of bacterial conditions that all simultaneously change many aspects of bacterial physiology (including almost certainly many not assayed by the authors), and also change worm cuticle stiffness. As such, a more sound conclusion would be something along the lines of: "Altogether, our findings do not support the hypothesis that inhibiting bacterial proliferation per se is a driving factor for improved stiffness and cuticle senescence in aged worms. Alterations to other aspects of bacterial physiology, potentially including metabolism and/or respiration, appear to drive these changes in *C. elegans* cuticle stiffness and structure."

5) I don't think I would really call Pacs-2::GFP a measure of "bacterial indigestibility". It (may) be a reporter for "effective degree of dietary restriction". (I.e. metformin increases Pacs-2::GFP while probably not rendering bacteria substantially less "digestible", whatever that would mean exactly.) I think the results are interesting, but it's odd to call this an "indigestibility" index.

I think the authors may have misunderstood my point in the original review about "digestibility". The point that I was making there was essentially the same one made above: just because the authors measure some difference in bacterial physiology that correlates with a change in worm cuticles doesn't mean that those bacterial change CAUSED the cuticular change. There may be many unmeasured, confounding effects of interventions like heat-killing bacteria (such as the purported changes in "digestibility") that are very difficult to rule out.

I do apologize for sending the authors on a wild-goose-chase to rule out any differences in "digestibility". Nevertheless, though the authors' Pacs-2::GFP experiments (may) rule out changes in "digestibility" (at least insofar as those changes would mimic DR), the point still stands that it's logically unsound to say that "metabolism" is the driving factor when it could be any number of related, unmeasured, aspects of bacterial biology (proteome, say, or ribosome number, or pH or whatever).

6) The section on feeding of *B. subtilis* and *C. aquatica* has similarly shaky logic in its language and conclusions. Yes, the nutrition that *C. elegans* receive is very different under these different bacterial food sources. But much else is ALSO different: the bacteria provide different olfactory cues, they have different levels of pathogenicity to *C. elegans* compared to OP50, probably alter the media pH, and so forth. Claiming that the difference observed are due to "nutrition" per se (i.e. what molecules the worms take up from the foods) is wholly unjustified given the rest of the differences that go along with changing to a completely new food bacteria.

Overall, the authors present a collection of different conditions (genetic mutants, bacterial treatments, different bacteria) that alter worm cuticles to different degrees and potentially in different ways. That's fine and perfectly interesting. But the authors grasp at straws when they try to ascribe these changes in worm cuticles to specific causes on the basis of correlative evidence alone.

MINOR NOTES:

Lines 227-236: the phrasing here is very unclear / hard to follow. There's also a stray comma on line 227.

I'm not sure why the authors want to fight so hard to keep the point about metformin and blue fluorescence accumulation in. As the authors mention Zachary Pincus's work, they might note the follow-up (Aging 2016) which seems pretty unambiguous: in bulk culture, blue fluorescence really seems to reflect the fraction of the population currently undergoing "death fluorescence". The cited metformin results are from bulk experiments, so it is irrelevant that the original death fluorescence paper showed that it is also possible, in single-animal longitudinal experiments, to detect minuscule changes in blue autofluorescence related more to age than incipient death. The authors should not repeat claims that they know to be suspicious / spurious just because those claims are a "reflection of the previous literature." (Especially since there's plenty of other literature calling such claims into question -- including the death fluorescence paper by some of the authors themselves!)

Reviewer #2 (Remarks to the Author):

The authors addressed my concerns adequately and substantially improved the manuscript.

However, regarding the details on AFM data analysis, there is still some crucial information missing:

Which range was used to fit the Hertz/Sneddon model?

Is there a tendency of rising or decreasing Young's modulus when extending or shrinking that fit range? Also, please show a raw curve including the fit in Supp Info.

Please also state the fit equation in the Supp Info part for sake of completeness.

What were the variations of spring constants of the cantilever as determined by thermal noise? Please add the measured spring constants to Supp Info.

It would be good to combine all this information to a section in Supp Info, as AFM is the central technique of this paper.

Please find below our response to the Reviewer's comments. We have fully addressed their concerns with additional experimental data, statistical analysis and manuscript text changes so that our claims are supported solely by the experimental work produced.

In **black** the Reviewer's original and unmodified comments, in **blue** our response and in **green** extracts from the manuscript.

Reviewers' comments:

Reviewer #1 (Remarks to the Author):

Let me preface this by stating unambiguously: I like this work. I think it is interesting, novel science that should be published. However, I also think it should only be published once the authors confine themselves to claims that are actually warranted by their data.

The authors have responded well to many of the previous concerns; however, certain concerns remain inadequately addressed and some of the responses raise new concerns. These are detailed below.

Overall, I do not feel like there is any need for additional experiments or controls here -- just a need for the authors to be more careful with their language and their claims.

REMAINING CONCERNS:

1) Moving to Young's Modulus measurements seems to have been an excellent choice: the figures are much easier to understand, and showing the scatterplots of all of the YM measurements per condition is a really excellent way to do the visualization.

I have one question: does each point in the YM scatterplots represent a distinct individual *C. elegans*, or one of multiple measurements made on a smaller number of individual worms? I ask because in the latter case where each worm may be represented by multiple points in a scatterplot, the IID assumptions of standard statistical tests would be violated and the resultant p-values (etc.) incorrect. If the *n* represents the number of worms, though, the authors are to be commended for the large number of individuals assayed! 215 day-18 wild-type individuals, as in Fig 1D, for example, is a significant feat. (The reporting summary document says that the authors confirm the existence of a statement on whether measures are from distinct samples, but I couldn't find this anywhere for the YM data. My apologies if I missed it.)

Yes. Each point represents a single worm. This is now stated in the material and methods section under Worm preparation for Atomic Force Microscopy.

Beyond that question, I do have a remaining concern about the YM fits: to what degree does the Hertz model accurately fit the data? This is not shown, so it's not easy to evaluate whether *C. elegans* cuticles (within the force ranges probed) meet the idealized assumptions of that model. At a minimum, it would be good to see something in Fig S1 showing several actual force-indentation curves vs. the analytic curves that would be expected for an ideal Hertzian material with the same YM. This would be reassuring that the data are well modeled with the Hertz assumptions.

As suggested also by Reviewer 2, we have now included examples of the original JPK files created for the force-indentation curve measurements and the respective Hertz fit for worms at different ages.

2) Data reproducibility.

Despite the authors extensive response-to-reviewer text, these issues are not well addressed in the response or (more important) in the manuscript itself.

Overall, the response-to-reviewers document mentions a reasonable number of important caveats that limit data reproducibility, particularly surrounding cantilever calibration. The response also makes the very valid point that such inter-replicate variability is acceptable in the case that (a) side-by-side experiments are always conducted between control and experimental groups, and (b) replicates of those experiments always show the same relative relationship between control and experiment, regardless of the absolute magnitude of the measurements.

Unfortunately, the authors don't spend any time demonstrating that condition (b) holds. If that were clearly demonstrated, and if the authors restricted their analysis to within-replicate comparisons (and showed that all such comparisons across replicates went in the same direction), this would be completely reasonable.

This is how the field deals with replicate-to-replicate variability in lifespan, for example: careful labs do multiple replicates of experiments with side-by-side wild-type controls. They present all replicates in the figures (ideally), or at least 1 replicate and state that the relative effects in the other replicates were comparable.

I was sad to see, however, that none of these reasonable and important caveats from the response were incorporated in the manuscript! Instead, the ms. simply declares "Importantly, YM measures were reproducible between both technical (Figure S1C) and biological replicates (Figure S1D)". Examination of Fig S1C and D seems to contradict that claim, however.

First off, Fig S1D could be more accurately read to show that YM measurements are NOT reproducible among biological replicates. S1D does nicely show that the relative ordering between experimental groups (e.g. day 1 vs. 12 vs. 18) is indeed reproducible across replicates. However, this was not the point the authors chose to make. Instead, they assert that the YM measurements themselves are reproducible between replicates. I'll go into more detail below, but overall Fig S1D shows that replicates differ from one another within a ~2-fold range. That's not a lot to write home about. (Also compare Figure 4G, where a day-14 wild-type sample has a mean YM of around 32 kPa. In S1D, the range of day-12 replicates is ~128-256 kPa, and day-18 replicates from 64-128 kPa. So clearly, there are cases where inter-replicate variability is actually worse than S1D would indicate.

Second, if Fig S1C is the best case for technical replicates, things are equally bad. What I take from S1C is that even within technical replicates (where there should be no batch differences between peptone, or between cantilevers and calibration), it's still not really possible to localize the mean YM better than to a 2-fold range. (In this case, 128-256 kPa).

In replicate lifespan experiments (which are, as the authors point out, among the noisiest of biological assays), 25-30% differences are common between replicates. In comparison, Figure S1C and D show

close to 100% differences among replicates! (I.e. 2-fold.) Overall, the authors' comparison with lifespan measurements doesn't really hold water: the inter-replicate variation here is on a totally different scale.

The authors try to minimize this by comparing replicates to the overall mean in Fig S1D, and come up with a worst-case figure of ~1.5-1.6 fold. If there's a 1.5-fold difference from a bad-case replicate to the mean (e.g. the middle of the range of replicates), then one can conclude that all replicates will lie within a band of roughly 3-fold in width (1.5-fold above and 1.5-fold below the mean). My eyeball estimate of a ~2-fold range is thus entirely consistent with the authors' numbers.

In the response text, the authors try to paint this range in the best-possible light in comparison to the 8.93-fold difference between the means of day-1 and day-18 YM values. (They also irrelevantly compare this to the 250-fold range of individual worms measured, but that's beside the point for a comparison of means across replicates.) But 8.93-fold is basically the LARGEST difference the authors have encountered. Many of the experiments presented in the paper show differences between control and experimental groups in the range of 2-3 fold. This is uncomfortably close to the differences the authors seem to observe between biological and experimental replicates.

Moreover, the authors misleadingly call this a "[maximum] quantifiable difference of 1.66 between experiments", when really the 1.66-fold value is between an experiment and a pooled mean. As above, this leads to something more like a 2-3-fold maximum difference between any two actual experiments. Again, compare the day-14 sample in Fig 4G with a mean YM of 32 to any of the other comparable day-15-to-18 samples in the paper (with YMs ranging from 64-128). The imprecision between replicates does seem to be on the order of 2-3 fold, not the 1.66-fold that the authors suggest. (And 1.66-fold is bad enough compared to the 1.3-fold differences that make lifespan assays problematic and difficult to replicate.)

The authors may protest that their statistics say that there is no significant difference between the technical replicates in S1C. Well, first, what were the actual p-values? Figure S1C is silent on that score. (Were they closer to 0.5 or to 0.06? Given how far apart the 95% CIs are in S1C, I would wager that most of those "ns" p-values are pretty close to 0.05. Even if they didn't cross the totemic $p=0.05$ threshold, a moderately low p-value is not exactly reassuring for technical replicates. For example, $p=0.1$ would mean that just 1 time in 10 would sampling errors lead to inter-replicate differences as large as observed. So either the authors got extremely unlucky in the data from S1C, or there are other systematic problems.)

Moreover, regardless of the exact p-value, it's an elementary error in statistics to infer "no meaningful difference" from " $p \geq 0.05$ ". Absent a careful power analysis, one can't conclude much at all from the failure to reject the null hypothesis in Fig S1C.

On that note, the less said about S1E the better. One cannot conclude with an $n=4$ (per condition) experiment (i.e. absurdly underpowered) that "no significant p-value" means "the conditions are indistinguishable".

Now, the authors make a good point that there is a lot of inter-individual variance within populations in both lifespans and in many health measures. It could be that the replicate-to-replicate variability is exactly what one would expect given the wide range of YM values in the population. But the authors haven't carefully made that case either.

To do so, the authors would first need to characterize the degree of error inherent in measuring YM. Fig S1A is wholly insufficient: in it, one worm is demonstrated to have fairly reproducible YM scores across both the "neck" and "hip" regions. Is this typical? The best among 100 worms tested? Population statistics would be valuable here, rather than an n=1 example.

Next, if the measurement error is very small in magnitude compared to the population spread (which I do assume will be the case), the authors would then need to show that the variance among technical replicates observed is no greater than what would be expected given the population variation. This would demonstrate that there are no unexpected batch effects in the measurements (or other conditions): a necessary condition for statistical tests to give meaningful/interpretable p-values.

Fig S1C is an OK start, but more examples of multiple sets of technical replicates, with a proper multi-class ANOVA F-statistic used to compare the overall degree of between-group to within-group variation (and p-values shown to generally be well away from 0.05) would be reassuring on this point.

Hopefully the authors have more such data on hand, beyond what's shown in Fig S1C. (Also, why is just day 9 data shown?) I would hate to be asking for more experiments at this point -- but I would also find it somewhat shocking if the authors haven't bothered to carefully characterize the technical reproducibility of their central assay until two rounds of revision in.

So, what is my advice? I propose that the authors have two avenues:

(A) Be honest in the manuscript about the reproducibility between and within replicates; demonstrate clearly that qualitative differences among groups hold across multiple biological replicates; and be cautious about interpreting differences in the 2-3 fold range. (And/or show that any claimed 2-3 fold differences are indeed repeatable across biological replicates.)

(B) Perform the appropriate experiments and analyses (described above) to show that the wide inter-replicate differences (even among technical replicates) are strictly attributable to sampling effects. This would build confidence that the statistical tests employed will have the expected false-positive rates.

We have followed the Reviewer's suggestion A and B and have now included an additional table that shows all the individual biological replicates and the combined data and the respective statistical analysis and comparative effect size analysis (Table S1). We performed additional measurements for datasets where effect sizes were below 3-fold (Figure 1E, 2E, 3H, S2C, S4D and S6A) and for datasets that required additional measurements to confirm the robustness of our observations (3D, 4D, 4G, S1F, S2C and S6E). We have adapted our description of the data in the manuscript accordingly.

In addition, we have replaced old figure S1D with current S1C to highlight the variability between measurements across WT ageing cohorts measured on the same days of age. We have also, increased the numbers for S1A and S1E as suggested by the Reviewer, so that our original points are better supported.

For S1C (now S1D)- This experiment had been included to address a specific suggestion by the Reviewer in their first review. We have now also included in this graph additional data aimed at probing technical replicability. We would like to highlight that one variable was changed when we

performed these measurements which may explain the variable but non-significant difference observed between measurements obtained at day 9. Each plate was seeded from a culture of OP50 obtained by growing a distinct individual bacterial colony from the same LB plate. We routinely do this in our lab if running technical replicates, which is a standard microbiology technique performed to account for differences in the physiology of different bacterial colonies obtained from an isogenic population of bacteria. This is now stated in the figure legend S1D. “Variability within technical replicates for AFM measurements. Mechanical properties as YM (kPa) from 1-day or 9-days old WT *C. elegans* from different plates grown in parallel under same media and temperature conditions seeded with the culture of the same bacterial colony (A1-3) or seeded with a culture obtained from growing different individual bacterial colonies (A, B, C) from an isogenic OP50 population streaked on an LB plate. Error bars indicate 95% CI. One-way ANOVA Tukey’s multiple comparison test for statistical comparison between the replicates of day 1 or day 9.”

Overall, no significant differences are observed between technical replicates. Apart from S1D, we do not have technical replicates in our study.

We would like to bring the attention of the Reviewer to the comparison between WT (n= 544) and *daf-16(mgDf50)* (n=530). This comparison was performed 24 times in independent biological replicates for distinct datasets throughout the manuscript and at different points in time over the length of this study. These comparisons were performed in exactly identical experimental conditions (data in figure S2C was not included as the protocol used was not identical). The effect sizes between their comparisons and p values for all other datasets are:

Effect size (FC)	p-value t-test	Effect size (FC)	p-value t-test	Effect size (FC)	p-value t-test
1.18	0.9489	1.2	0.2495	0.8	0.2769
1.28	0.7742	1.03	0.7784	1.23	0.4114
0.59	0.3387	1	0.9999	1.05	0.8203
1.48	0.7074	1.36	0.1436	0.78	0.892
1.18	0.9606	1.22	0.2326	1.06	0.9997
0.68	0.1094	0.96	0.8751	0.77	0.9089
0.88	0.8126	0.93	0.7158	0.95	0.9998
0.84	0.3396	0.94	0.7188	1.11	0.9996

We have now changed the text to state the following which acknowledges the limitations of our technique: “YM measures were reproducible between technical replicates from young and aged worms with an effect size of 1.00 -1.47 fold (Figure S1D, Table S1). Based on the sample size and population variance, our result suggests that the groups of young 1-day old worms are unlikely to be more than 2.6-20.1% different, and the groups of aged 9-days old worms unlikely to be more than 3-68% different (Post-hoc power analysis). Despite variability between biological replicates, similar trends in these measurements were observed in ageing cohorts (Figure S1C, Table S1). In addition, these measurements were not affected by the compound 2,3-butanedione monoxime BDM (1.03 fold, Post-hoc power = 2.9%, Figure S1E, Table S1) used to paralyse the worms.”

Further we have replaced the sentence in the discussion: “These stiffness measurements provide a quick, reproducible and a quantitative approach for measuring a common observable feature when handling aged worms” to “These stiffness measurements provide a quantitative approach for measuring a common observable feature when handling aged worms”

3) The statistical analyses of the YM data are somewhat wanting.

- There is no examination of whether the YM data are (approximately) normal or lognormal or other (or what it would mean for worm YMs to be lognormally distributed). Given how the data seem to cluster nicely on the log-base-2 plot, I suspect some kind of logarithmic distribution. I'm pretty sure that the statistics employed perform fine for lognormal data, but the authors should also verify that presumption if the YMs are in fact logarithmically distributed.

As suggested by the reviewer we have now performed statistical analysis on log transformed data for two datasets: Figure 2C and 2E. The analysis shows that the tests perform comparably well. This comparative analysis can be found in Table S1.

- Next, there are many cases where " $p > 0.05$ " is incorrectly interpreted to mean "the conditions are biologically indistinguishable". Almost every figure contains panels where the "not significant" p-values are interpreted in this way. The correct approach is to perform a power calculation and use that to make a claim along the lines of "based on the sample size and population variance, our result suggests that these two groups are unlikely to be more than X% different". Or, potentially better, the authors might calculate confidence intervals around the difference in the means of the two groups, which would both give an indication about the effect size (i.e. difference in means) and the precision with which that effect was measured.

We have now included the numerical p-values for each measurement both in all figures and tables. We have measured effect sizes for all the comparisons made in the manuscript (including when evaluating the interaction of terms) and measured the post-hoc power for each non-significant comparison as requested. Additional information was also added to the Material and methods section. We have also adopted the statement suggested by the reviewer to introduce our post-hoc power analysis for non-statistically significant comparisons.

- The ANOVA-with-interaction-terms is a good way to quantify the extent of genetic (or other) interactions, and I applaud the authors' use of these approaches. However, there are a few problems to address. First, the authors don't indicate (or interpret) the magnitude of the interaction effects, just whether it was statistically significant. But the magnitude matters a lot for understanding the biology.

In the same vein, the authors need to be more careful with their terminology about describing the interaction results. For example, the authors state on line 422: "Furthermore, we show that the earlier observed improvements on stiffness of worms with age through feeding on *B. subtilis* and *C. aquatica* diets were also dependent on the regulatory role of *daf-16* (Figure 5F)" From that phrase one might imagine that in a *daf-16*-null background, there is no longer an effect on YM from feeding *B. subtilis*. But this is not the case at all! All that a non-zero interaction term indicates is that there is some degree of non-additivity in the relationship between diet and *daf-16*-genotype. Going from "non-zero non-additivity" to "dependent on" is a pretty major terminological leap, especially without ever specifying the magnitude of interaction effect. Elsewhere the authors use the term "modulated by", which is much better in cases where the non-additivity isn't complete (e.g. total suppression).

- The authors may want to note (somewhere in the ms., or just for their own reference) that a 1-way ANOVA between just two groups is mathematically identical to a t-test. Usually the term ANOVA is reserved for comparisons among 3+ groups (for asking whether any one group is different than the

rest). So I was momentarily confused to see pairwise comparisons of multiple groups described as using 1-way ANOVA. (E.g. Fig. S1C, but also elsewhere.)

We have corrected and changed the description of each one of the statistical tests used to analyse the data for each figure, including the correction method used, as recommended. We included additional statistical tests in Table S1, that were utilised in the previous version of the manuscript. As suggested, when appropriate, we have adopted the term modulated by, to describe more accurately the interaction of terms and included the magnitude of effect sizes between interactions and the contribution of the main factor in the interaction.

- Regarding the above points, the "statistics" section of reporting summary document is a bit odd. Why have the authors marked "n/a" for the requirement to provide the exact p-value, the test statistic, confidence intervals and so forth for hypothesis tests? The authors perform null hypothesis testing, so this requirement should be met. As described above, simple things like effect sizes and exact p-values are routinely omitted.

We thank the Reviewer for pointing this out. We have corrected the reporting summary document as highlighted by the Reviewer.

4) The bacterial physiology section is dramatically improved in the precision of language and the claims made. In particular, the authors set things up well when they describe several different interventions "all aiming at modulating distinct aspects of bacterial physiology such as proliferation, growth, respiration and metabolism."

However, the authors should make it more clear which interventions influence which of those aspects of physiology. Figure S5 is helpful, but I also suggest a summary table (supplemental or main-text) with columns specifying the degree to which (either qualitative or quantitative would be fine) each condition alters (for example): bacterial proliferation, growth, respiration, metabolism, worm YM early in life, and worm YM late in life.

Response: As suggested by the Reviewer we have now included a summary Table 1 which includes quantitative and qualitative information on bacterial growth, proliferation, metabolism and the respective effects of these treatments on worm's stiffness both at chronological age and mean lifespan.

(Also, while I understand the distinction between "growth" and "proliferation" -- i.e. adding biomass vs. dividing -- it's not clear if the authors have actually tested those aspects of physiology as distinct from one another. Not that it would be necessary to do so, but the authors should be clear about what properties they are actually comparing.)

As suggested, we have now evaluated the effects of our treatments on proliferation by measuring the growth rate during exponential phase. These data can now be found in Table 1 and the analytical description in the Material and methods section.

Next, the language does get imprecise in places. What specifically distinguishes a "viable non-proliferative" state (after UV treatment) from an "inviable non-proliferative" state (after heat treatment)? My understanding is that "viable non-culturable" refers to bacteria with intact membranes that presumably could proliferate in the right conditions, even if we don't necessarily know what those conditions are. But that's not exactly the state that the UV-treated bacteria are in... Do the authors

mean to say that the UV treated bacteria are still performing some aspects of metabolism / respiration while the heat-killed bacteria are not? The authors should use the same terms throughout (making sure to define them carefully) and speak precisely about the effects of each condition.

This gets back to my comment in the original review that the authors should clearly distinguish between "killed" bacteria that are replication incompetent but metabolically active, and "killed" bacteria that are metabolically inert as well. (I note with some amusement that in their response statement, the authors cite a purportedly "straightforward" definition of bactericidal vs. bacteriostatic from a paper which was actually written to debunk the whole notion of a clear distinction between those two terms! Moreover nothing in that work actually addresses the point that there may be different kinds of "killed" bacteria. A patient in the clinic might not care about the specific cell biology of killed bacteria -- only that they will never divide again for whatever reason. But apparently *C. elegans* do care, and so the authors should endeavor to describe those cell biological states clearly and concisely.)

Yes. We intended to use the term viable when referring to the remaining metabolic activity that can be measured in *E. coli* under U.V. but not heat treatment. For clarity we have now removed the use of the terms viable/inviable, killed and also culturable/non-culturable since we have not performed viability measurements in our study or performed additional measurements to evaluate types of cellular death in *E. coli*. We now refrain our description solely to parameters that have been measured in the context of each intervention.

Next, I am concerned about the respiration measurements in S5C. The measurements basically seem to track with the growth measurements from S5B. This not surprising, since both were 18-hour timecourses, and obviously one will get higher total respiration from strains that are doubling faster. It would be good to measure respiration separate from growth, perhaps by normalizing by final OD or the area under the OD curve or something? (The authors would also need to make sure that the non-replicating bacteria are present at a high enough concentration throughout the experiment to be in the linear range of the assay.) As is, I'm not sure what if anything can be concluded from the respiration data given how confounded it is by growth effects. The authors should fix this or remove it.

We agree that untangling these from bacterial growth would be difficult and beyond the scope of this paper, we have therefore followed the Reviewer's suggestion and removed it.

I am also puzzled by the "missing metabolites" analysis. From both the heatmap in Fig S5D and the raw data in Table S2, there are no clear "missing data" in the abundances of the 228 metabolites. (Nowhere are the units for these values described, by the way! So I have no idea what the numeric values actually mean, but none are empty / marked as "missing" in any way.) Possibly the authors chose some numeric threshold for the abundance values below which the metabolite would be considered "missing"? I can't seem to find a threshold that gives the stated number of missing values for all of the conditions, however. So what do the authors mean by "missing metabolites"? And what do the numerical values in Table S2 actually mean?

Moreover, the "missing metabolites" analysis in Fig S5E suggests that the heat-killed bacteria are vastly different than all other conditions. But from the heatmap, PCA plot, and data in Table S2, there appears to be only minimal differences, metabolically, between heat- and UV-killed bacteria. What's going on here?

Untargeted MS-based datasets often contain missing values that can arise from different technical issues. For example, these missing values could be metabolites not detected because they are not present, or just because their concentrations are below the detection limits of the machine. Missing value imputation is an important step in the data analysis of every metabolomics dataset, and according to the bibliography there are several methods to do so: substituting missing values by zero; by the mean; by half of the minimum value; random forest algorithm or SVD (singular value decomposition) among other (PMID: 19429898, PMID: 22039212; PMID: 24957035; PMID: 24039616 PMID: 17344241).

In our case, as we were dealing with some conditions where we had a large amount of missing values (Heat-treated bacteria), we chose to work with a modification of the 'substitution by zero' method, substituting every non-detected value with a very small value of $2E-52$ (as recommended by HMT-the metabolomics company). There are two main reasons for this choice: 1) the fact that if we are not detecting a metabolite it does not mean that the metabolite is not present in the sample; 2) We needed a complete matrix to calculate the PCA and for the sake of visualisation. However, we have tested additional imputation methods to our substitution by an epsilon value of $2E-52$. See below from left to right the analysis of our data using: Substitution by half of absolute minimum of the dataset, Substitution by half of the minimum score by metabolite, Mean imputation, Random Forest, Singular value decomposition (SVD) imputation) to show that the interpretation of our results remains the same independently of the analytical strategy utilised. The departure point used for every test is the same and the matrix of the relative peaks score that was also used originally for the PCA and heatmap.

The data is shown as Relative Peak Area and this information was previously stated in the material and methods and Table S2 (now Table S3). We have now included in Table S3, in addition to the relative peak area data, the quantitative estimation concentration (pmol/O.D.mL) for each metabolite per sample, and further described in detail in the material and methods our analytical approach.

More generally, I only partially agree with the authors's conclusion (line 381), which states: "Altogether, our findings support the hypothesis that inhibiting bacterial proliferation per se is not a driving factor for improved stiffness and cuticle senescence in aged worms but altered bacterial physiology resulting from changes in respiration and/or metabolism play a major role in regulating host fitness".

The first part seems basically fine: they have identified one condition (carbenicillin) that is replication incompetent but has a normal metabolic profile and also has normal cuticle aging. So I agree that it doesn't seem to be replication-incompetence per se that's driving the cuticle changes.

However, the second claim represents a basic lapse in logic. Yes, the treatments examined change both the worm cuticle and respiration and metabolism in bacteria. But that doesn't mean there's a

direct causal relationship between the two. There are probably thousands of aspects of bacterial physiology that the authors could have assayed, all of which are likely altered by these conditions. (Say the authors did proteomics instead of metabolomics, and found, as one might expect, a difference in the proteomes of heat-killed vs. tryptone-fed bacteria. Would it then be valid to conclude that the driving factor in worm cuticle stiffness is the bacterial proteome?)

Absent additional cases like carbenicillin that allow a clear-cut ruling out of different hypotheses, at the end of the day the authors show a collection of bacterial conditions that all simultaneously change many aspects of bacterial physiology (including almost certainly many not assayed by the authors), and also change worm cuticle stiffness. As such, a more sound conclusion would be something along the lines of: "Altogether, our findings do not support the hypothesis that inhibiting bacterial proliferation per se is a driving factor for improved stiffness and cuticle senescence in aged worms. Alterations to other aspects of bacterial physiology, potentially including metabolism and/or respiration, appear to drive these changes in *C. elegans* cuticle stiffness and structure."

We thank the Reviewer for the suggestion and have now included their conclusion instead.

5) I don't think I would really call *Pacs-2::GFP* a measure of "bacterial indigestibility". It (may) be a reporter for "effective degree of dietary restriction". (I.e. metformin increases *Pacs-2::GFP* while probably not rendering bacteria substantially less "digestible", whatever that would mean exactly.) I think the results are interesting, but it's odd to call this an "indigestibility" index.

I think the authors may have misunderstood my point in the original review about "digestibility". The point that I was making there was essentially the same one made above: just because the authors measure some difference in bacterial physiology that correlates with a change in worm cuticles doesn't mean that those bacterial change CAUSED the cuticular change. There may be many unmeasured, confounding effects of interventions like heat-killing bacteria (such as the purported changes in "digestibility") that are very difficult to rule out.

I do apologize for sending the authors on a wild-goose-chase to rule out any differences in "digestibility". Nevertheless, though the authors' *Pacs-2::GFP* experiments (may) rule out changes in "digestibility" (at least insofar as those changes would mimic DR), the point still stands that it's logically unsound to say that "metabolism" is the driving factor when it could be any number of related, unmeasured, aspects of bacterial biology (proteome, say, or ribosome number, or pH or whatever).

Since the reviewer thinks these data are interesting we kept it in the manuscript. We now refer to it as suggested – a reporter for effective degree of dietary restriction. We have also changed the conclusion to state the following instead: Generally, this suggests that the DR-like physiological response measured by *Pacs-2::GFP* and induced in worms by genetic mutations, metformin, and interventions that alter bacterial physiology, may not be the driving mechanism regulating these two health biomarkers.

6) The section on feeding of *B. subtilis* and *C. aquatica* has similarly shaky logic in its language and conclusions. Yes, the nutrition that *C. elegans* receive is very different under these different bacterial food sources. But much else is ALSO different: the bacteria provide different olfactory cues, they have different levels of pathogenicity to *C. elegans* compared to OP50, probably alter the media pH, and so forth. Claiming that the difference observed are due to "nutrition" per se (i.e. what molecules

the worms take up from the foods) is wholly unjustified given the rest of the differences that go along with changing to a completely new food bacteria.

We agree and changed the word nutrition for bacterial diets, which is a more encompassing term. We have also changed the conclusion accordingly:

Overall, our AFM method captures the complex bacteria diet-evoked effects on host health parameters such as stiffness and cuticle roughness and the associated host transcription factors modulating these effects.

We have also deleted the following sentences:

Regional variation in health is mostly governed by lifestyle choices rather than host genetics. Nutrition, a key environmental factor, shapes the relationship between host genetics and physiological outputs, ultimately determining health over the course of life.

Overall, the authors present a collection of different conditions (genetic mutants, bacterial treatments, different bacteria) that alter worm cuticles to different degrees and potentially in different ways. That's fine and perfectly interesting. But the authors grasp at straws when they try to ascribe these changes in worm cuticles to specific causes on the basis of correlative evidence alone.

MINOR

NOTES:

Lines 227-236: the phrasing here is very unclear / hard to follow. There's also a stray comma on line 227.

This has been corrected. The sentences now read: We observed that the decline in stiffness over time is different between the diverse DR interventions tested (Figures 3B, S3H, Table S1). Interestingly, *phm-2* mutants that have a similar lifespan extension to that of *eat-2* mutants (27.9% and 32.63% increase in mean lifespan, respectively compared to wild-type; log rank $p < 0.001$, Table S2) maintained their stiffness with age (day 23 vs day 1 = 0.85 fold, Post-hoc power = 16.6%), unlike both WT (0.0093 fold) and *eat-2* mutants (0.189 fold, Figure 3B, Table S1). Comparison between physiologically matched *phm-2* or *eat-2* mutants and wild-type worms at approximately their mean lifespan age (day 17 for wild-type and day 23 for *phm-2* and *eat-2*) shows that both mutants have higher stiffness than WT, but a greater magnitude of effect was observed for the *phm-2* mutant (*phm-2* vs WT = 4.86 fold; *eat-2* vs WT = 1.64 fold, Figure 3B, Table S1).

I'm not sure why the authors want to fight so hard to keep the point about metformin and blue fluorescence accumulation in. As the authors mention Zachary Pincus's work, they might note the follow-up (Aging 2016) which seems pretty unambiguous: in bulk culture, blue fluorescence really seems to reflect the fraction of the population currently undergoing "death fluorescence". The cited metformin results are from bulk experiments, so it is irrelevant that the original death fluorescence paper showed that it is also possible, in single-animal longitudinal experiments, to detect minuscule changes in blue autofluorescence related more to age than incipient death. The authors should not repeat claims that they know to be suspicious / spurious just because those claims are a "reflection of the previous literature." (Especially since there's plenty of other literature calling such claims into question -- including the death fluorescence paper by some of the authors themselves!)

This has been removed as suggested.

Reviewer #2 (Remarks to the Author):

The authors addressed my concerns adequately and substantially improved the manuscript.

However, regarding the details on AFM data analysis, there is still some crucial information missing:

Which range was used to fit the Hertz/Sneddon model?

We fitted the entire curve including the full baseline. The full fit is pre-set as the general setting within the JPK analysis software. Fitting only parts of the curve requires an extra step within the analysis software, which we have not applied as we are interested in the characteristics of the curve over the full indentation range.

Is there a tendency of rising or decreasing Young's modulus when extending or shrinking that fit range?

Also, please show a raw curve including the fit in Supp Info.

Please also state the fit equation in the Supp Info part for sake of completeness.

When decreasing the fit range (right image) the Hertz fit curve (green) no longer overlaps to the full curve and the YM changes in this case from 536.4 kPa to 424.7 kPa.

As suggested by the Reviewer we have now included the fit equation together with examples of the Hertz fit to individual force curves acquired from worms at different ages into Figure S1B.

What were the variations of spring constants of the cantilever as determined by thermal noise? Please add the measured spring constants to Supp Info.

The range of spring constants determined by thermal noise have now been added to the material and methods section. $k = 5.79\text{-}10.81$ N/m stiffness (NSC12 7.5 N/m μ Masch produced by sQUBE www.sQUBE.de)

It would be good to combine all this information to a section in Supp Info, as AFM is the central technique of this paper.

As suggested by the Reviewer, this information is now included into Figure S1B and in material and methods section of the manuscript.

Additional changes to figures and/or text

We removed the force-indentation curve panels from Figure 1 (Fig 1C) and Figure S1 (Fig S1B, S1C) as they have become redundant in the context of the current presentation of the data. For simplicity, all AFM force measurements are now presented as Young's Modulus. These adjustments have also been made to the manuscript text within the material and methods section under "AFM data analysis", where we removed the following sentences: "To display a mean force indentation curve (Figure 1D) the processed indentation curves were saved in text format and further analysed using a custom made Matlab script kindly provided by Haoyun Zhan. The script removes the baseline (values below 0 force), plots all curves of one condition (Figure S1B), and calculates the mean +/- S.E.M of these curves". Under the code availability section the following sentence was removed: "The Matlab script used to generate Figures 1C, S1B and S1C will be made available from the corresponding authors upon request."

We have removed data at day 14 in Figure S3H according to the "guidelines" suggested by Reviewer 1. The difference between treatment and control was below < 3-fold, the number of animals was on the low end, and it had been replicated only once.

As suggested by Reviewer 1, we have removed the data (Figure S5C) and the following text in the material and methods: "For respiration measurements, a similar protocol was used as for bacterial growth assays and in addition we used the Redox tetrazolium dye A from Biolog inc. at a final concentration of 1X per well. Plates were maintained at 37°C with constant shaking at 180 rpm. An end point assay was performed with a measurement at time 0 and 18 hours at OD 750 nm using a Tecan Infinite M2000 microplate reader operated via Magellan V6.5 software (Tecan)."

All panels in Figure S8 have been re-analysed to accommodate the new AFM datasets. No changes to the interpretation of the data resulted from the inclusion of these additional datasets.

Reviewers' comments:

Reviewer #1 (Remarks to the Author):

Overall impression: the claims made in the manuscript are now supported by the data shown. The authors have, commendably, bolstered the evidence shown for the repeatability of the YM measurements, and have not over-interpreted results with fold-changes smaller than the inter-replicate repeatability.

Minor comments:

(1) The presentation of power analysis, fold-change values, and p-values is a bit scattershot / perplexing. The p-values are shown in the figures, while the fold-change values are shown in the text (but not figure legends). The power calculations are shown in percent, which is a bit odd when the effect sizes are shown in fold-change. Nowhere are the proper interpretation of the power values, or the effect sizes for interaction terms really laid out clearly. Overall, this makes the statistical arguments much harder to follow.

First off, the authors need only invoke the power analyses in cases where they aim to interpret "change is not statistically significant" to mean "change is of small magnitude". In these cases, one might write something like "in condition X, YM was increased 6-fold compared to WT. In condition Y, the increase in YM was not statistically significant (fold-change=1.3, $p=0.14$). Post-hoc analysis suggests that this experiment was powered to detect a fold-change ≥ 2.4 (or ≤ 0.42), indicating that the true effect of condition Y is likely under 2.4-fold, which is well less than the effect of condition X".

This seems wordy (and would be needed in full detail only once), but anyway there really are very few cases remaining where the authors interpret non-significant p-values in this way. Moreover, this full logic is probably not really necessary in cases where condition X has a significant fold-change in one direction, but condition Y has a non-significant fold change in the opposite direction. In that case, one doesn't really need the power analysis to make the case that X and Y are qualitatively different.

The authors should be commended for adding additional statistical analysis, but they should not just "sprinkle the statistics on at the end" -- the statistical analysis ought to be clearly integrated into the interpretation of the results.

(2) One example of unclearly explained effect size for interaction term can be found on line 275: "the interaction of terms: metformin and bacterial strain was also significant (effect size of OP50-MR =

7.57, Figure 3D, Table S1)." What are the units? Is this effect size large or small compared to the non-interaction terms? The authors should seek out other similar cases and fix them.

(3) The entries for the columns in Table 1 aren't explained at all. It could definitely use an expanded legend.

(4) Overall the authors' distinction between bacterial "growth" vs. "proliferation" is super-unclear, and never really explained in clear language anywhere. The authors seem to be using "growth" to mean "increase in number of particles over a defined time" by measuring the area under the OD600 curve. In contrast, "proliferation" is used to mean "maximum rate of increase in number of particles based on the OD600 curve". I'm not sure this is really standard terminology, nor have the authors explained the interpretation of changes to one vs. the other value.

(5) I'm a bit confused about the claim on lines 438-442: "We found that the effects on lifespan induced by a *B. subtilis* diet were mediated by *daf-16* (CPH<0.001 for the interaction of terms: bacterial diet and host genotype, Figure 5A, 5E, Table S2), however, the effects mediated by a *C. aquatica* diet were not (CPH=0.1661 for the interaction of terms: bacterial diet and genotype Figure 5A, 5E, Table S2)." But visual inspection of the figures 5A and E tell a somewhat different story: in WT, *subtilis* extends lifespan and *aquatica* shortens it, compared to OP50. In *daf-16(-)*, all three bacteria yield very similar median lifespans, though *subtilis* and *aquatica* each show much more squared-off lifespan curves. So it certainly seems like eating *aquatica* does something different to *daf-16(-)* worms compared to wild-type. This does suggest some kind of interaction with *daf-16*, even if the CPH model doesn't capture it. (Does that model properly deal with survival curves that cross? I don't believe so...)

Reviewer #2 (Remarks to the Author):

The authors have addressed all remaining concerns and added necessary Supplemental Material.

Reviewers' comments:

Overall, we thank the reviewers for their time and dedication to importantly improving this manuscript.

Reviewer #1 (Remarks to the Author):

Overall impression: the claims made in the manuscript are now supported by the data shown. The authors have, commendably, bolstered the evidence shown for the repeatability of the YM measurements, and have not over-interpreted results with fold-changes smaller than the inter-replicate repeatability.

Minor comments:

(1) The presentation of power analysis, fold-change values, and p-values is a bit scattershot / perplexing. The p-values are shown in the figures, while the fold-change values are shown in the text (but not figure legends). The power calculations are shown in percent, which is a bit odd when the effect sizes are shown in fold-change. Nowhere are the proper interpretation of the power values, or the effect sizes for interaction terms really laid out clearly. Overall, this makes the statistical arguments much harder to follow.

As suggested, for consistency and clarity we have now placed all p-values next to the fold-changes (FC) in the main text.

We have changed the units of the power calculations as requested, and have adopted the interpretation for these power calculations as suggested by the reviewer. We have also added an explanation and interpretation to the effect sizes for the interaction of terms into the main text at first mention in line 199-204: "There was a significant interaction between the effects of *daf-16* and *daf-2* on worm stiffness with an effect size of 2.17-fold, which was obtained by subtracting the fold-change in stiffness of *daf-2* mutants compared to wild-type from the fold-change in stiffness of double mutants *daf-2;daf-16* compared to *daf-16* mutants (Interaction effect size = 4.19 fold for FC between *daf-2* and WT – 2.02 for FC between *daf-2;daf-16* and *daf-16*). Together with a significant value of $p < 0.0001$ for the interaction of terms, our data suggest that the effects of *daf-2* on stiffness maintenance with age are modulated by *daf-16* (Figure 2C, Table S1).

First off, the authors need only invoke the power analyses in cases where they aim to interpret "change is not statistically significant" to mean "change is of small magnitude". In these cases, one might write something like "in condition X, YM was increased 6-fold compared to WT. In condition Y, the increase in YM was not statistically significant (fold-change=1.3, $p=0.14$). Post-hoc analysis suggests that this experiment was powered to detect a fold-change ≥ 2.4 (or ≤ 0.42), indicating that the true effect of condition Y is likely under 2.4-fold, which is well less than the effect of condition X".

This seems wordy (and would be needed in full detail only once), but anyway there really are very few cases remaining where the authors interpret non-significant p-values in this way. Moreover, this full logic is probably not really necessary in cases where condition X has a significant fold-change in one direction, but condition Y has a non-significant fold change in the opposite direction. In that case, one doesn't really need the power analysis to make the case that X and Y are qualitatively different.

As suggested by the reviewer, we have removed the power analysis where not needed and adopted their interpretation into the text where remained. The power by which a fold change is detected as significant was assumed 0.8.

The authors should be commended for adding additional statistical analysis, but they should not just "sprinkle the statistics on at the end" -- the statistical analysis ought to be clearly integrated into the interpretation of the results.

We have added more interpretations to the text including those suggested by the reviewer and made great efforts to integrate the statistical analysis with the interpretation of the results. These are all highlighted in green.

(2) One example of unclearly explained effect size for interaction term can be found on line 275: "the interaction of terms: metformin and bacterial strain was also significant (effect size of OP50-MR = 7.57, Figure 3D, Table S1)." What are the units? Is this effect size large or small compared to the non-interaction terms? The authors should seek out other similar cases and fix them.

As suggested, to increase clarity we have introduced in the main text an explanation for the interaction effect size (see under point 1).

(3) The entries for the columns in Table 1 aren't explained at all. It could definitely use an expanded legend. This has now been corrected and can be found in lines 1387-1392.

(4) Overall the authors' distinction between bacterial "growth" vs. "proliferation" is super-unclear, and never really explained in clear language anywhere. The authors seem to be using "growth" to mean "increase in number of particles over a defined time" by measuring the area under the OD600 curve. In contrast, "proliferation" is used to mean "maximum rate of increase in number of particles based on the OD600 curve". I'm not sure this is really standard terminology, nor have the authors explained the interpretation of changes to one vs. the other value. We have now included a clarification in the main text referring to these two proliferation parameters (lines 328-330) and modified the text accordingly. Changes to the text can be found in green.

(5) I'm a bit confused about the claim on lines 438-442: "We found that the effects on lifespan induced by a *B. subtilis* diet were mediated by *daf-16* (CPH<0.001 for the interaction of terms: bacterial diet and host genotype, Figure 5A, 5E, Table S2), however, the effects mediated by a *C. aquatica* diet were not (CPH=0.1661 for the interaction of terms: bacterial diet and genotype Figure 5A, 5E, Table S2)." But visual inspection of the figures 5A and E tell a somewhat different story: in WT, *subtilis* extends lifespan and *aquatica* shortens it, compared to OP50. In *daf-16(-)*, all three bacteria yield very similar median lifespans, though *subtilis* and *aquatica* each show much more squared-off lifespan curves. So it certainly seems like eating *aquatica* does something different to *daf-16(-)* worms compared to wild-type. This does suggest some kind of interaction with *daf-16*, even if the CPH model doesn't capture it. (Does that model properly deal with survival curves that cross? I don't believe so...)

We agree and thank the reviewer for highlighting it. Since the curves from the *B. subtilis* conditions also behave in the same manner, we have now included the comparative figures in S7B and S7C and modified the text accordingly in lines 456 to 460 to read: "Since the Cox proportional-hazards statistical model cannot resolve and compare survival curves that cross (Figure S7B), close inspection of the longevity curves of both the WT and mutants in these dietary conditions suggest an interaction between *daf-16* and both dietary interventions in mediating longevity effects (Figure 5A, 5E, S7B)"

Reviewer #2 (Remarks to the Author):

The authors have addressed all remaining concerns and added necessary Supplemental Material.

REVIEWERS' COMMENTS:

Reviewer #1 (Remarks to the Author):

The statistics are more cogently presented; at this point I have no further concerns.